# ERG-driven prostate cancer initiation is cell-context dependent and requires KMT2A and DOT1L

Weiran Feng [1,10,11,12] ✉, Erik Ladewig[2,12], Matthew Lange [1], Nazifa Salsabeel[1], Huiyong Zhao[3], Young Sun Lee[1], Anuradha Gopalan[4], Hanzhi Luo[5,10,11], Wenfei Kang[6], Ning Fan[6], Eric Rosiek [6], Elisa de Stanchina[3], Yu Chen [1], Brett S. Carver [1,7,8], Christina S. Leslie [2] & Charles L. Sawyers [1,9] ✉

Despite the high prevalence of ERG transcription factor translocations in prostate cancer, the mechanism of tumorigenicity remains poorly understood. Using lineage tracing, we find the tumor-initiating activity of ERG resides in a subpopulation of murine basal cells that coexpress luminal genes (Basal^Lum) and not in the larger population of ERG⁺ luminal cells. Upon ERG activation, Basal^Lum cells give rise to highly proliferative intermediate (IM) cells with stem-like features that coexpress basal, luminal, hillock and club marker genes, before transitioning to Krt8⁺ luminal cells. Transcriptomic analysis of ERG⁺ human prostate cancers confirms the presence of rare ERG⁺ Basal^Lum cells, as well as IM cells whose presence is associated with a worse prognosis. Single-cell analysis revealed a chromatin state in ERG⁺ IM cells enriched for STAT3 transcription factor binding sites and elevated expression of the KMT2A/MLL1 and DOT1L, all three of which are essential for ERG-driven tumorigenicity in vivo. In addition to providing translational opportunities, this work illustrates how single-cell approaches combined with lineage tracing can identify cancer vulnerabilities not evident from bulk analysis.

Multiple lines of evidence implicate translocations in ERG, which are present in ~40–50% of prostate cancers (PCa) in Western cohorts, as an initiating event in PCa. For example, ERG expression is seen in prostatic intraepithelial neoplasia (PIN) and proliferative inflammatory atrophy, lesions that can precede the development of cancer, and is uniform across all cells. Experimental models, notably through transgenic expression of ERG in the mouse prostate, establish that ERG is sufficient to induce PCa, particularly in the context of PI3K pathway activation[1–9].

Despite the availability of several models of ERG-driven disease initiation[3,10–29], the mechanism by which ERG initiates PCa remains unclear. ERG binding sites in chromatin, as well as ERG-induced transcriptional changes, have been extensively annotated across a range of mouse and human models, but these datasets fail to reveal a clear sequence of downstream events responsible for oncogenicity. RNA sequencing (RNA-seq) analysis has shown that ERG activates a pro-luminal epithelial differentiation program, with consequent loss of basal epithelial cells[3,10,17,27]. But the question of how activation of this pro-luminal program is coupled with oncogenesis remains unclear. Cooperativity of ERG with androgen receptor (AR) has been documented at the level of chromatin binding but the consequences on AR target gene activation can be activating, re-directing or inhibitory depending on the cellular context[3,10,27,29,30]. Of note, datasets in these earlier studies were generated using bulk tissue methodologies, raising the possibility that critical changes relevant to tumor initiation may be overlooked if they occur only in subpopulations of cells. Lineage tracing

and tissue recombination studies have shown that tumors can initiate in basal cells or luminal cells[31–40]. In the context of ERG translocations, these cell of origin distinctions could be relevant in defining the critical cellular context required for ERG function and uncover dependencies. Recent advances in multiomic technology provide an opportunity to explore these questions at a single-cell level.

As a first step, we used a combination of mouse organoid transplantation and lineage tracing to address whether ERG-mediated prostate tumorigenesis initiates in basal cells or luminal cells. Both approaches implicate basal cells as the preferred cell of origin, with the important nuance that initiation occurs in a rare subpopulation of basal cells, which we call Basal[Lum], based on coexpression of a subset of luminal marker genes including *Tmprss2* and *Nkx3.1*. Rare ERG-expressing Basal[Lum] cells are also detectable in ERG[+] human PCa. Within weeks following ERG activation, Basal[Lum] cells give rise to a highly proliferative population of IM cells expressing basal and luminal genes, which subsequently give rise to a larger population of mature luminal cells that account for the bulk of invasive PCa seen in mice. Through single-cell transcriptomic and chromatin accessibility profiling, we find these ERG[+] IM cells have a unique, ERG-specific chromatin landscape with increased chromatin accessibility of binding sites for the STAT3 transcription factor (TF) and elevated expression of two histone methyltransferase genes, *Kmt2/Mll1* and *Dot1l*. In orthotopic transplantation assays, we find that STAT3, KMT2/MLL1 and DOT1L are all required for ERG-dependent prostate tumorigenicity in vivo, raising the possibility of pharmacologic intervention using existing clinical-grade inhibitors[41,42]. Importantly, expression signatures derived from IM cells in ERG[+] genetically engineered mouse models (GEMMs) are enriched in ERG[+] human PCa and correlate with shorter disease-free survival, suggesting that cell of origin of the ERG translocation may impact outcome.

## Results

### ERG-driven prostate cancer preferentially initiates in basal cells

We previously reported a robust mouse model of ERG-driven PCa (*Rosa26-ERG[LSL/LSL]*; *Pten[flox/flox]*; *Pb-Cre4*, hereafter called *EPC*)[3] in which ERG is expressed in basal and luminal prostate epithelial cells. To explore whether these tumors preferentially arise from basal or luminal cells (or both), we used a previously reported method to interrogate cell of origin by collecting primary mouse prostate tissue, sorting into luminal and basal subpopulations, genetically manipulating each subpopulation ex vivo, then immediately (same day) reintroducing these cells orthotopically to score subsequent tumor formation[43]. To address the cell of origin using ERG, we isolated luminal and basal cells from multiple lobes of the prostates of *Rosa26-ERG[LSL/LSL]*; *Pten[flox/flox]* (*EP*) mice, activated the *EP* genotype by infection with Ad-Cre virus and performed orthotopic implantations (Fig. 1a). Post-sort analysis confirmed >85% purity of the sorted populations preimplantation and

>90% Cre recombination efficiency (Extended Data Fig. 1a). Although luminal cells are intrinsically fragile, our approach enabled successful engraftment of luminal-derived orthografts as indicated by the detection of ERG[+] cells after 5 months. However, ERG[+] cells were present at a much higher rate in basal-derived grafts (~45% versus ~11%) (Fig. 1b). Furthermore, only the basal-derived orthografts developed invasive adenocarcinomas (6 of 6) (Fig. 1c and Extended Data Fig. 1b,c). Histologically, the basal-derived orthografts displayed luminal morphology, with glandular architecture and evidence of nuclear and nucleolar enlargement. By contrast, luminal-derived orthografts were smaller, less penetrant (4 of 6) and displayed benign histology (Fig. 1c and Extended Data Fig. 1b,c). These findings are consistent with earlier work using human prostate epithelial cells, which showed superior engraftment of basal cells in tissue recombination experiments following infection with lentivirus expressing ERG + activated AKT + AR (ref. 31).

Having implicated basal cells as cell of origin in this transplantation assay, we turned to conventional lineage tracing approaches to validate our findings in an autochthonous model. Due to a severe, early-onset skin phenotype in *EP* mice crossed with *K5-CreER[T2]* or *K14-CreER* mice (Extended Data Fig. 1d–j and Supplementary Note 1), we delivered Cre locally by intraprostatic injection of adenoviruses expressing Cre downstream of K5 (Ad-K5-Cre), together with a parallel set of experiments targeting luminal cells using Ad-K8-Cre virus (Fig. 1d). To evaluate the fidelity and robustness of Cre delivery, we used *Rosa26-YFP[LSL/LSL]* reporter mice as a control. At 1 week after injection of Ad-K5-Cre, we found all YFP[+] cells were K5[+] with a flat-shaped basal morphology. By contrast, nearly all YFP[+] cells in Ad-K8-Cre-injected mice were K8/18[+] with a columnar luminal morphology (Extended Data Fig. 1k). Having demonstrated the lineage specificity of the Ad-K5 and Ad-K8-Cre viruses, we performed similar injections in *EP* mice (two injections per mouse into either anterior or dorsal lobes), then followed each cohort for disease onset. In Ad-K5-Cre-injected mice, expansion of ERG[+] cells occurred at 12 of 13 injection sites, with histologic evidence of invasive adenocarcinoma in half (Fig. 1e,f and Extended Data Fig. 1l). Despite targeting ERG activation in K5[+] basal cells, the ERG[+] invasive cells had luminal morphology with glandular architecture (Fig. 1e,f). By contrast, we did not observe consistent expansion or invasion of ERG[+] cells in mice injected with Ad-K8-Cre virus (small foci were seen in 2 of 13 mice) (Fig. 1e,f and Extended Data Fig. 1l). Thus, both the transplantation and in situ models implicate basal cells as the preferred cell of origin for ERG-driven PCa, whereas luminal cells support limited, noninvasive cell expansion.

### A rare ERG[+] basal subset exists in ERG[+] human prostate cancer

ERG expression in ERG[+] human PCa is under the control of the TMPRSS2 regulatory locus, a gene highly expressed in normal luminal epithelial cells. Having shown that ERG-driven mouse PCa initiates in basal cells

**Fig. 1 | ERG-driven prostate cancer preferentially initiates from an NKX3.1[+] subset of basal cells but not luminal cells. a**, Orthotopic transplantation comparing *EP* orthografts freshly derived from basal versus luminal origin. **b**, Flow cytometry quantifying ERG[+] cells in grafts collected at the 5-month endpoint (*n* = 6 mice per group). **c**, Histological analysis of orthografts at 5-month endpoint. Inset: a high-power view. Scale bars, 100 µm. **d**, An intraprostatic adenoviral (Ad) injection approach to compare *EP* activation by Ad-K8-Cre versus Ad-K5-Cre. **e,f**, Histology (**e**) and characterization (**f**) of *EP* prostates collected at 40 weeks postinfection (*n* = 13 mice per group). Inset: a high-power view. PCa is defined as invasive adenocarcinoma. Scale bars, 100 µm. **g**, UMAP of epithelial cells from normal human prostates[52] (cell types annotated in circles). Cells from normal samples are colored based on gene expression, with cells from tumor samples in gray. **h**, Intracellular flow cytometry quantifying YFP[+] basal cells (*n* = 4 mice). **i**, Prostate IF highlighting YFP[+]K5[+] cells. Inset: a high-power view. Scale bars, 25 µm. **j**, Diagram highlighting distinct morphology and cytokeratin expression of normal prostate epithelial cell types. **k**, UMAP of epithelial cells from patients[52] highlighting the presence of ERG[+] cells, as colored

by ERG expression (left) and patient ID (right). ERG[+] PCa samples are shown (Methods). Basal and PCa cells are annotated in black and pink circles/arrows, respectively. **l**, Lineage tracing activating *EP* from indicated population. **m**, Flow cytometry quantifying ERG[+] cells from indicated mice (left to right: *n* = 3, 2, 4, 8, 8, 6 mice). **n**, Prostate IF. Dashed line encircles clusters of K5[+]ERG[+] cells. Arrows highlight the K5-single-positive (K8/18-negative) cells at the gland periphery. Inset: a high-power view. Scale bars, 50 µm. **o**, Prostate histology. High-power view (insets) highlights cell morphology. Scale bars, 500 µm. **p**, PCa-free survival analysis (top to bottom, *n* = 20, 12, 19 mice). PCa is defined as invasive adenocarcinoma originating from any region of the prostate. Data represent mean ± s.d.; unpaired two-tailed *t*-test (**b**); chi-squared test (**f**); two-way ANOVA with Tukey posttest (**m**); log-rank test (**p**). ANOVA, analysis of variance; Bas, basal; Bas[Lum], Basal[Lum]; DN, double-negative; DP, double-positive; H&E, hematoxylin and eosin; IF, immunofluorescence; Lum, luminal; mo, month(s); NS, not significant; SP[bas], single-positive for basal markers; SP[lum], single-positive for luminal markers; Tam, tamoxifen; wk, week(s).

(using a R26 lox-STOP-lox cassette), we explored the human relevance of this result by asking whether TMPRSS2 is expressed in basal cells by examining single-cell RNA sequencing (scRNA-seq) datasets of normal mouse and human PCa. As expected[44–47], TMPRSS2 was uniformly expressed in nearly all luminal subpopulations. However, we also noted small numbers of TMPRSS2+ cells residing in basal clusters in both human

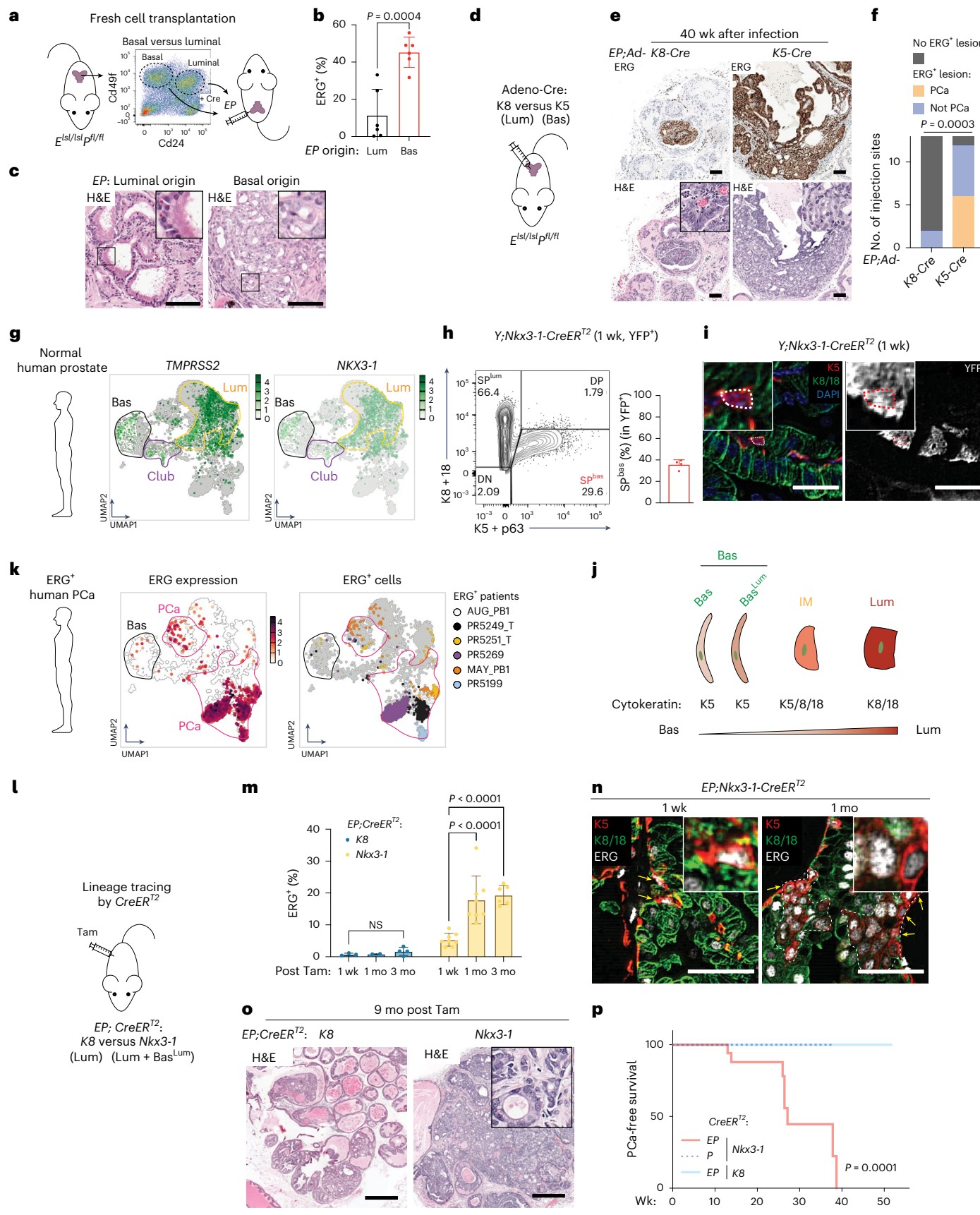

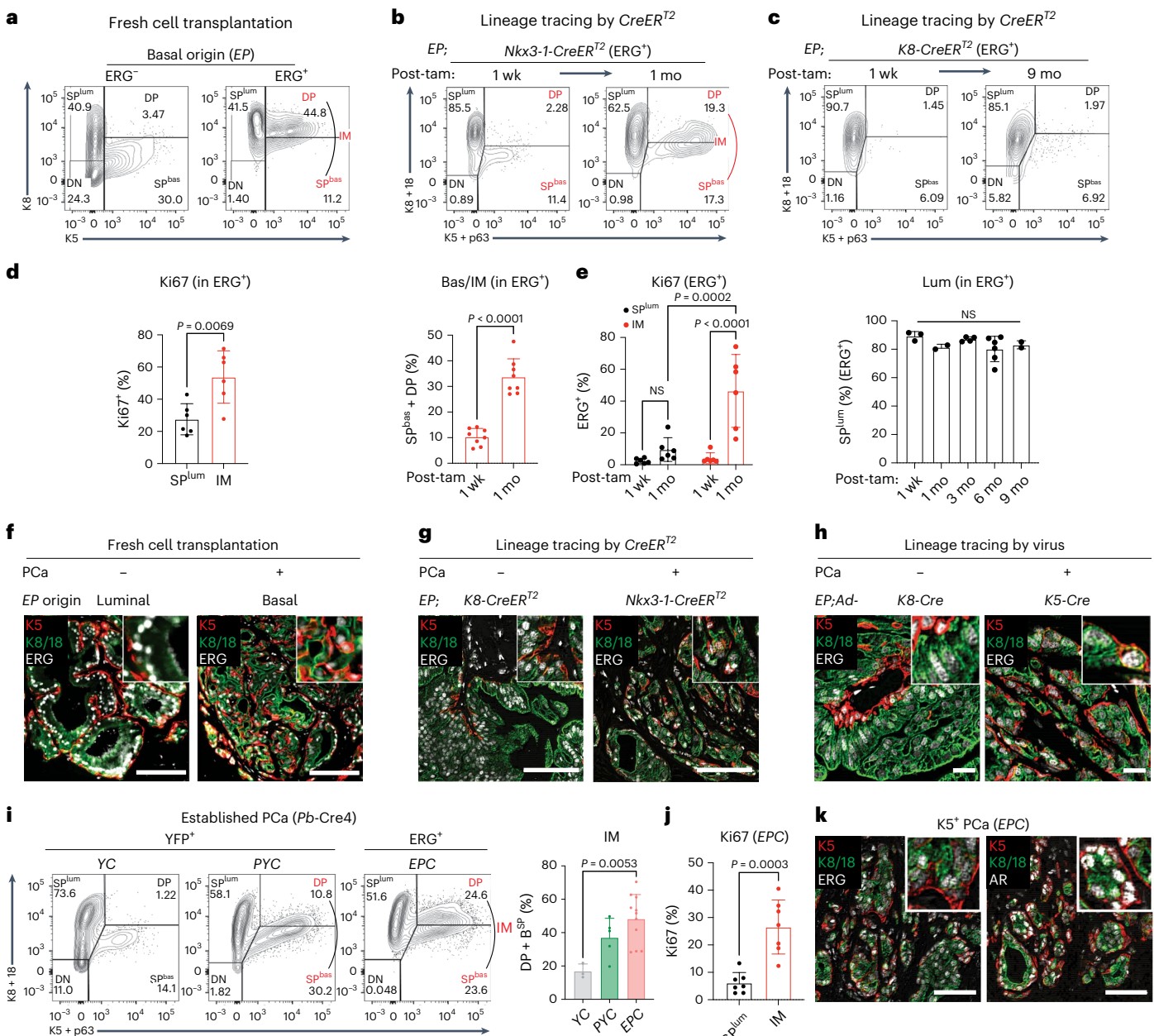

**Fig. 2 | Basal cells expand into highly proliferative basal–luminal IM cells in cancer. a,** Flow cytometry quantification of indicated populations from orthografts from Fig. 1a. The DP and SP$^{bas}$ quadrants in ERG$^+$ cells form one continuous population and are hereafter grouped together as IM cells. **b,** Flow cytometry (top) and quantification (bottom) of populations from ERG$^+$ cells from whole prostates of indicated mice ($n = 8$ mice per group). **c,** Flow cytometry quantifying luminal populations of ERG$^+$ cells from whole prostates of indicated mice (left to right: $n = 3, 2, 4, 6, 2$ mice). **d,** Flow cytometry quantifying Ki67$^+$ cells from ERG$^+$ populations of basal-derived *EP* orthografts from **a** at the 5-month endpoint ($n = 6$ mice). **f,** IF images of indicated *EP* orthografts from **a** at the 5-month endpoint. Inset: a high-power view. Scale bars, 100 μm. **g,** IF images of indicated mice. Inset: a high-power view. Whole prostates were collected at 9 months after tamoxifen. Scale bars, 100 μm.

**h,** IF of *EP* prostates collected at 40 weeks after infection. Inset: a high-power view. PCa from **f**–**h** is defined as invasive adenocarcinoma. Scale bars, 20 μm. **i,** Flow cytometry (left) and quantification (right) highlighting the expansion of IM populations in *Pten$^{flox/flox}$;Rosa26-YFP$^{LSL/LSL}$;Pb*-Cre4 (*PYC*) and *EPC* mice (left to right: $n = 3, 5, 11$ mice). Whole prostate cells were collected at 3 months of age. Recombined cells labeled by YFP (*YC/PYC*) or ERG (*EPC*) were analyzed. **j,** Flow cytometry quantification of Ki67$^+$ cells from indicated population ($n = 7$ mice). Whole prostate cells were collected at 3 months of age. **k,** IF staining showing expression of ERG (left) and AR (right) in both K5$^-$ luminal and K5$^+$ IM cells from invasive adenocarcinomas. Whole prostate tissues were collected from 3-month *EPC* mice. Inset: a high-power view. Scale bars, 50 μm. Data represent mean ± s.d.; unpaired two-tailed *t*-test (**b,d**); one-way ANOVA (**c**); two-way ANOVA with Sidak posttest (**e**); one-way ANOVA with Tukey posttest (**i**). *YC, Rosa26-YFP$^{LSL/LSL}$;Pb-Cre4*.

and mouse normal prostates. Furthermore, these basal cells express other canonical luminal genes such as *NKX3-1, NDRG1, FKBP5, KLK3* and *Pbsn* (Fig. 1g and Extended Data Fig. 2a–c), indicative of a mixed identity. Of note, expression of these genes is regulated by androgen in basal cells, just as seen in luminal cells[48] (Extended Data Fig. 2d).

To provide additional evidence for luminal gene expression in basal cells, we crossed *Nkx3-1*-CreER$^{T2}$ mice into the *Rosa26-YFP$^{LSL/LSL}$* background. The largest population of YFP$^+$ cells were luminal (K8/18$^+$), but a subset of basal cells (K5$^+$) clearly express YFP (Fig. 1h). Because these K5$^+$, P63$^+$, YFP$^+$ cells have a flat morphology typical of basal cells

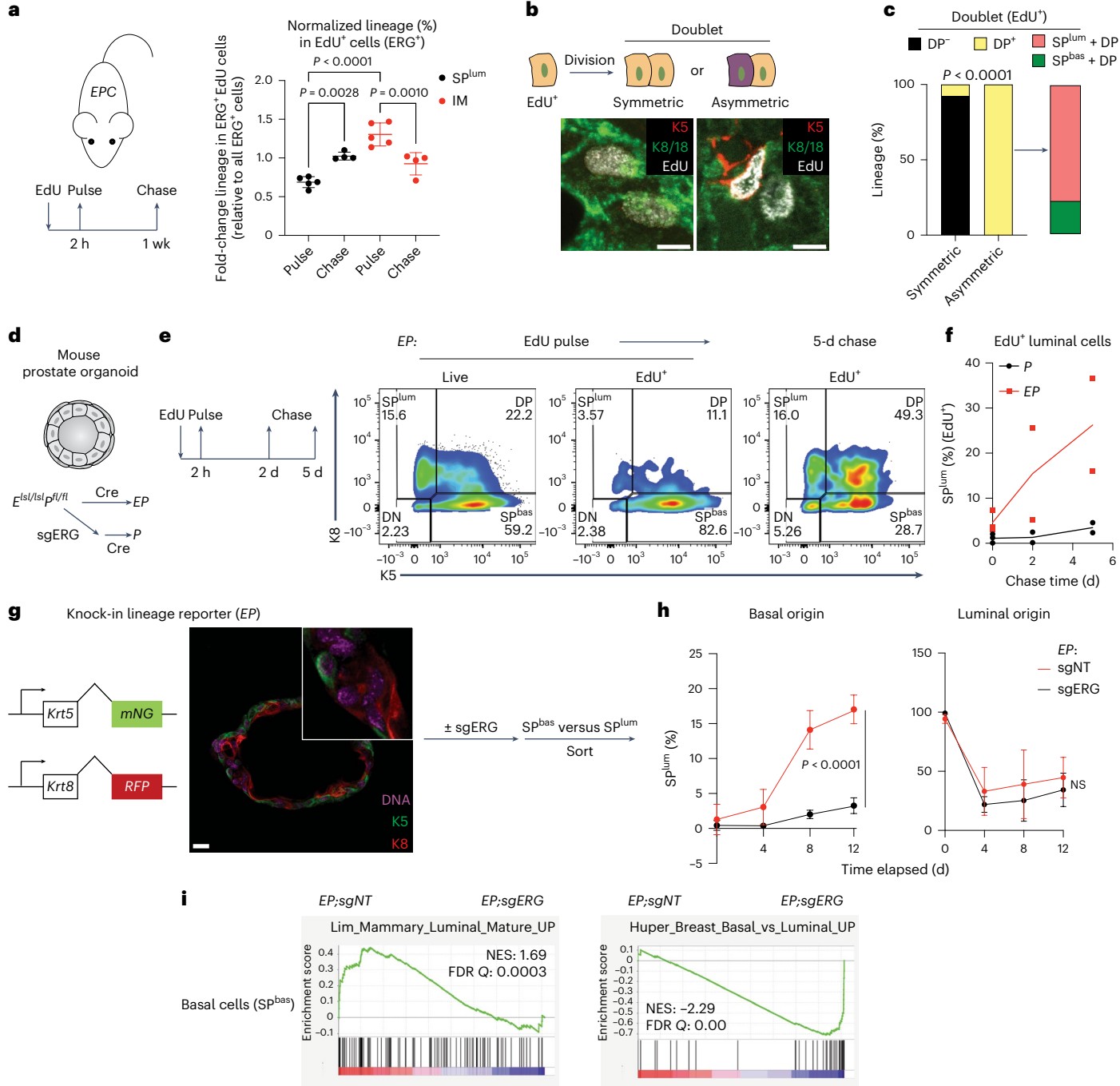

**Fig. 3 | IM and basal cells transit to a luminal fate in the presence of ERG.**
**a**, EdU pulse–chase assay on *EPC* mice at 2 months of age (schematic on the left). Indicated lineage representation was analyzed from whole prostate cells on the right (*n* = 5 mice for pulse, *n* = 4 mice for chase). The EdU+ population of ERG+ cells (normalized to all ERG+ cells) shows an increase of SP^lum cells and decrease of IM after the chase. **b**, Schematic (top) and representative IF images (bottom) exemplifying symmetric and asymmetric divisions of EdU-labeled cells using *EPC* mice from **a**. Scale bars, 5 μm. **c**, The cell type composition of EdU+ doublets highlights that DP cells are present in all asymmetric divisions and preferentially associate with an SP^lum daughter cell (*n* = 7 mice). **d**, Schematic showing in vitro generation of an isogenic pair of *EP* and *P* organoids. **e**,**f**, EdU pulse–chase assay in organoids followed by flow cytometry assessment highlights an ERG-dependent shift of the EdU-labeled cells towards a luminal (SP^lum) fate (*n* = 3 independent assays for pulse, *n* = 2 independent assays for

chase). Representative data (**e**) and quantification of the luminal fraction in EdU-labeled populations (**f**) are shown. **g**, Left: schematic showing *EP* organoids with a dual lineage reporter knocked in to form C-terminal fusions with endogenous K5 and K8. Middle: live cell imaging highlighting the expected spatial expression pattern of the reporter signals, which enables live sort of basal and luminal cells (schematic on the right). Scale bar, 20 μm. **h**, The basal- and luminal-derived *EP* reporter organoids from **g** were analyzed by flow cytometry based on the reporter signals along the time course postsort (*n* = 3 independent assays). An ERG-dependent luminal fate transition from basal cells from luminal cells was observed. **i**, GSEA showing ERG-regulated pathways in basal cells. Data represent mean ± s.d.; one-way ANOVA with Tukey posttest (**a**); two-sided Fisher's exact test (**c**); two-way ANOVA (**h**). mNG, mNeonGreen; NES, normalized enrichment score; RFP, TagRFP.

yet are adjacent to luminal cells at the basement membrane (Fig. 1i and Extended Data Fig. 2e), we refer to them henceforth as Basal[Lum] cells (Fig. 1j). Basal cells with a similar profile of luminal gene expression (called Basal-B or Lum[I]) were recently reported by others[49,50].

We next asked whether ERG[+] human PCa, which typically display uniform ERG expression in luminal cells, also have evidence of ERG expression in rare basal cells by annotating ERG and TMPRSS2 expression in two independent published scRNA-seq datasets of localized PCa (total PCa n = 29; ERG-positive PCa n = 9)[51,52]. As expected, large clusters of ERG[+] luminal cancer cells are present in the ERG[+] patients in both datasets, but we also observed rare ERG[+] cells in the corresponding basal clusters of these same patients (Fig. 1k and Extended Data Fig. 3). Basal cell expression of TMPRSS2 has been previously documented in normal human prostate tissues, in primary cultures derived from ERG-positive human tumors and in mice engineered to express ERG from human *TMPRSS2* BAC construct[53–55].

## ERG drives expansion of Nkx3.1[+] basal cells

Having implicated basal cells as a preferred cell of origin for ERG-positive PCa and demonstrated *Tmprss2* expression in a subset of these basal cells (Basal[Lum]), we next asked whether these tumors originate from Basal[Lum] cells using the *Nkx3-1*-CreER[T2] driver allele[38,56,57]. Because *Nkx3.1* is expressed in luminal cells as well as a subpopulation of basal cells[37], we compared the phenotype of ERG expression in *Nkx3-1*-CreER[T2] mice with that seen using the luminal-restricted *K8-CreER[T2]* driver to provide insight into consequences of ERG expression in Nkx3.1[+] Basal[Lum] cells (Extended Data Fig. 1d)[58,59]. Within 1 month of tamoxifen induction, ERG[+] cells expanded more than threefold (~5% to ~18% at 1 month) in *EP;Nkx-1-CreER[T2]* mice but not in *EP; K8-CreER[T2]* mice (Fig. 1l,m). Initially, ERG[+] basal cells in *EP;Nkx-1-CreER[T2]* mice are individual, flat-shaped, K5[+] cells adjacent to the basement membrane, but, by 1 month, progress to clusters of K5[+] single-positive and K5[+], K8/18[+] double-positive cells (Fig. 1n), which we call IM cells based on previous use of this term to describe rare (<1%) K5[+]/K8[+] double-positive cells in normal prostate tissue[60–67]. After 9 months, all *Nkx3-1-CreER[T2];EP* mice developed invasive adenocarcinomas featuring luminal histology (Fig. 1o,p). By contrast, ERG[+] cells did not expand in mice with luminal-restricted ERG expression (*EP; K8-CreER[T2]*) or generate invasive cancer after 9–12 months despite early and sustained luminal-specific ERG induction throughout the entire observation period (Fig. 1m,o,p). The only histologic changes in *EP; K8-CreER[T2]* mice were hyperplasia and PIN in nearly all mice (Fig. 1o), a phenotype seen after *Pten* loss alone[33,36,38]. These results, as well as additional experiments comparing the efficiency of different Cre drivers (Extended Data Fig. 4 and Supplementary Note 2), lead us to conclude that ERG[+] cancers preferentially initiate in Nkx3.1[+] Basal[Lum] cells.

## ERG drives basal–luminal transition via an IM state

We next sought to understand why the histologic phenotype of these cancers is nearly exclusively luminal by examining the relative proportions of ERG[+] basal (K5[+]) and luminal (K8[+]) cells at different timepoints in the mouse models discussed above (Supplementary Fig. 1). In the fresh cell transplantation model, ERG[+] cells in basal cell-derived tumors contained a mix of K8[+] single-positive and K5[+]/K8[+] double-positive IM cells (Fig. 2a). The same IM population was evident 1 month after ERG activation in the Nkx3.1-CreER[T2] lineage tracing model (where ERG is induced in luminal and Basal[Lum] cells) (Fig. 2b), but not in the

K8-CreER[T2] model where ERG expression is limited to luminal cells (Fig. 2c and Extended Data Fig. 5a). Furthermore, IM cells in the transplantation model are highly proliferative, with a larger fraction of Ki67[+] cells (40–50%) compared with K8 single-positive cells (SP[Lum]) (25–30%) (Fig. 2d,e). Coexpression of luminal (K8, K18, AR) and basal genes (K5) in IM cells was confirmed by immunofluorescence and displayed luminal morphology (Fig. 2f–h and Extended Data Fig. 5b). IM cells were also detected in invasive adenocarcinoma lesions, but only following ERG activation in basal cells (Fig. 2f–h and Extended Data Fig. 5c). IM cells in the *EPC* mice and those with *Pten* deletion alone (*Pten[flox/flox];Pb-Cre4*, hereafter called *PC*) showed similar features but only those generated in the *EPC* model result in invasion (Fig. 2i–k, Extended Data Fig. 5d–l and Supplementary Note 3). We conclude that basal cells generate highly proliferative IM cells in response to ERG activation (in the setting of PTEN loss).

To assess the lineage relationships between IM and luminal cells, we used in vivo 5-ethynyl-2′-deoxyuridine (EdU) labeling of *EPC* mice with early cancers (age 2 months) to trace the proliferative cells. As expected from their elevated proliferation rate, ERG[+] IM cells were preferentially labeled over luminal cells after a brief (2-h) EdU pulse. However, within 1 week of chase, the label shifted to K8[+], K5-negative, ERG[+] luminal cells, indicating that ERG[+] IM cells give rise to ERG[+] luminal cells (Fig. 3a and Extended Data Fig. 6a–c). To provide direct visual evidence for this transition, we identified ~300 EdU-labeled cell doublets (dividing cells), of which ~26% showed asymmetric divisions, always with one DP daughter cell paired with an SP[lum] or SP[bas] daughter cell, suggestive of a fate transition within one cell division. Approximately 80% of these asymmetric divisions consisted of DP–SP[lum] pairs, providing further evidence that DP cells give rise to SP[lum] cells (Fig. 3b–c). Additional experiments using prostate organoid cultures confirm luminal lineage specification by ERG induction in basal cells (Fig. 3d–i, Extended Data Fig. 6d–l, Supplementary Table 1 and Supplementary Note 4). Collectively, ERG-driven cancers initiate in basal cells, which expand initially as highly proliferative IM cells then transition to luminal cells that typify human PCa.

## IM cells represent a multi-lineage, stem-like state

Having implicated Basal[Lum] cells as the cell of origin for ERG-driven PCa, we next turned to single-cell analysis to gain insight into the gene expression and chromatin landscape changes in these cells following ERG induction. We selected *EPC* mice for these studies given the potent ERG-driven phenotype and ERG expression in basal and luminal cells, which ensures representation of both cell types in the analysis. At age 3 months, EPC mice have small foci of invasive cancer which progress within 6 months to highly penetrant glandular invasion (Extended Data Fig. 7a). By contrast, *PC* mice develop intraductal hyperplasia and/or PIN over this 3–6-month time frame[3,68].

To capture early neoplastic changes, we performed scRNA-seq of whole prostate tissue from *EPC*, *PC* and *WT* mice at age 3 months, which revealed genotype-specific differences in epithelial and nonepithelial populations (Fig. 4a–c and Extended Data Fig. 7b–f). For example, *EPC* mice have much more pronounced myeloid infiltration than *PC* and *WT* mice[69] (Extended Data Fig. 7d–f). Focusing on the epithelial cells (Epcam[+]), we noted that large clusters of secretory luminal cells, which account for >90% of luminal cells in *WT* mice (called Lum_L1)[48,70,71], are absent in *EPC* and *PC* mice. By contrast, a distinct subset of luminal cells

**Fig. 4 | IM cells represent a multi-lineage, stem-like state. a**, Experimental design including *WT* (1 mouse), *PC* (2 mice) and *EPC* (2 mice). Whole prostates from indicated mice at 3 months of age were collected for scRNA-seq analysis. **b,c**, UMAP of epithelial cell clusters with all genotypes together (**b**) or assigned to each genotype (**c**). Luminal and IM clusters specific to mutant samples (*PC* and *EPC*) are highlighted in circles and defined as Lum_Mut and IM cells, respectively. Cluster IDs are shown. A cluster with low expression of epithelial lineage markers is defined as lineage-negative. **d**, UMAP plotting the *ERG* transgene expression.

**e**, Cell cycle scores (top) and GSEA (bottom) across all epithelial clusters. The clusters are numbered according to the UMAP in **b**. Pathway enrichment is calculated via GSEA using clusterProfiler and shown for those with an FDR-adjusted *P* < 0.05. **f**, Violin plots comparing canonical prostate lineage marker expression across all epithelial clusters. The clusters are numbered according to the UMAP in **b**. **g,h**, Transcriptional programs in indicated cell types from single-cell transcriptomes in mice (**g**) and human (**h**). LN, lineage-negative; C2, MSigDB Collection 2.

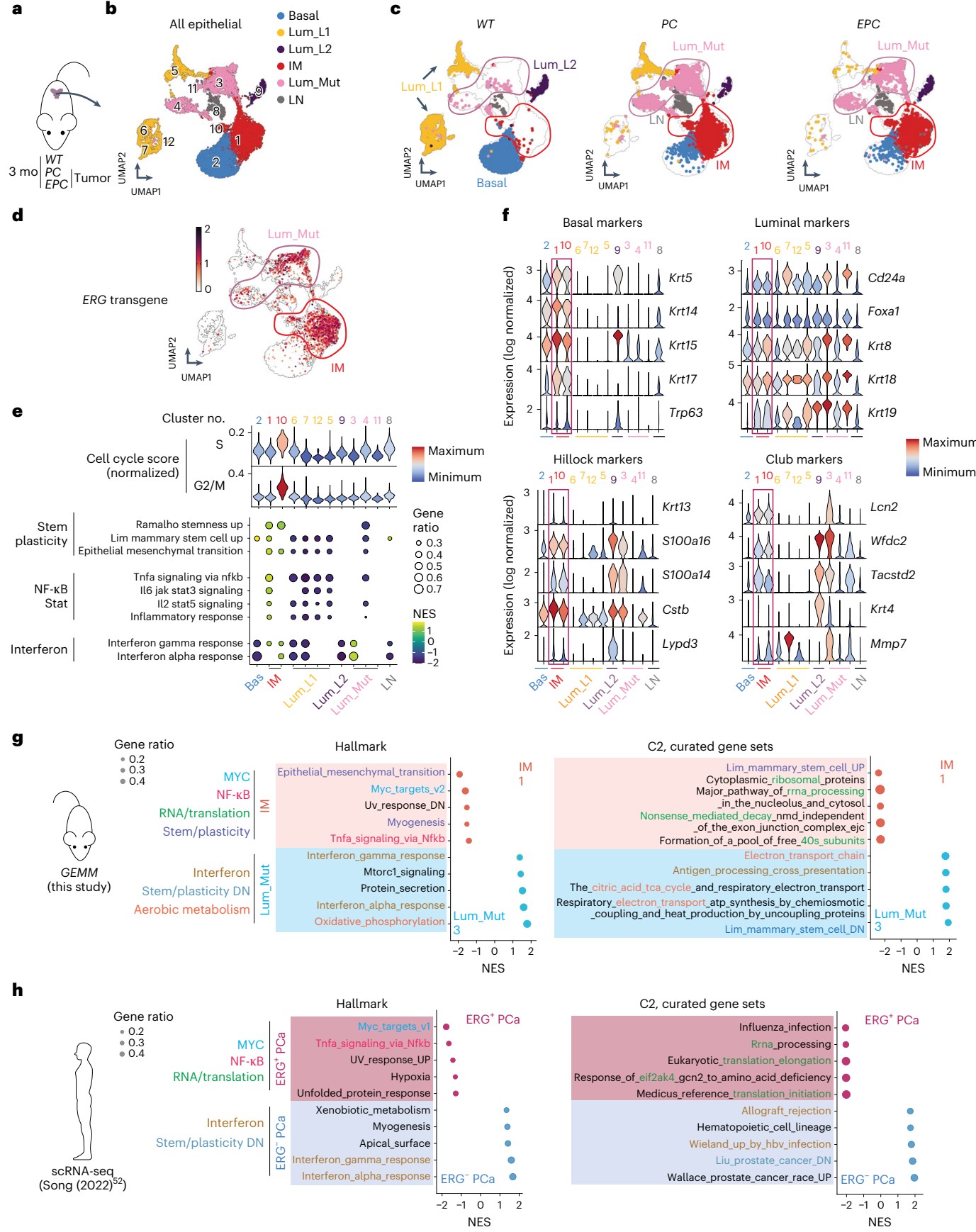

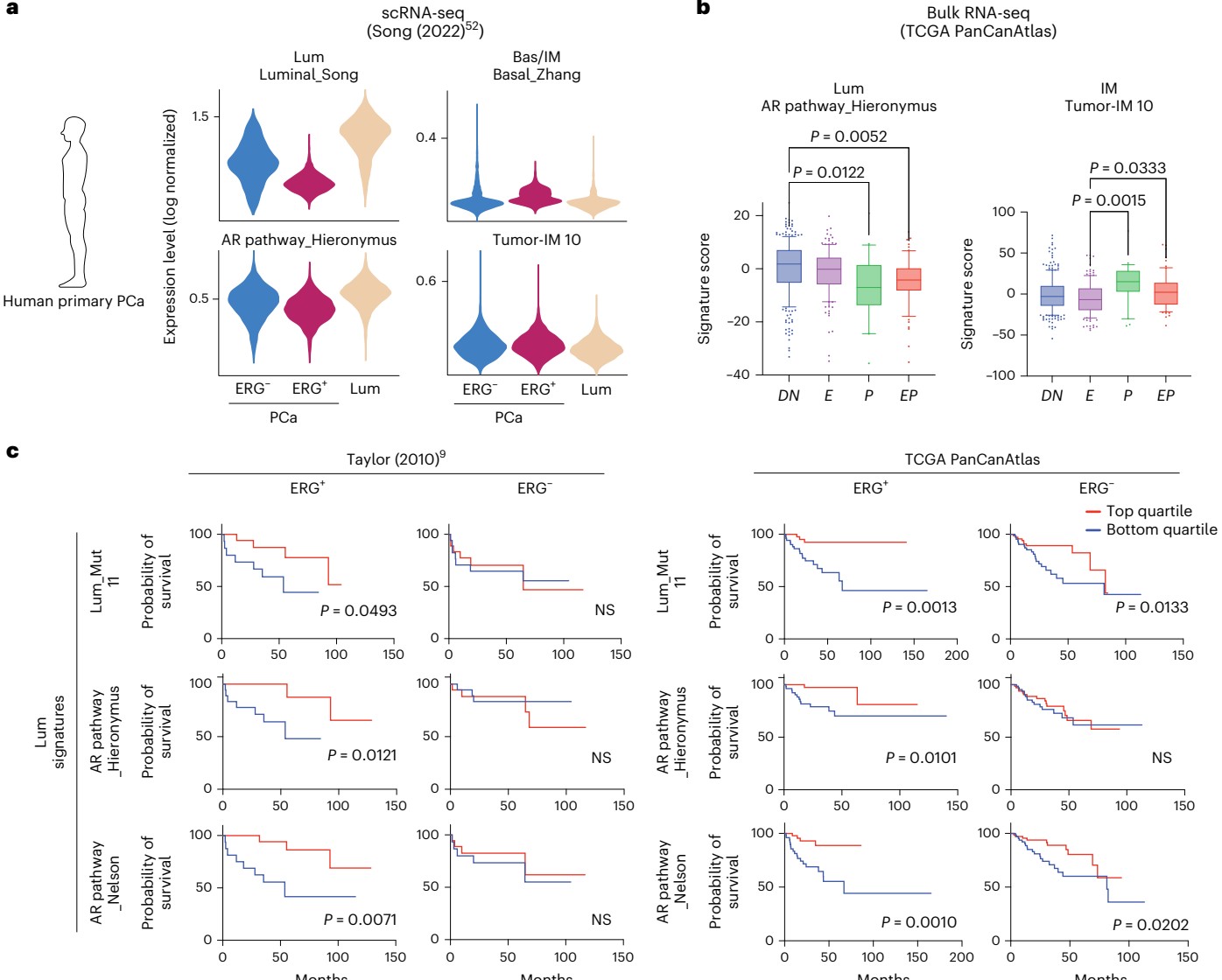

**Fig. 5 | IM signatures are associated with a worse outcome in ERG⁺ patients.** **a**, Expression of basal/IM and luminal signatures across indicated scRNA-seq cell types from human primary PCa[52]. **b**, Expression of basal/IM and luminal signatures in bulk RNA-seq data of primary human PCa, separated based on ERG and PTEN statuses. E, ERG-fusion-positive ($n = 141$); P, PTEN deep deletion ($n = 23$); EP, ERG-fusion-positive and PTEN deep deletion ($n = 62$); DN, ERG-fusion-negative, PTEN deep deletion-negative ($n = 268$). The center line represents the

median, the box limits represent the upper and lower quartiles, and the minimum and maximum whiskers represent the 10th and 90th percentiles, respectively. **c**, Progression-free survival outcome using indicated signatures from two independent cohorts of patients with primary PCa, stratified based on ERG status. Data represent mean ± s.d.; one-way ANOVA with Tukey posttest to correct for multiple comparisons (**b**); log-rank (Mantel–Cox) test (**c**).

(also known as Lum_L2 or LumP) that express stem-like markers (Sca1, Trop2) and have increased regenerative potential[48,70,71] are more abundant in *EPC* and *PC* mice (Fig. 4a–c and Extended Data Fig. 7g–j) (together designated Lum_Mut)[72]. Most striking, however, are two cell clusters (1, 10) that coexpress basal (*Krt5/14/15/17, Trp63*) and luminal (*Krt8/18/19, Cd24a, Foxa1*) lineage genes, which we collectively refer to as IM, based on earlier use of this terminology (Fig. 4d–f and Extended Data Fig. 7k).

Differential gene expression analysis of IM cells revealed elevated stemness, MYC, NF-κB, Stat signatures, cellular processes related to RNA processing and translation, as well as growth factor signaling specifically in IM cells in *EPC* mice (Fig. 4e,g, Extended Data Fig. 7l and Supplementary Tables 2 and 3). Cluster 10 is noteworthy for increased S and G2/M cycle scores and elevated levels of Ki67, Top2a and Pcna, indicative of high proliferation. IM cells also express marker genes seen in Hillock and club cells (Fig. 4e,f), two stem-like epithelial subtypes of lung and prostate involved in tissue regeneration[73–76].

## Relevance of IM state in ERG⁺ human prostate cancer
To determine whether the expanded IM population seen in GEMMs is relevant in human PCa, we first verified that gene sets identified through single-cell analysis of our *EPC* mice are present in single-cell data from ERG⁺ human PCa (Fig. 4g,h, Extended Data Fig. 8a, Supplementary Table 3 and Supplementary Note 5). To explore whether the IM cell state in ERG⁺ GEMMs is present in human PCa, we first noted that IM cells in EPC mice have reduced AR target gene signature expression compared with luminal cells, consistent with the fact that they are in transition between basal and luminal cell states (Extended Data Fig. 8b). Similarly, ERG⁺ human PCa (particularly those with concurrent PTEN loss) display reduced AR pathway activation, reduced luminal signatures and elevated GEMM-derived IM signatures, as measured transcriptionally in single-cell and bulk RNA-seq datasets (Fig. 5a,b). We next asked whether luminal or IM cell state signatures derived from ERG⁺ GEMMs might distinguish between different clinical outcomes in ERG⁺ human

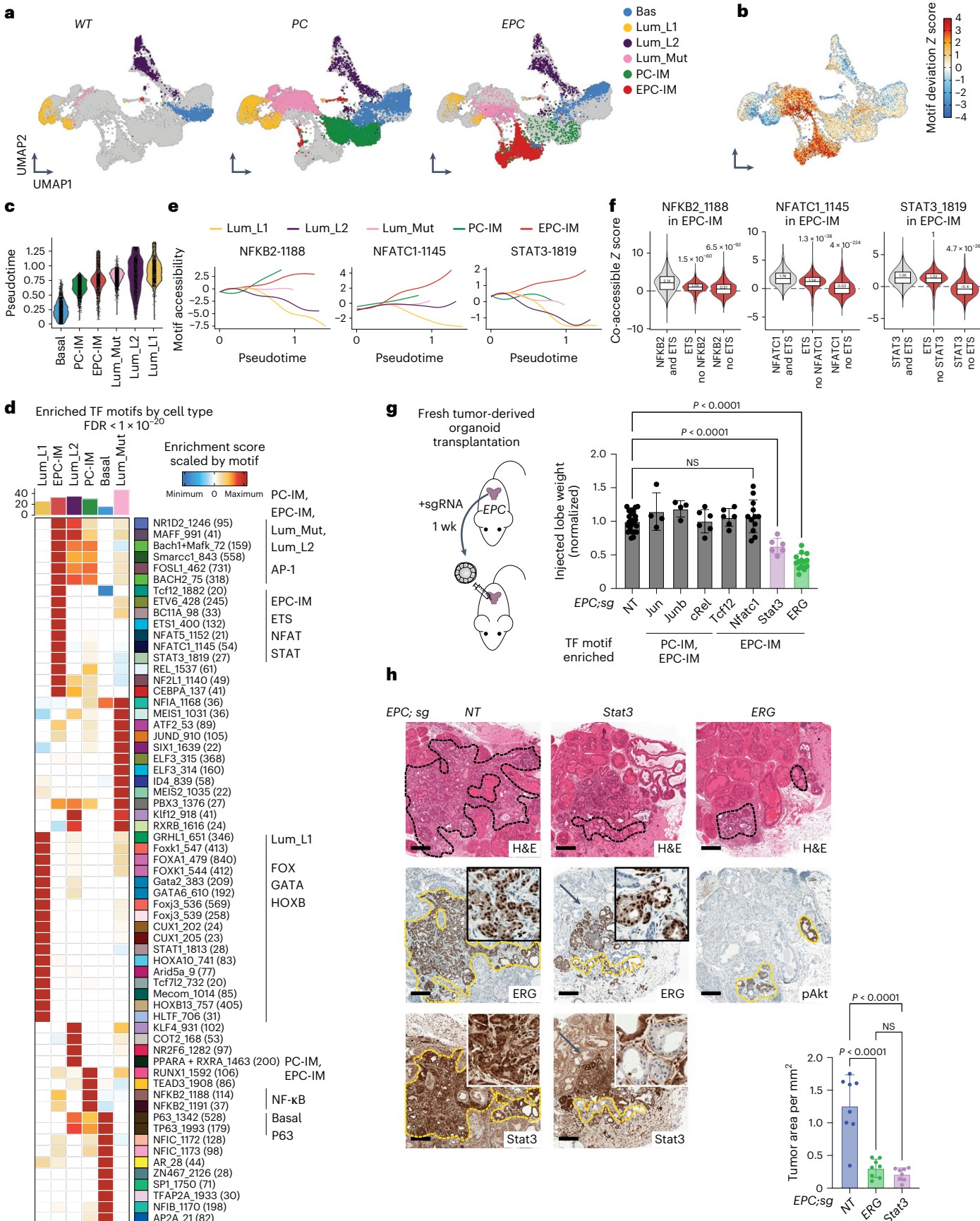

**Fig. 6 | ERG drives a unique chromatin state in IM cells featuring STAT3 and other TFs. a**, UMAP of scATAC-seq showing epithelial cell clusters assigned to each genotype ($n = 16,006$ total number of epithelial cells). Dorsal–lateral lobes from indicated mice at different ages (4 weeks, 3 months and 6 months) were pooled together and analyzed (Extended Data Fig. 9a). **b**, Single-cell ChromVar enrichment of ETS family (ERG, ETS, ETV, FLI) motifs. **c**, Violin plots showing cells grouped by cell type on the *x* axis and arranged by pseudotime on the *y* axis. **d**, Heatmap of TF motifs enriched per cell type. Hypergeometric tests were performed for each motif's accessibility within each cell type compared with all other cell types. Motifs with an FDR-adjusted $P < 10^{-20}$ are shown. Major cell-type-specific motif families are highlighted. **e**, Trajectories of three EPC-IM-enriched motifs. **f**, Co-accessible ChromVar *Z* scores of indicated motifs in EPC-IM cells with or without ETS factors. Boxplots are centered at the median value and the lower and upper hinges correspond to the interquartile range (IQR) at the 25th and 75th percentiles, respectively. The lower and upper whiskers extend from the hinge to the smallest or largest value at most 1.5 × IQR of the hinge, respectively. FDR-adjusted *P* values above the violin were calculated with one-sided Wilcoxon rank sum test using ChromVar *Z* scores of peaks with co-occurring motifs versus exclusively one motif. **g**, Left: schematic showing individual genes perturbed in fresh tumor-derived organoids (*EPC*) before orthotopic transplantation. Whole prostates of *EPC* mice (3–9 months) were pooled, edited and shortly expanded as organoids before orthotopic transplantation into host dorsal lobes. Right: normalized weight of injected lobes from indicated orthografts (left to right: $n = 26, 4, 4, 6, 6, 12, 6, 14$ grafts). TFs are categorized based on motif accessibility enrichment in indicated clusters. **h**, Histological analysis on orthografts collected at the 1-month time point (8 grafts per group). Tumor areas were quantified based on ERG/Stat3/pAkt status as highlighted in yellow dashed lines. Arrows in the *EPC;sgStat3* group highlight ERG+ cells that retain Stat3 expression. Scale bars, 250 μm. Data represent mean ± s.d.; one-way ANOVA with Dunnett posttest (**g**); one-way ANOVA with Tukey posttest (**h**).

patients with PCa by interrogating baseline tumor transcriptomes (bulk RNA-seq) in two widely studied cohorts. Because luminal tumor cells should be highly represented in these samples, we first performed the analysis using Lum_Mut signatures (derived independently from GEMM clusters 11, 3 and 4), or using either of two widely used human AR pathway signatures. ERG+ patients whose tumors scored in the top luminal signature quartile had longer disease-free survival than those with ERG+ tumors that scored in the bottom luminal quartile (Fig. 5c and Extended Data Fig. 8c), an association not seen in ERG-negative tumors. We performed the same analysis using the GEMM-derived IM signature, as well as a previously reported human basal stem cell signature, to ask whether the reciprocal association (IM signature and worse outcome) was observed. ERG+ patients in the top quartile had shorter disease-free survival than those in the bottom quartile (Extended Data Fig. 8c). Although the strength of the IM signature association was less pronounced than for the Lum_Mut signature, we find it remarkable considering the IM signature is likely present in only a subpopulation of cells but is sufficient to generate a signal detected through bulk RNA-seq analysis. With the caveat that these associations are based on retrospective analysis, the prognostic signal seen here using IM and luminal cell state signatures raises the provocative hypothesis that the cell of origin in which the ERG translocation occurs (basal or luminal) could impact outcome.

## The ERG-unique IM chromatin state reveals epigenetic dependencies

The lineage tracing, single-cell profiling and cell state signature analyses reported here all point to the importance of the IM population. To gain further mechanistic insight into this cell state, we profiled chromatin accessibility changes within the *EPC* model using single-cell assay for transposase-accessible chromatin with high-throughput sequencing (scATAC-seq). Cell identity of the resulting assay for transposase-accessible chromatin (ATAC) clusters was inferred by aligning scRNA-seq and scATAC-seq datasets using anchor-based integration (Methods). Notably, a prominent cluster unique to *EPC* mice was detected and was transcriptionally defined as IM cells (Fig. 6a and Extended Data Fig. 9a–h). This cluster (hereafter called EPC-IM) is highly enriched for ETS (putatively ERG) binding motifs (Fig. 6b), implicating ERG as the primary contributor to the shift in chromatin landscape. The ETS binding motif is also enriched in *EPC* Lum_Mut cells but, in contrast, luminal cells of *EPC* and *PC* mice (EPC-Lum and PC-Lum) belong to the same scATAC-seq cluster (Fig. 6a,b and Extended Data Fig. 9b). To examine the relationship between cell types, we performed trajectory analysis with Palantir[77], starting with a cell in the basal cluster based on the lineage tracing data (Methods). The calculated trajectories suggest EPC-IM cells precede and may ultimately give rise to Lum_Mut cells (Fig. 6c and Extended Data Fig. 9i), consistent with the lineage tracing studies described earlier (Fig. 3).

The scATAC-seq findings, as well as the scRNA-seq and lineage tracing studies described earlier, collectively point to the EPC-IM cluster as the cell population where ERG activates a tumor initiation program. To gain insight into how this happens, we searched for additional binding motifs enriched within this cluster to identify a TF that might cooperate with ERG. As a quality control for the robustness of this approach, clusters from *WT* mice with inferred basal and luminal identity showed enrichment of known TF motifs including P63 and FOX/GATA/HOXB13, respectively[44,45,78–83] (Fig. 6d). Two patterns emerged in IM clusters: (1) NF-κB and AP-1 motifs enriched in PC-IM and EPC-IM clusters, and (2) STAT and NFAT motifs enriched only in *EPC* mice (Fig. 6d–f, Extended Data Fig. 9j,k and Supplementary Note 6).

The analysis of enriched motifs raises the possibility of cooperativity between ERG and one or more of the above TFs. To test this hypothesis at a functional level, we performed orthotopic transplantation experiments using fresh tumor-derived organoids from *EPC* mice in which we deleted either a candidate TF using single guide RNAs (sgRNAs), or a nontargeting sgRNA as a control, using weight of the injected lobe as a quick readout of disease burden (Fig. 6g). As expected, ERG ablation impaired graft weight, establishing the robustness and ERG

**Fig. 7 | ERG oncogenicity depends on KMT2A/MLL1 and DOT1L but not MENIN. a**, Violin plots comparing inferred expression of a panel of histone-modifying enzymes across all epithelial clusters from scATAC-seq. **b**, Box plot showing expression of indicated genes in human PCa from two public datasets, separated by ERG-fusion status. $n = 181$ (ERG−), 152 (ERG+) for TCGA; $n = 108$ (ERG−), 100 (ERG+) for SU2C. The center line represents the median, the box limits represent the upper and lower quartiles, and the minimum and maximum whiskers represent the 10th and 90th percentiles, respectively. **c**, DepMap dependency scores of human prostate cell lines. A negative CRISPR score indicates dependency ($n = 8$ for ETS−, $n = 2$ for ETS+ lines). ETS TF positive cell lines, VCaP (ERG+) and LNCaP (ETV1+), are highlighted. The center line represents the median, the box limits represent the upper and lower quartiles, and the minimum and maximum whiskers represent the minimum and maximum, respectively. **d**, Luminal output quantified by flow cytometry (left to right: $n = 7, 2, 2, 2, 3$, 3 independent assays). *EP* reporter organoids were treated with indicated guide RNAs and analyzed 2 weeks later. **e**, Schematic showing orthotopic transplantation assay using engineered tumor-derived organoids. Whole prostates of *EPC* mice (3–9 months) were pooled, edited and shortly expanded in organoid culture before transplantation into dorsal lobes of host mice (2 months old). Prostates were collected 5-7 weeks after grafting. **f–h**, Histology and tumor area quantification of orthografts with Kmt2a (**f**), Dot1l (**g**) and Men1 (**h**) perturbation. Left to right: $n = 16, 8, 6$ grafts (**f**); $n = 14, 12, 6, 14$ grafts (**g**); $n = 6, 2, 6, 6$ grafts (**g**). Tumor areas were quantified as highlighted, based on ERG (**f**), ERG and loss of H3K79me2 (**g**), ERG and loss of Menin (**h**). H3K79me2 immunohistochemistry is shown in Extended Data Fig. 10c. Scale bars, 250 μm. Data represent mean ± s.d.; two-tailed unpaired *t*-test (**b,c**); one-way ANOVA with Dunnett posttest (**d**); one-way ANOVA with Tukey posttest (**f–h**). Neg, negative; Pos, positive.

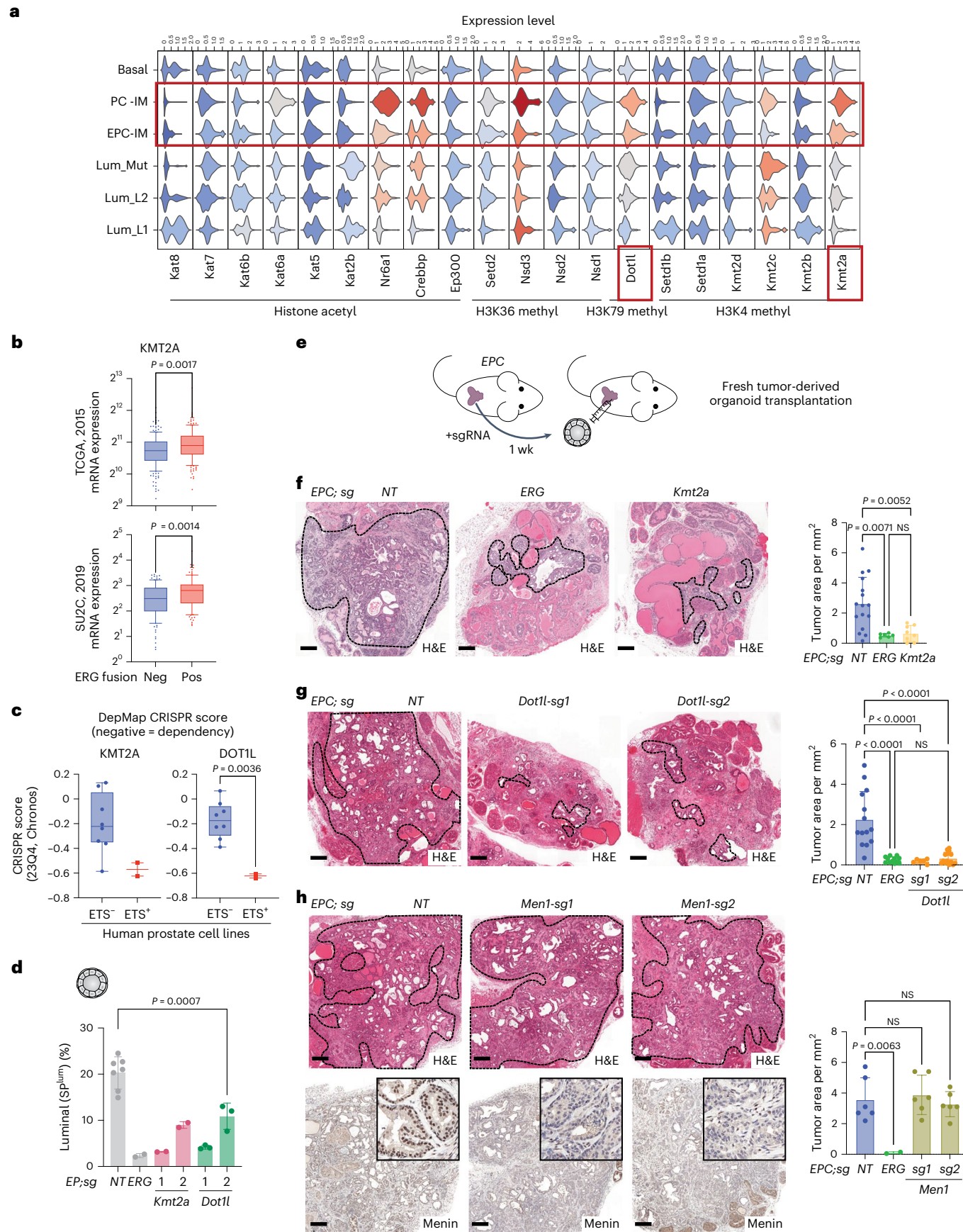

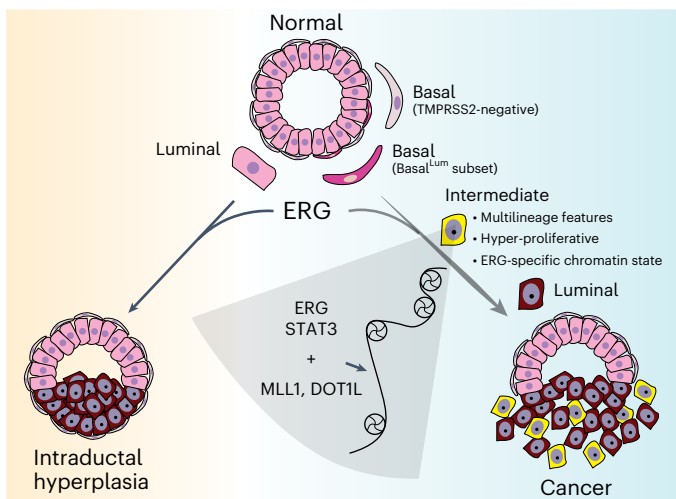

**Fig. 8 | Model on cell context dependency of ERG oncogenicity.** ERG translocations (in the context of PTEN loss) that occur in a subset of basal cells with luminal transcriptomic features (Basal^Lum cells) enter a highly proliferative multi-lineage IM state that, in turn, drives invasive cancer with luminal differentiation. By contrast, ERG translocations that originate in luminal cells may develop intraductal hyperplasia but fail to progress to invasive cancer. The basal-derived IM cells provide a chromatin context to support maximal ERG-driven chromatin changes, featuring TFs (NF-κB, AP-1, NFAT, STAT) in addition to ERG and epigenetic regulators (MLL1, DOT1L).

dependence of the assay. Of six candidate TFs tested (representing AP-1, NF-κB, NFAT and STAT), only *Stat3* deletion led to a reduction in graft weight similar to that seen with ERG deletion (Fig. 6g). We therefore focused on STAT3 for further analysis, recognizing that AP-1 and NF-κB cannot be fully eliminated as candidates due to redundancy of other proteins within the same family that might compensate for single gene deletion. Prostate adenocarcinomas developed in all mice transplanted with nontargeting (NT) sgRNA organoids but not in those in which either ERG or Stat3 was deleted (Fig. 6h). The few residual areas of ERG^+ adenocarcinoma that retained Stat3 expression likely represent cells that escaped Stat3 knockout (Fig. 6h).

Having defined an ERG-specific chromatin context associated with tumor initiation, we postulated that the transcriptional programs activated by ERG likely involve one or more chromatin modifying enzymes. Using the single-cell data we identified two candidates (Kmt2a, Dot1l) highly and preferentially expressed in IM cells from *PC* and *EPC* mice (Fig. 7a). Of these, KMT2A messenger RNA expression also correlated with ERG-positive human PCa (Fig. 7b). Furthermore, the only two ETS-positive PCa cell lines examined in DepMap both show selective dependency on both KMT2A and DOT1L relative to seven ETS-negative PCa cell lines (Fig. 7c). Finally, sgRNA knockdown of either *Kmt2a* or *Dot1l* impaired ERG-induced luminal lineage specification in the organoid assay described earlier, providing evidence both enzymes play a role in ERG function (Fig. 7d) (see also Fig. 3h).

In light of the development of pharmacological inhibitors for KMT2A (Menin inhibitors)[84–86] and DOT1L[41,87,88], we evaluated both *Kmt2a* and *Dot1l* in the same ERG-dependent in vivo tumorigenicity model used earlier for *Stat3*. Remarkably, both *Kmt2a* and *Dot1l* deletion impaired the growth of *EPC*-derived tumors to a comparable magnitude as that seen with *ERG* deletion (Fig. 7e–g and Extended Data Fig. 10a). As with the Stat3 experiments, the small foci of residual adenocarcinoma remaining in *Kmt2a* and *Dot1l* sgRNA orthografts retained target protein expression (Kmt2a/Mll1) or activity (H3K79me2 for Dot1l) (Extended Data Fig. 10b,c), further supporting the critical dependency of ERG-driven cancer on *Kmt2a/Mll1* and *Dot1l*. Interestingly, *Men1* (encoding Menin) sgRNA did not impair

ERG-dependent tumor growth despite efficient reduction of Menin protein levels in tumors (Fig. 7h and Extended Data Fig. 10d), raising the possibility of ERG/KMT2A cooperativity through a mechanism independent of Menin.

## Discussion

ERG translocations are the presumed driver event in nearly half of PCa in Western cohorts (~150,000 new cases per year), yet we have limited insight into precisely how ERG initiates PCa. This lack of understanding is, in part, due to limited availability of human PCa cell lines, patient-derived xenografts or organoid models that retain ERG expression. Through a combination of lineage tracing experiments coupled with cell-type-specific activation of ERG in basal versus luminal cells, we find that ERG-driven PCa initiate in a rare subset of basal cells present in healthy mouse and human prostates (which we call Basal^Lum cells) that coexpress various canonical luminal lineage genes including, importantly, TMPRSS2. Upon ERG activation, Basal^Lum cells give rise to the highly proliferative IM cells, which subsequently differentiate into invasive luminal adenocarcinomas (Fig. 8). These findings help refine previous conclusions using various mouse and human models that implicate both basal and luminal cells as cells of origin for PCa[31–40].

Taking advantage of recent advances that make it possible to profile transcriptomic and chromatin landscape changes at single-cell resolution, we then interrogated disease initiation and progression over time in an ERG GEMM model that accurately models ERG-driven disease initiation and progression. Through this analysis we identified a subfraction of highly proliferative epithelial cells with multi-lineage features (basal, luminal, hillock, club) (IM cells) that appear within weeks of ERG activation and give rise to the invasive luminal adenocarcinomas that represent the clinical manifestation of the disease. Postulating that this IM cell subpopulation reflects a cellular context optimally primed to respond to the oncogenic potential of ERG, we searched for transcriptomic and chromatin landscape features uniquely present in ERG^+ IM cells versus ERG^+ luminal cells. This analysis revealed an ERG-specific chromatin state exclusively in IM cells, with selective enrichment in regions of newly open chromatin of binding motifs for ERG, NF-κB, AP-1, NFATC1 and STAT3, as well as elevated expression of the lysine methyltransferases KMT2A/MLL1 and DOT1L (Fig. 8).

To test whether these IM-specific differences in chromatin modifying enzyme expression and TF motif accessibility are linked with the oncogenic potential of ERG, we developed an ERG-dependent in vivo tumorigenicity assay through which we could efficiently score the consequences of perturbing each candidate (via CRISPR deletion). STAT3, KMT2A/MLL1 and DOT1L emerged as clear hits from these in vivo studies, with all three blocking tumorigenicity at levels comparable to that seen with ERG deletion. The fact that ERG remains robustly expressed in regions where *Stat3*-, *Kmt2a*- or *Dot1l*-deleted cells persist (as small noninvasive clusters) suggests that ERG may partner with all three proteins to activate tumorigenicity programs. Furthermore, prostate data from DepMap, albeit limited due to the small number of prostate cell lines evaluated, suggest that ERG/ETS^+ human PCa cell lines may be selectively dependent on KMT2A and DOT1L (Fig. 7c).

The fact that pharmacological inhibitors of KMT2A/MLL1, DOT1L and STAT3 are in various stages of clinical investigation begs the question of whether these compounds may have activity in PCa[41,84,85,88,89]. KMT2A/MLL1 has been previously implicated as a dependency in castration resistance PCa due to its role as a co-activator of AR signaling[90]. The in vivo sensitivity to Menin/MLL1 inhibition reported in this earlier work is based on ERG^+ (VCaP) and ETV1^+ (LNCaP/AR) PCa xenograft models, raising the possibility that these phenotypes may in fact be a consequence of ETS-specific dependencies. It is also worth noting that KMT2/MLL1 dependency has been reported for gastrointestinal stromal tumors[91], a KIT-driven malignancy that is also ETS dependent based on the lineage requirement for ETV1 in interstitial cells of Cajal, the cell of origin for gastrointestinal stromal tumors[92]. Thus,

several lines of evidence from different tumor types and model systems suggest a dependency of ETS-driven malignancies on KMT2A/MLL1 function. One critical caveat regarding the allograft data reported here is the ablation of ERG-driven tumorigenicity by Kmt2a/Mll1 deletion but not by Menin deletion (Fig. 7f–h). Further work is required to determine whether the *Kmt2a/Mll1*-dependent, *Menin*-independent phenotype seen here is specific to the model used here. In any case, further studies of KMT2A/MLL1, DOT1L and STAT3 dependency across a range of ETS-driven mouse and human tumor models are warranted, particularly with the potential for near-term translational impact using existing clinical-grade inhibitors.

In closing, our work expands the notion that cancers can initiate in a cell with a different lineage identity from the resultant tumor, a concept first established in chronic myeloid leukemia, where the BCR–ABL translocation initiates in a hematopoietic stem cell but the clinical manifestation of the disease is elevated levels of terminally differentiated myeloid cells (neutrophils)[93,94]. Furthermore, our identification of a highly proliferative, stem-like population with a unique chromatin state may provide an opportunity to reveal ERG-specific dependencies that could be exploited therapeutically (see Supplementary Notes 7 and 8 for further discussion of cell context and study limitations).

## Online content

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

[1]Human Oncology and Pathogenesis Program, Memorial Sloan Kettering Cancer Center, New York, NY, USA. [2]Computational and Systems Biology Program, Memorial Sloan Kettering Cancer Center, New York, NY, USA. [3]Antitumor Assessment Core Facility, Memorial Sloan Kettering Cancer Center, New York, NY, USA. [4]Department of Pathology, Memorial Sloan Kettering Cancer Center, New York, NY, USA. [5]Molecular Pharmacology Program, Memorial Sloan Kettering Cancer Center, New York, NY, USA. [6]Molecular Cytology Core Facility, Memorial Sloan Kettering Cancer Center, New York, NY, USA. [7]Department of Surgery, Memorial Sloan Kettering Cancer Center, New York, NY, USA. [8]Division of Urology, Memorial Sloan Kettering Cancer Center, New York, NY, USA. [9]Howard Hughes Medical Institute, Memorial Sloan Kettering Cancer Center, New York, NY, USA. [10]Present address: Nuclear Dynamics and Cancer Program, Fox Chase Cancer Center, Philadelphia, PA, USA. [11]Present address: Cancer Epigenetics Institute, Fox Chase Cancer Center, Philadelphia, PA, USA. [12]These authors contributed equally: Weiran Feng, Erik Ladewig. ✉e-mail: weiran.feng@fccc.edu; sawyersc@mskcc.org

## Methods

### Ethical statement

Mouse experiments were conducted under protocol 0607012 approved by the Institutional Animal Care and Use Committee of Memorial Sloan Kettering Cancer Center (MSKCC), New York.

### Mouse strains and dosing

*Rosa26-ERG*[LSL/LSL], *Pten*[flox/flox], *Pb-Cre4* and *Rosa26-YFP*[LSL/LSL] (*Gt(ROSA)26Sor*[tm3(CAG-EYFP)Hze]*/J*) mice have been previously described[3,95–97].

For tamoxifen-inducible Cre recombination, *K8-CreER*[T2] (*Tg(Krt8-cre/ERT2)17Blpn/J*)[58], *K5-CreER*[T2] (*Krt5*[tm1.1(cre/ERT2)Blh]*/J*)[58], *K14-CreER* (*Tg(KRT14-cre/ERT)20Efu/J*)[98] or *Nkx3-1-CreER*[T2] (*Nkx3-1*[tm4(cre/ERT2)Mms]*/AbshnJ*)[37] mice were crossed to the desired floxed strains and introduced as hemizygous/heterozygous alleles. To induce CreER or CreER[T2]-mediated recombination, tamoxifen (Sigma) was suspended in corn oil and injected intraperitoneally to mice at 7–9 weeks of age. For *K14-CreER, K8-CreER*[T2] and *Nkx3-1-CreER*[T2], mice were treated with two doses of 3 mg of tamoxifen per 25 g of body weight on day 1 and day 3, respectively. For *K5-CreER*[T2], mice were administered one dose of 0.5 mg of tamoxifen per 25 g of body weight. For EdU pulse–chase experiments, 2-month-old mice were treated with EdU (40 mg kg$^{-1}$; Thermo Fisher) by intraperitoneal injection.

The maximal burden for prostate tumors allowed in our protocol is when animals show signs of abdomen distension or other signs of disease/distress, none of which has been observed throughout the course of our study.

### Histology and immunostaining

For GEMM experiments, whole mouse prostates containing all lobes were collected at autopsy. Histological data from all lobes were pooled for quantification. For intraprostatic transplantation and adenoviral infection experiments, injected prostate lobes were collected at autopsy. Prostate tissues were fixed using 4% paraformaldehyde, dehydrated with 70% ethanol, paraffin-embedded and sectioned. H&E staining was performed following standard protocols by the MSKCC Molecular Cytology Core or IDEXX. Immunohistochemistry and immunofluorescence were performed on a Leica Bond RX automatic stainer using antibodies listed in Supplementary Table 4. All formalin-fixed paraffin-embedded stained tissue was scanned using a MIRAX scanner. Histological assessments were performed blinded by an MSKCC pathologist.

### Tissue dissociation, organoid culture and fresh isolation of prostate epithelial cells

Whole mouse prostates containing all lobes were isolated as described previously[99,100]. Briefly, prostates were collected and subsequently digested with collagenase type II (Gibco) for 2 h at 37 °C, followed by TrypLE (Gibco) digestion at 37 °C until a single-cell suspension was obtained. Digestions were supplemented with Y-27632 (10 µM) to inhibit anoikis and filtered to obtain a single-cell suspension.

Murine prostate organoid culture was established and maintained under standard conditions as described previously[99,100]. Briefly, dissociated prostatic epithelial cells were embedded in 20-µl drops of Matrigel (Corning) and overlaid with mouse prostate organoid medium. Organoids were passaged by mechanical splitting by repetitively passing through a 200-µl pipette tip or by trypsinization using TrypLE. For lineage marker analysis, organoids were trypsinized into single cells and seeded at 20,000–50,000 cells per well of a 12-well plate in the absence of EGF (counted as day 0). For EdU pulse–chase experiments, the organoid culture was incubated with EdU (10 µM, Thermo Fisher) for 2 h before collecting (pulse) or wash-off and re-seeding (chase). To compare cell fate from reporter cells, basal and luminal cells were sorted based on single-positive reporter signals and seeded at equal cell number between sgNT- and sgERG-expressing organoids in the absence of EGF (counted as day 0). Basal cells were seeded at 20,000–50,000 cells per well of a 12-well plate; fewer luminal cells were seeded (5,000–10,000 per well) due to a low recovery from the sgERG group.

To isolate basal and luminal cells from donor prostates[101], freshly dissociated prostate cells were stained for 1 h on ice with CD49f-PE (1:200; BD, cat no. 555736) and CD24-Alexa Fluor 647 (1:200; BioLegend, cat. no. 101818). Sorted basal and luminal cells were transduced with Cre-expressing adenovirus (Ad-CMV-iCre, Vector Biolabs) and directly used for downstream experiments.

To induce Cre recombinase expression, adenoviral transduction was performed as described previously[100]. Briefly, 50,000 dissociated organoid cells or sorted fresh tissue cells were mixed with 1 µl of Ad-CMV-iCre ($1.0 \times 10^7$ plaque-forming units per µl, Vector Biolabs) in 500 µl of growth medium. The suspension was spin-infected at 600$g$ for 1 h before a PBS wash and collection of the infected cells for downstream experiments. The amount of virus used was scaled according to the number of cells available.

### Orthotopic transplantation and intraprostatic adenoviral infection

For orthotopic transplantation, 250,000–500,000 freshly sorted prostate epithelial cells or organoid cells were resuspended in 20 µl of 50% Matrigel (Corning) and 50% organoid culture medium before injection into prostate anterior or dorsal lobes of immunodeficient NSG mice (The Jackson Laboratory) at 2 months of age.

For intraprostatic adenoviral administration, the approach was adapted from previous reports[102–104] with modifications. Ad-K5-Cre (Ad5-bK5-Cre) (A. Berns, the Netherlands Cancer Institute) and Ad-K8-Cre (Ad5-mK8-nlsCre)[105] were obtained at high titer ($1.0 \times 10^8$ plaque-forming units per µl) from the Viral Vector Core at the University of Iowa. Adenoviruses were prepared in two ways, mixed with 1/9th volume of either 10 mM CaCl$_2$ or 16 µg ml$^{-1}$ Polybrene (Millipore Sigma). Then, 10 µl of adenoviral solution was injected into either dorsal or anterior lobes of floxed mice at 2–3 months of age. Similar results were obtained which were pooled together in downstream analysis.

### Organoid engineering

Genetic perturbation in organoids was carried out by CRISPR-RNP as previously described[43]. Briefly, Cas9 protein (IDT) was first mixed with sgRNA (IDT) to form the RNP complex before nucleofection. A total of 1.2 µM cRNP was used per individual sgRNA. In total, 500,000–1,000,000 dissociated organoid cells were resuspended with nucleofection buffer, RNP complexes and electroporation enhancer (IDT, 1:1 molar ratio to RNP) in a total volume of 100 µl. The cell suspension was transferred to a nucleofection cuvette and nucleofected using Lonza Amaxa Nucleofector II (program T-030). Cells were centrifuged and seeded for culture.

To generate *EP* organoids, the parental floxed organoids (*Rosa26-ERG*[LSL/LSL]*; Pten*[flox/flox]) were transduced with Cre-expressing adenovirus (Ad-CMV-iCre, Vector Biolabs), sorted based on GFP signal (linked with ERG expression as ires-GFP) and maintained under a non-EGF condition. To generate isogenic *P* organoids, the ERG transgene was preemptively knocked out by CRISPR-RNP from the same parental floxed organoids before Cre transduction. The GFP-negative population were sorted out and maintained under a non-EGF condition. The expected ERG and Pten statuses were confirmed by intracellular flow cytometry at the protein level.

The knock-in lineage reporters were engineered by CRISPaint as previously described[106,107]. An *EP* organoid line with a stable transduction of lentiCas9-blast (Addgene plasmid no. 52962)[108] was used as the starting material. To knock in TagRFP at the *Krt8* locus, a targeting vector, pCRISPaint-TagRFP (Addgene plasmid no. 67167), was electroporated into the organoids together with CRISPR-RNPs directed to the targeting vector (sgFrame) and *Krt8* (targeting exon 9). The TagRFP$^+$ population was enriched by sorting and prepared for the next steps of engineering. To knock in mNeonGreen at the *Krt5* locus,

a targeting vector, pCRISPaint-mNeonGreen, was electroporated into the organoids together with CRISPR-RNPs directed to the targeting vector (sgFrame) and *Krt5* (targeting exon 9). The TagRFP-mNeonGreen[+] population was first enriched by sorting before a secondary enrichment for the TagRFP[+]mNeonGreen[+] population. The resulting population was used for further study without picking single clones. The knock-in products were first verified by PCR across the break junction (oligos 1 and 2 for Krt8-TagRFP; 3 and 4 for Krt5-mNeonGreen) and Sanger sequencing using the PCR primers. A further verification was performed by intracellular flow cytometry comparing the expression pattern of K8 versus the TagRFP reporter (not possible with K5 due to the epitope ablation by the targeting event). As a further validation of the Krt5-mNeonGreen engineering, Krt5 depletion by short interfering RNA (siGENOME, Horizon Discovery) led to a reduction of the mNeonGreen reporter signal. The expected spatial expression patterns of the reporters were validated by live cell confocal imaging using Leica SP5 or SP8 confocal microscopes.

sgRNA target and oligo sequences are listed in Supplementary Table 5.

### Intracellular flow cytometry

Intracellular flow cytometry was performed using a Fixation/Permeabilization Solution Kit (BD cat. no. 554714). Dissociated prostate or organoid cells were fixed and permeabilized with the fixation/permeabilization buffer for 30 min on ice, followed by incubation with primary antibodies for at least 1 h at room temperature and secondary antibodies (for nonconjugated primary antibodies) for 30 min at room temperature with washes between the steps before analysis by flow cytometry (BD Fortessa or MACSQuant 16). Dead cells were excluded using either live/dead fixable dyes (Thermo Fisher) on ice for 15 min before fixation and gated by excluding the positive population, or co-stained with DAPI during antibody incubation and gated by excluding the sub-G1 population. Where applicable, EdU[+] cells were detected using Click-iT Plus EdU Flow Cytometry Assay Kit (Thermo Fisher) following the manufacturer's instructions before antibody incubation. Antibody concentrations and catalog numbers are listed in Supplementary Table 6.

### Western blot

Boiled whole cell lysates were run on a precast Tris-acetate gel (Thermo Fisher) or a Mini-PROTEAN TGX protein gel (Bio-Rad) and then transferred to a nitrocellulose membrane (Bio-Rad). The membrane was incubated overnight at 4 °C with primary antibodies against Kmt2a (1:1,000; Cell Signaling Technology, cat. no. 14689), Menin (1:1,000; Cell Signaling Technology, cat. no. 6891S), Hsp90 (1:1,000; Cell Signaling Technology, cat. no. 4877S) and Actin-HRP (horseradish peroxidase) (1:10,000; Abcam, cat. no. ab49900). Signal was visualized with secondary HRP-conjugated antibodies and chemiluminescence detection.

### Plasmid construction

To construct pCRISPaint-mNeonGreen, mNeonGreen coding sequence was synthesized and cloned into pCRISPaint-TagRFP (Addgene plasmid no. 67167)[106] between BamHI/NotI sites to replace the TagRFP cassette. The correct cloning was confirmed by Sanger sequencing.

### Bulk RNA-seq analysis of sorted organoid cells

Basal and luminal organoid cells were sorted based on reporter signals. Three replicates were performed. RNA was extracted from bulk cells and sequenced at the Integrated Genomics Operation Core (MSKCC). Briefly, the SMARTer (Switching Mechanism At the 5′ end of RNA Template) method was performed for minimal cells RNA extraction. After amplification, complementary DNA was subjected to automated Illumina paired-end library construction. Libraries were sequenced on Illumina HiSeq2000 instruments with paired reads of 100 base pairs (bp) in length per sample. The samples were sequenced at 40–50 million

reads per sample. Sequence data were analyzed by Basepair (https://www.basepairtech.com/). Briefly, FASTQ data were aligned using the tophat software to the mouse (mm10) reference genome. Fragments per kilobase of transcript per million mapped reads were calculated using the cufflink software.

Gene set enrichment analysis (GSEA) was performed using the GSEA software (https://www.gsea-msigdb.org/gsea/index.jsp). Briefly, differentially expressed gene rank lists were generated based on gene expression profiles. Annotated gene sets were obtained from MSigDB (https://www.gsea-msigdb.org/gsea/msigdb/index.jsp) or manually curated. Enrichr pathway analyses were performed using the Enrichr website (https://maayanlab.cloud/Enrichr/)[109–111]. Gene sets generated (cutoff padj < 0.05, FC > 2) were used. The combined score was computed by taking the log of the $P$ value from the Fisher exact test and multiplying that by the $Z$ score of the deviation from the expected rank. $c = \log(p) \times z$, where $c$ is the combined score, $p$ is the Fisher exact test $P$ value and $z$ is the $Z$ score for deviation from expected rank.

### Sample preparation for single-cell analysis

For scRNA-seq, whole mouse prostates from the indicated age were freshly dissociated (described above) and flow sorted for live single cells using DAPI, before processing for scRNA-seq using 10X Genomics Chromium following the manufacture's instructions. For scATAC-seq, dorsal–lateral lobes of mouse prostates from the indicated age were freshly dissociated (described above) and flow sorted for live single cells using DAPI. Fresh nuclei were isolated before processing for 10X Genomics Chromium Single Cell Multiome ATAC + Gene Expression sequencing following the manufacturer's instructions. ATAC data were retrieved for downstream analysis.

### scRNA-seq processing and filtering

scRNA-seq was preprocessed using 10X Cell Ranger software. This included genome alignment and minimal cell filtering for RNA-seq. We used a modified genome reference to align sequencing reads to the *Mus musculus* genome (mm10) plus an additional sequence for the *ERG* transgene. After alignment, the 10X Cell Ranger software performs unique molecular identifier and read filtering to create a filtered cell by gene read count matrix. We examined this filtered count matrix to visualize cell and gene coverage distributions and to gather cell/gene statistics. Based on these metrics, several plots were visualized and used to remove poor quality cells (those with fewer than 500 transcripts) or putative multiplets (defined as cells containing greater than 100 k transcripts). Cells detected with 40% or more mitochondrial reads were removed without further consideration.

### scRNA-seq matrix normalization and variance stabilization

To account for technical variation and normalize expression data, the R (v.4.4.0) package Seurat (v.5.2.1)[112] was used on the filtered raw count matrix. SCT transform was run to perform normalization, variance stabilization and feature selection. We regressed out cell cycle scores and mitochondrial percentage in addition to Seurat defaults.

### Dimensional reduction and visualization of single-cell RNA

Dimensionality reduction and uniform manifold approximation and projection (UMAP) were applied to the count-corrected single-cell read matrix to enable a two-dimensional cell embedding. A nearest neighbor graph of cells was learned and used as input to the Leiden algorithm for cell clustering. Ten different cell clustering resolutions were evaluated and scored by silhouette score, a clustering metric. UMAPs were generated and used to hypothesize cell similarities and identify clusters for differential expression testing.

### Differential genes and pathways

Differential expression testing was performed using the R package presto (v.1.0.0) (https://github.com/immunogenomics/presto). Genes

were ranked using the average log fold change and a false discovery rate (FDR)-adjusted *P* value. A GSEA analysis with fsGSEA (v.1.30.0)[113] utilized a list of ranked genes per comparison and the collection of molecular signatures database (MSigDB).

### Epithelial cell identification

We investigated all single cells for epithelial cell populations by summarizing clusters of Epcam[+] cells. Cell clusters containing 50% or more of Epcam[+] cells were flagged as epithelial related. Next, individual cells were scored using an epithelial cell signature set of differentially expressed genes defined in wild-type mouse in ref. 48. Those cells found in flagged epithelial clusters and defined as basal, luminal 1 (L1), luminal 2 (L2) or luminal 3 (L3) were retained as confident epithelial cells and used in downstream analysis and clustering.

### RNA-seq data from human prostate cancer

The source of bulk RNA-seq data for RNA signature analysis was primary PCa samples initially reported by The Cancer Genome Atlas (TCGA) and in refs. 9,30. The sources of scRNA-seq data were primary PCa samples reported in refs. 51,52. Data were reproduced from these studies using the same cell type classification with the following exceptions. For the ref. 52 study, the two subtypes of PCa cells which were originally annotated as ERG-positive and ERG-negative tumor cells are now together named as PCa in this study because ERG[+] cells also exist in the 'ERG-negative' clusters. The ERG-positive and mixed (denoting those carrying both ERG-positive and ERG-negative tumor cells by scRNA-seq) patients defined by the original study are now grouped together as ERG[+] patients in this study considering the typically uniform ERG expression pattern in PCa by histology[2,4,7]. Similarly for the ref. 51 study, ERG[+] patients were defined as those with >50% PCa cells found to be ERG[+].

### scATAC-seq filtering features matrix

scATAC-seq data were evaluated from the Cell Ranger ARC (10x Genomics) filtered count matrix. Cell filtering, feature selection and analyses were performed using ArchR software (v.1.0.2)[114]. We generated output plots to visualize distributions of cell counts within promoters, gene bodies and transcription start sites. Quality metrics were used to remove poor quality cells, including low transcription start site enrichment (<4) or low number of unique nuclear fragment (<1,000) or putative multiplets (defined as cells containing greater than 1 million reads).

### Dimensional reduction

Dimensionality reduction was calculated via ArchR with Iterative Latent Semantic Indexing (LSI) using 500-bp genomic tiles. A nearest neighbor graph of the LSI reduced data was performed by Seurat's shared nearest neighbor implementation. We excluded the first LSI component that was correlated >75% to read depth. Clusters were then identified with the Leiden algorithm.

### Peak and motif identification

Peaks with scATAC-seq data were called using MACS3 (v.3.0.0b1)[115] through the ArchR interface. Differentially accessible regions were defined as those having significant (FDR < 0.05) difference with $\log_2$ fold change ≥ 0.5. For motif detection we utilized two databases, (1) CisBP and (2) the Non-redundant TF motif matches genome-wide[116]. Motif matching within peaks was performed using MotifMatchr (v.1.26.0) with defaults. The gene identity of each position weight matrix match was recorded so that we retained only those motifs with corresponding normalized average gene expression per cell type >1. To identify motifs statistically significant between comparisons, we calculated motif ChromVar[117] scores per cell and performed Spearman correlation of those scores to the motif's gene score values. Those motifs having positive correlation and significant ChromVar scores per cell type were retained.

### Motif co-accessibility

The ETS family of factors includes several highly similar TF motif position weight matrices. To simplify the identification of ETS family factors within scATAC-seq, we visualized the distribution of similar ETS family motif deviation *Z* scores before reducing the following motifs from oncogenic ETS family TFs in PCa[30]: ERG, ETS, ETV, FLI, into one identifier, hereafter referred to as ETS motifs. To evaluate co-accessibility of ETS motifs with other significantly accessible motifs within cell types, called candidate motifs (described above), we identified peaks containing each candidate motif and the ETS motifs. We then calculated the per cell deviation and deviation *Z* score with ChromVar. Similarly, we calculated ChromVar *Z* scores for each candidate motif whose peaks did not overlap ETS motifs and then the complement, that is, ETS motifs peaks, that do not harbor the candidate motif. For each candidate motif a two-sample Wilcoxon rank sum test was used to establish whether co-occurences of the candidate and ETS motifs were significantly different (one-sided alternative 'greater') from: (1) ETS motifs that do not co-occur with the candidate motif, or (2) candidate motifs that do not co-occur with ETS motifs. For NFKB2, NFAC1 and STAT3 the co-occurrences with ETS were significantly greater than without ETS within EPC-IM cells (Fig. 6f).

### Statistics and reproducibility

Statistical analysis was performed as specified in the figure legends. No statistical method was used to predetermine sample size. No data were excluded from the analyses. Mice were randomized for in vivo experiments. Histological assessments were performed blinded by an MSKCC pathologist. Experiments were repeated with a minimum of two independent experiments as noted in figure legends. Data distribution was assumed to be normal but this was not formally tested.

### Reporting summary

Further information on research design is available in the Nature Portfolio Reporting Summary linked to this article.

## Data availability

Raw sequencing data are publicly available from the Gene Expression Omnibus: GSE257543 (scRNA-seq), GSE258962 (scATAC-seq) and GSE294013 (bulk RNA-seq). Existing scRNA-seq data reanalyzed in our paper are available as GSE176031 and GSE181294. Source data are provided with this paper.

## Code availability

This study uses standard preprocessing and published tools for analysis, which have been cited in the Methods and are available via Zenodo at https://doi.org/10.5281/zenodo.15802144 (ref. 118).

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

## Acknowledgements

We thank the Sawyers lab for valuable critiques and discussions. We appreciate the feedback from M. M. Shen and C. Abate-Shen, and advice from the C. Abate-Shen lab for setting up the intraprostatic adenoviral injection assay. We are grateful to R. Lester for help in mouse colony management, the Molecular Cytology Core Facility from MSKCC for help with microscopy, the Single Cell Analytics Innovation Lab Core Facility for help with single-cell library preparation, the Integrated Genomics Operation Core Facility for next-generation sequencing and the Flow Cytometry Core Facility from MSKCC for help with FACS experiments. This work is supported by the following funding sources: the Howard Hughes Medical Institute (HHMI) (C.L.S.); National Institute of Health grants no. CA193837 (Y.C., C.L.S.), no. CA092629 (Y.C., C.L.S.), no. CA224079 (B.S.C., Y.C., C.L.S.), no. CA224044 (Y.C.), no. CA208100 (Y.C.), no. CA155169 (C.L.S.), no. CA276888-01 (W.F.), no. CA257811 (B.S.C.), no. CA008748 (H.Z., W.K., N.F., E.R., E.D.S., Y.C.), no. DK128602-03 (H.L.) and no. CA274492 (E.L., C.S.L.); Department of Defense grants no. W81XWH-19-1-0323 (W.F.) and no. W81XWH-21-1-0431 (B.S.C.); the Prostate Cancer Foundation (W.F., Y.C.); and the STARR Cancer Consortium (Y.C.).

## Author contributions

Matthew Lange, Nazifa Salsabeel and Huiyong Zhao are co-second authors. Conceptualization: W.F. and C.L.S. Methodology: W.F., E.L., H.Z., H.L. and C.S.L. Investigation: W.F., E.L., N.S., H.Z., Y.S.L., A.G., M.L., W.K., N.F. and E.R. Formal analysis: W.F. and E.L. Resources: Y.C. and B.S.C. Project administration: E.D.S. Funding acquisition: W.F. and C.L.S. Software: E.L. and C.S.L. Supervision: C.S.L. and C.L.S. Writing—original draft: W.F., E.L., C.S.L. and C.L.S. Writing—review and editing: W.F., E.L., C.S.L. and C.L.S.

## Competing interests

C.L.S. served on the Board of Directors of Novartis (2013–2025), is a co-founder of ORIC Pharmaceuticals and is co-inventor of enzalutamide and apalutamide. He is a science advisor to Beigene, Blueprint, CellCarta, Column Group, Foghorn, Housey Pharma, Nextech and PMV. Y.C. received research funding from Foghorn therapeutics, consulted for Fog Pharma and Belharra Therapeutics and has stock ownership of ORIC Pharmaceuticals. The other authors declare no competing interests.

## Additional information

**Extended data** is available for this paper at https://doi.org/10.1038/s41588-025-02289-w.

**Correspondence and requests for materials** should be addressed to Weiran Feng or Charles L. Sawyers.

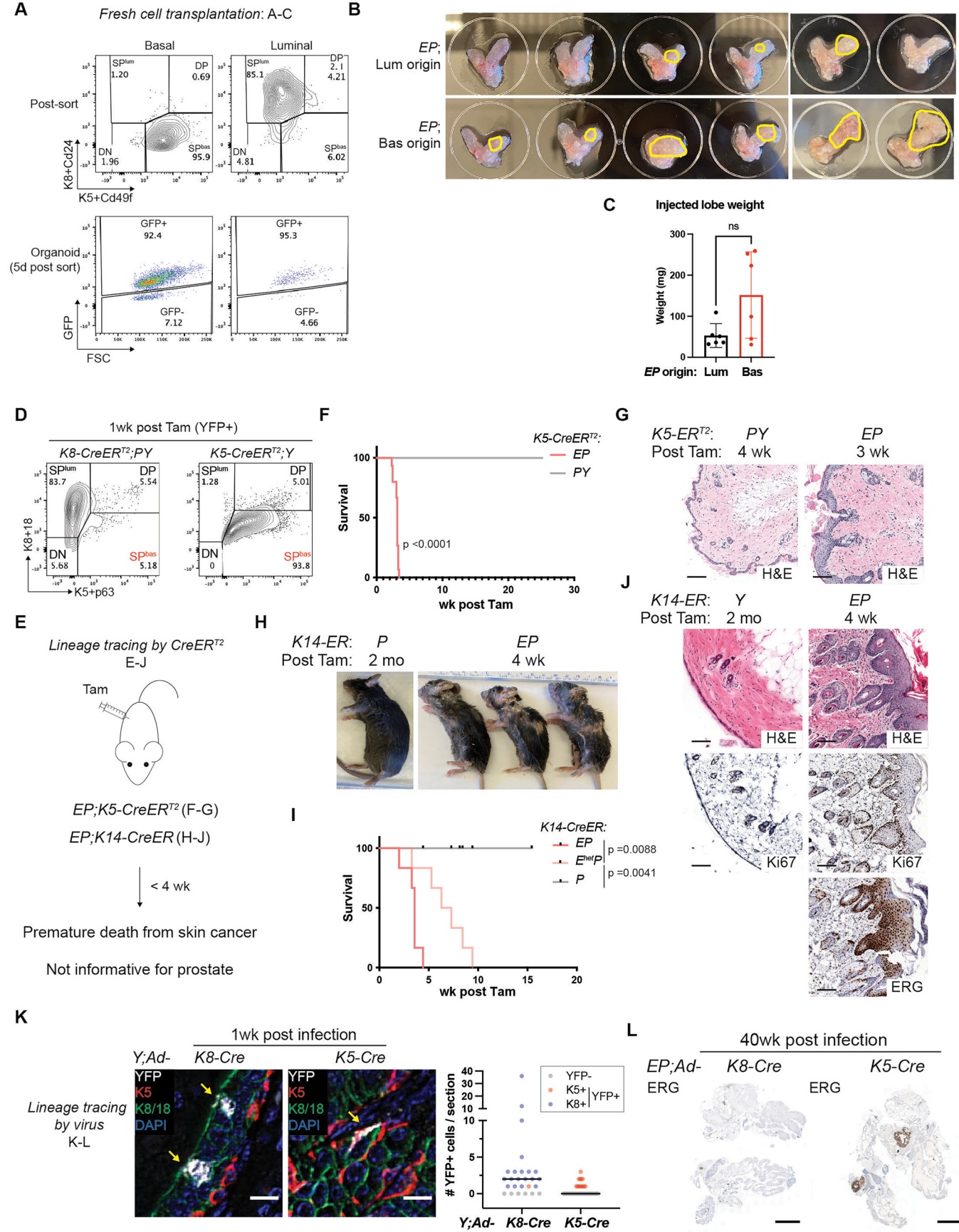

**Extended Data Fig. 1 | See next page for caption.**

**Extended Data Fig. 1 | Fresh cell transplantation and lineage tracing for gene activation from basal and luminal cells.** (**a**) Quality control for freshly isolated basal and luminal-derived cells generated in Fig. 1a. (Top) Post-sort analysis of sorted basal and luminal populations. (Bottom) Assessment of Cre recombination efficiency at 5 days post Cre. GFP was used as a surrogate for ERG expression. (**b**) Images showing prostates harvested at 5 months endpoint. Visible grafts are highlighted in yellow circles. (**c**) Weights of indicated orthografts harvested at the 5 months endpoint (n = 6 mice per group). (**d**) Flow cytometry assessing the cell type specificity of *K5-* and *K8-CreER^T2*. Whole prostates were harvested for analysis. (**e**) Schematic summarizing that lineage tracing using *K5-CreER^T2* and *K14-CreER* to activate *EP* leads to a severe skin disease and premature death. (**f-g**) Survival curve (**f**) and skin histology (**g**) from indicated mice. Scale bar, 100 μm. (**h-j**) Mice at harvest (**h**), survival curve (n = 6 mice per group) (**i**) and skin histology (**j**) of indicated mice. Scale bar, 100 μm. (**k**) IF imaging assessing the cell type specificity and infection efficiency of the indicated Ad-Cre using the *Rosa26-YFP^LSL/LSL* allele (*Y*) at the 1-week time point (n = 41 sections from 4 mice for K5-Cre group; n = 27 sections from 4 mice for K8-Cre group). Scale bars, 10 μm. (**l**) Sagittal view of whole prostates with ERG IHC. Two injection sites are shown. Scale bar, 2 mm. Data represent mean ± s.d.; ns, not significant. unpaired two-tailed t-test (**c**); Log-rank test (**f, i**).

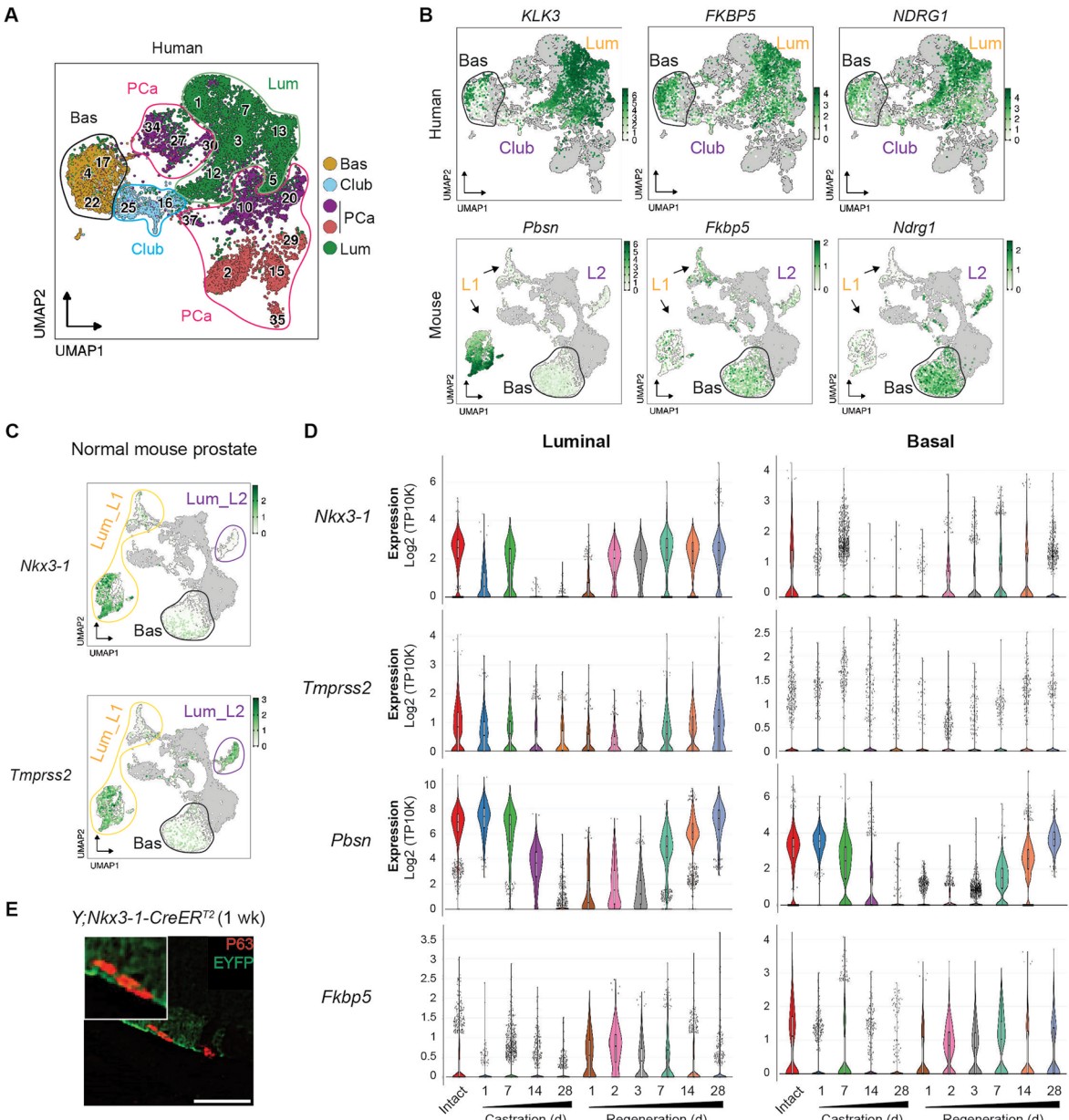

**Extended Data Fig. 2 | A subset of basal cells express androgen-regulated luminal genes.** ((**a**) UMAP of all epithelial cells from human prostates. Data were reproduced from Song et al.[52] with the same cell type classification, except that the two subtypes of the previously defined prostate cancer cells are together named as PCa here (see Methods). The basal and PCa clusters are highlighted in black and pink circles, respectively. (**b**) UMAP of epithelial cells from normal human prostates highlighting the expression of canonical luminal genes in a subset of basal cells. An expanded list of genes beyond Fig. 1g are shown. Normal basal clusters are highlighted in circles. Cells from normal samples are colored from white to green based on gene expression, with cells from tumor samples in grey in the background. The complete UMAPs and cell type annotations are shown in **A**. (**c**) UMAP of epithelial cells from normal mouse prostates highlighting the expression of canonical luminal genes (*Tmprss2, Nkx3-1*) in a subset of basal cells. Cell types from normal prostates are annotated based on Fig. 4b and highlighted in circles. Cells from normal samples are colored from white to green based on gene expression, with cells from tumor samples in grey in the background. (**d**) Changes in expression of luminal genes expressed in a subset of basal cells during a castration-regeneration cycle, indicative of androgen dependent expression, as seen in canonical luminal cells (n = 13,398 cells). Data were mined from Karthaus et al.[49]. The center line represents the median, the box limits represent the upper and lower quartiles and the minimum and maximum whiskers represent the minimum and maximum, respectively. (**e**) YFP+ cells that co-express P63 from *Y;Nkx3-1-CreERT2* mice; inset shows high-power view. Mice were treated with tamoxifen at 2 months before whole prostates were harvested 1 week post treatment. Scale bars, 25 μm.

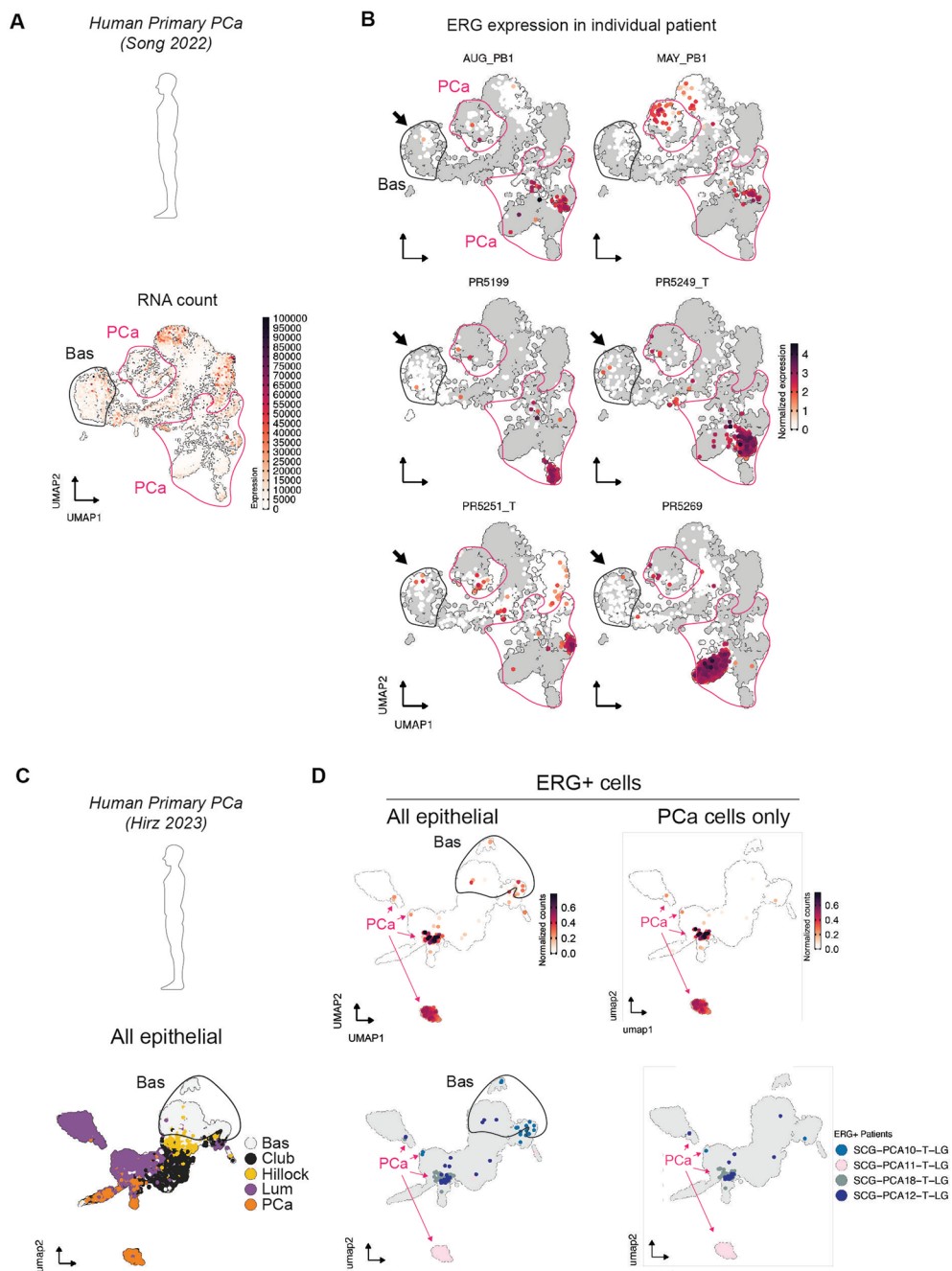

**Extended Data Fig. 3 | Evidence of ERG+ basal cells in human prostate cancer.**
(**a**) UMAP of epithelial cells from primary PCa patients (Song et al.[52]) showing RNA count to assess doublet potential. The low to intermediate level of RNA count in basal cluster suggest the ERG+ basal cells in Fig. 1k are unlikely an artificial outcome of doublet formation. (**b**) UMAP of epithelial cells showing ERG expression in individual patients. ERG+ PCa samples are shown (see Methods). Cells from each patient are colored from white to red based on gene expression, with cells from the rest of patients in grey in the background. Arrows highlight the presence of ERG+ cells in the basal cluster. Cell types in this figure are annotated based on Extended Data Fig. 2a, with basal and PCa clusters highlighted in black and pink circles, respectively. (**c**) UMAP of epithelial cells from a different primary PCa patient cohort (Hirz et al.[52]) showing all epithelial cells. (**d**) UMAP from **c** showing ERG+ cancer cells colored by ERG expression (top) and patient ID (bottom) from all epithelial cells (left) and PCa cells only (right).

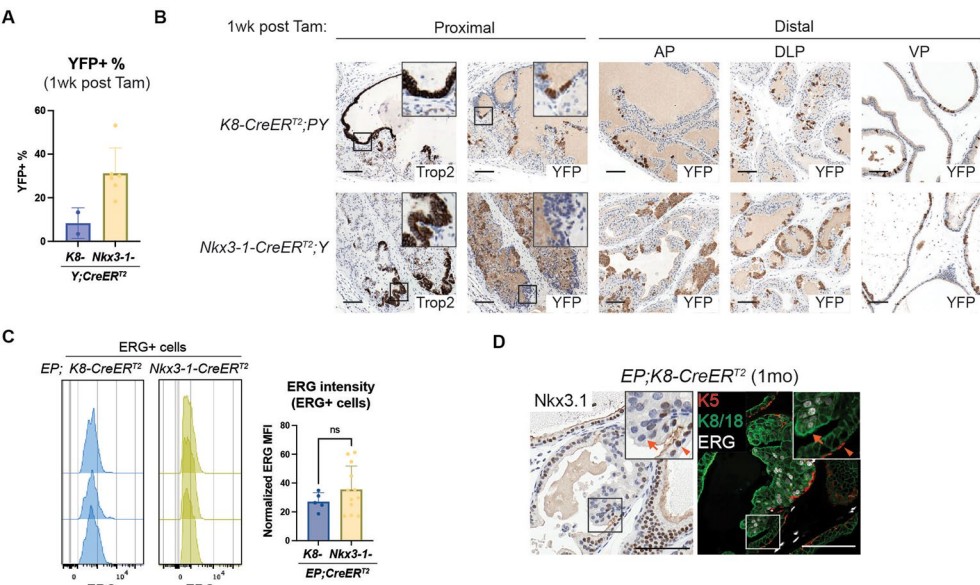

**Extended Data Fig. 4 | Comparison of Cre drivers across different genetic backgrounds.** (**a**) Cre activation as assessed by YFP reporter expression using flow cytometry (n = 2 mice for *K8* group; n = 6 mice for *Nkx3-1* group). Mice were treated with tamoxifen at 2 months before whole prostates were harvested at 1 week post treatment. (**b**) IHC documenting the presence of recombined cells resulting from *K8-CreER^T2* vs *Nkx3-1-CreER^T2* activity at different histological locations. Mice were treated with tamoxifen at 2 months before whole prostates were harvested 1 week post treatment. YFP was used as a surrogate for Cre recombination. Scale bars, 100 μm. (**c**) ERG expression activated by different Cre drivers, as quantified by flow cytometry from indicated mice (n = 5 mice for

*K8* group; n = 12 mice for *Nkx3-1* group). Mice were treated with tamoxifen at 2 months before whole prostates were harvested at 1 week post treatment. ERG+ cells are shown, with ERG intensity normalized to background signal from the corresponding ERG-negative population. MFI, mean florescence intensity. (**d**) IF and IHC highlighting Nkx3.1 loss in *EP;K8-CreER^T2* mice. Mice were harvested at 1 month post tamoxifen. High-power view in insets highlights the absence of Nkx3.1 signal in ERG+ cells (arrow) and the neighboring Nkx3.1+ normal luminal cells. Scale bars, 100 μm. Data represent mean ± s.d.; ns, not significant; unpaired two-tailed t-test.

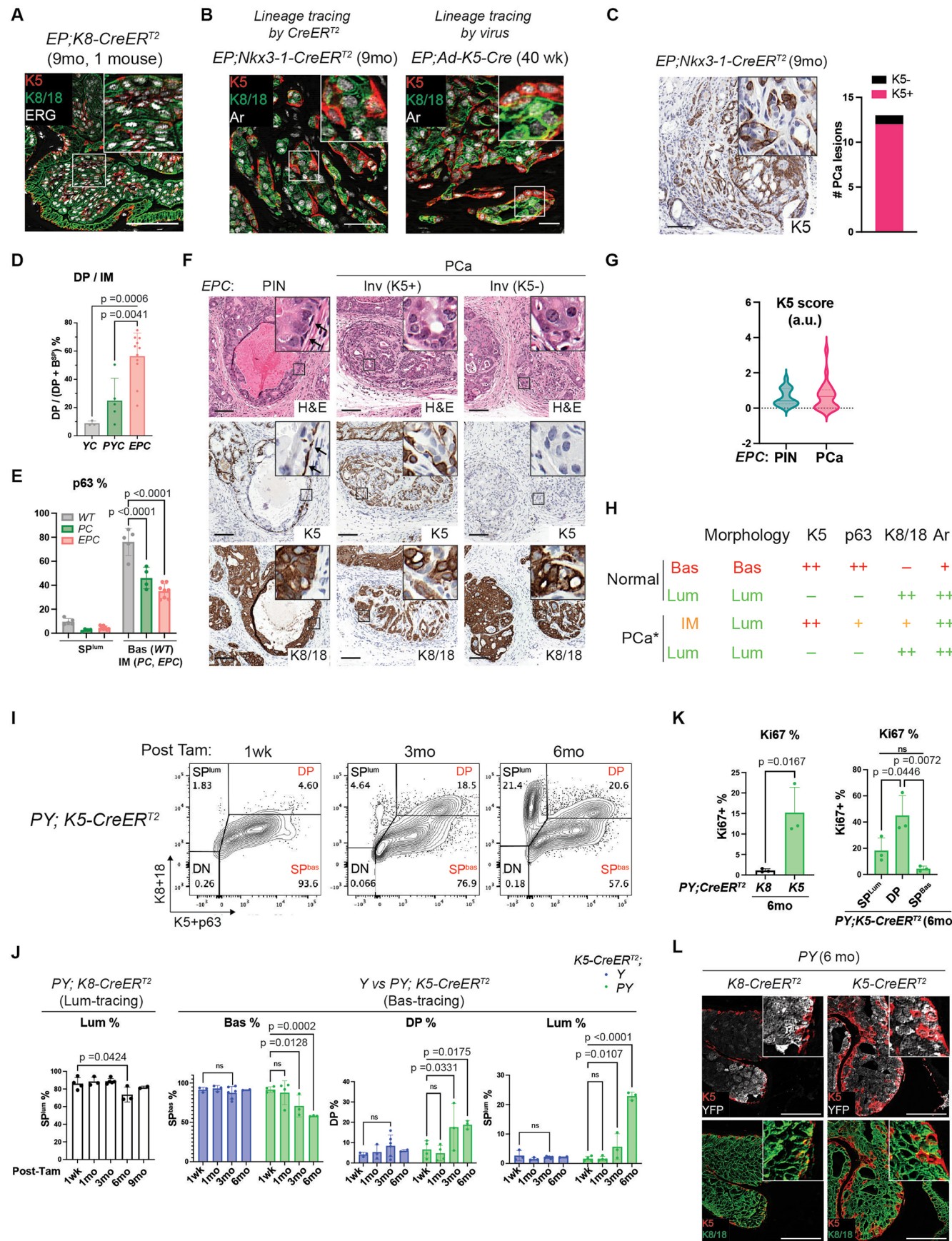

**Extended Data Fig. 5 | See next page for caption.**

**Extended Data Fig. 5 | Origin, presence and characterization of intermediate state in prostate cancer.** (**a**) IF image showing rare foci of K5 + IM cells from one mouse at the 9 months endpoint; inset shows high-power view. Scale bar, 100 μm. (**b**) IF showing Ar expression in invasive adenocarcinomas of indicated mice; inset shows high-power view. Scale bars, 50 μm (left), 20 μm (right). (**c**) immunohistochemistry IHC (left) and quantification (right) of K5 + IM cells in PCas (invasive adenocarcinomas) from indicated mice; insets show high-power view. Scale bars, 100 μm. (**d**) Flow cytometry quantification of DP/IM across of recombined cells (YFP+ for *YC/PYC*, ERG+ for *EPC*) from indicated mice (left to right: n = 3, 5, 11 mice). Whole prostate cells were harvested at 3-month age. (**e**) Flow cytometry quantifying p63 expression in indicated population (*WT*, n = 5 mice; *PYC*, n = 4 mice; *EPC*, n = 8 mice). Recombined cells labeled by YFP or ERG were analyzed for *YC/PYC* and *EPC* mice, respectively. (**f**) Prostate histology of 3-month *EPC* mice highlighting a luminal morphology of both K5+ and K5- cells from invasive adenocarcinomas, in contrast to the K5+ basal cells that

encapsulate the precursor PIN lesions (arrow); inset shows high-power view. PIN, prostatic intraepithelial neoplasia; inv, invasive adenocarcinoma; PCa, prostate cancer. Scale bars, 100 μm. (**g**) Quantification of K5 IHC signal in PIN and invasive adenocarcinomas of *EPC* prostates at 2 - 3-month age. (**h**) Summary of cell morphology and protein marker expression across different cell types based on flow cytometry and immunostaining. Asterisk, basal cells are absent in PCa, which displays IM cells instead. (**i**) Representative flow plot showing lineage identities of recombined cells (YFP + ) from indicated mice. Whole prostates were harvested at indicated time point. (**j**) Flow cytometry quantifying indicated cell states = from indicated mice. Left to right: n = 4,3,5,3,2 mice (*K8-CreERT2;PY*); n = 3,3,6,2 mice (*K5-CreERT2;Y*); n = 4,4,3,3 mice (*K5-CreERT2;PY*) (**k**) Flow cytometry quantifying Ki67 status (n = 3 mice per group). (**l**) Prostate IF of indicated mice. Scale bars, 100 μm. Data represent mean ± s.d.; one-way ANOVA with Tukey posttest (**d**, **e**, **k right**); one-way ANOVA with Dunnett posttest (**j**); unpaired two-tailed t-test (**k**, left).

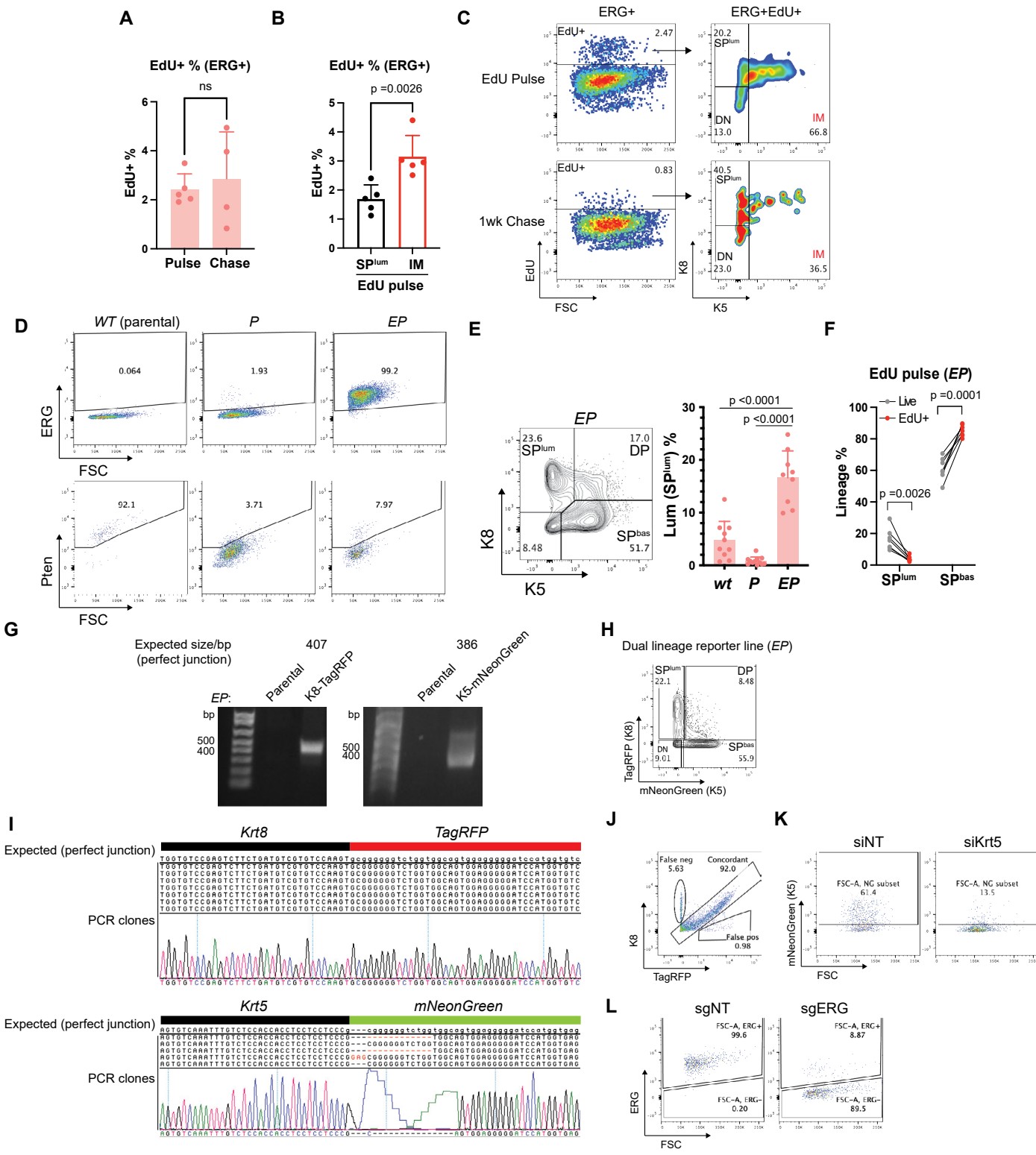

**Extended Data Fig. 6 | ERG+ basal and intermediate cells proliferate towards a luminal fate *in vitro* and *in vivo*.** (**a**) *In vivo* EdU pulse chase assay in ERG + *EPC* cells (n = 5 mice for pulse, 4 mice for chase). (**b**) EdU quantification in SP^lum and IM populations of ERG + *EPC* cells in pulse samples from **A** (n = 5 mice). (**c**) Flow cytometry analysis on ERG+ cells from indicated mice. (**d**) Flow cytometry validating the ERG and Pten expression status in *EP* organoids and the isogenic controls. (**e**) Flow cytometry quantifying luminal cells (SP^lum) in organoids (left to right: n = 10, 12, 9 independent assays). (**f**) Flow cytometry quantification in *EP* cells after a EdU pulse (n = 7 independent assays). (**g**) Junction PCR across the target loci and the engineered reporters in the bulk organoid population.

(**h**) Live cell flow cytometry using the engineered reporter signals. (**i**) Sanger sequencing of the junction PCR clones from **G**. (**j**) Intracellular flow cytometry comparing the expression pattern between the endogenous K8 and the TagRFP reporter signal in targeted organoid population. (**k**) Functional validation of the *Krt5-mNeonGreen* engineering by Krt5 depletion. (**l**) Intracellular flow cytometry quantifying ERG in *EP* organoids with indicated CRISPR engineering. Cells were harvested 2 days after introducing the indicated CRISPR-RNP. Data represent mean ± s.d.; ns, not significant; unpaired two-tailed t-test (**a, b**); unpaired two-tailed t-test (**e**); multiple paired t-test (two-sided) with FDR correction by Benjamini, Krieger and Yekutieli (**f**).

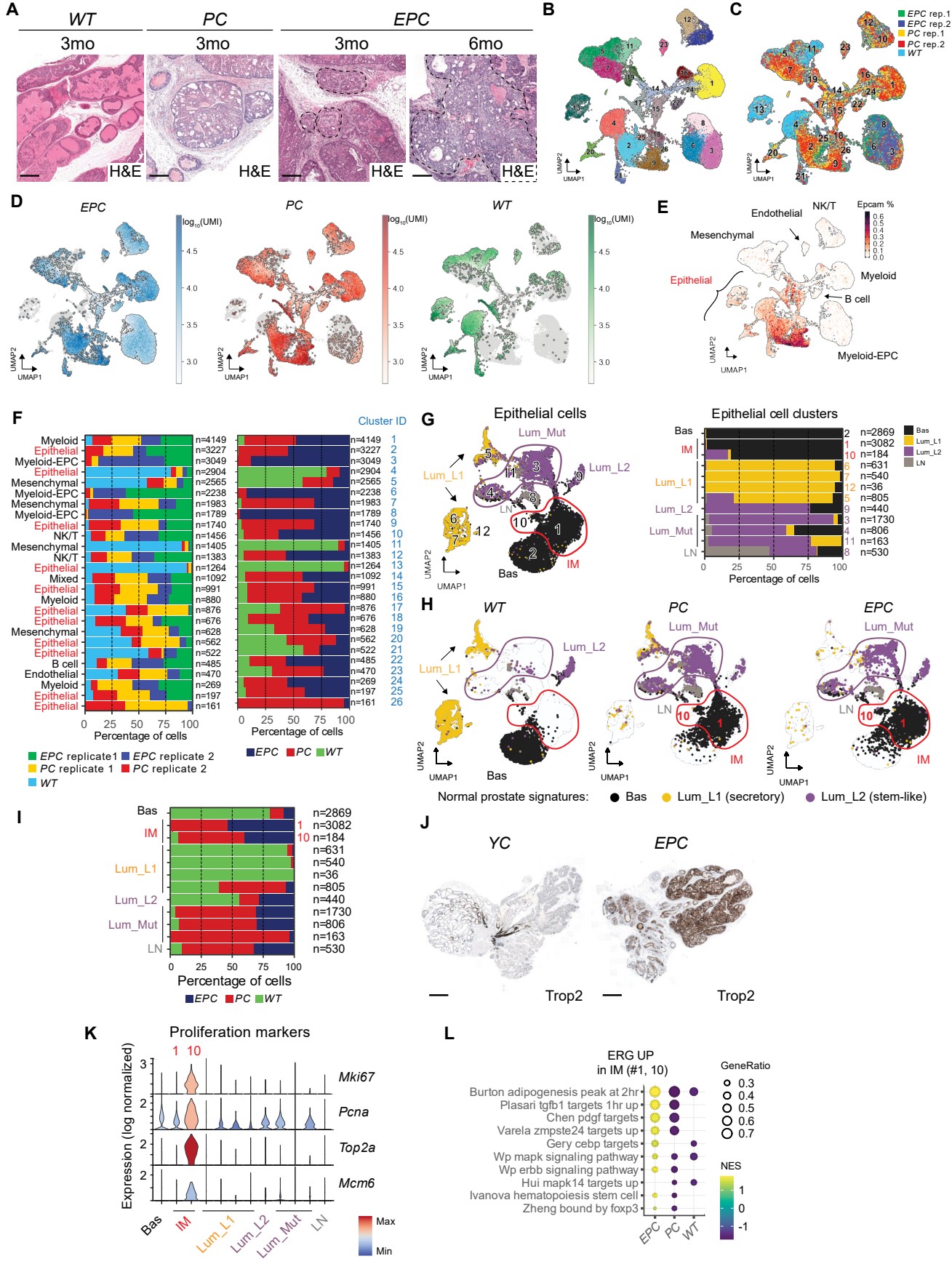

**Extended Data Fig. 7 | See next page for caption.**

**Extended Data Fig. 7 | Time point characterization and annotation of scRNA-seq clusters of genetically engineered mouse models (GEMMs).** (**a**) H&E prostate histology. Dashed lines encircle invasive adenocarcinoma (n = 4 mice per group). Scale bars, 250 μm. (**b**) UMAP of all cells (N = 36,961 cells) from the prostates of 5 mice (2 *EPC*, 2 *PC*, 1 *WT*) annotated based on leiden clustering. (**c**) UMAP of all cells colored by individual replicates per genotype. (**d**) UMAP of all cells showing cells assigned to each genotype and colored by log10(UMI). (**e**) UMAP of all cells previously defined normal mouse prostate signatures. Epithelial cell clusters defined based on *Epcam* expression are highlighted. Myeloid cells specific to the *EPC* samples were defined as Myeloid-EPC. (**f**) Barplot of all cells colored by individual replicates (left) or each genotype (right). Numbers of cells in each cluster were included (n = ) and cluster number corresponding to **B** is shown. (**g**) UMAP (left) and barplot (right) of epithelial cells colored by cell types based on normal prostate signatures from Karthaus et al.[49]. L2 cells (purple circles) and basal-like cells (red circles) specific to *PC* and *EPC* prostates are highlighted (termed Lum_Mut and IM respectively). (**h**) UMAP showing epithelial clusters assigned to each genotype. The IDs of the two IM clusters are shown. (**i**) Barplot of epithelial clusters showing the cell type composition of each cluster. Numbers of cells in each cluster were included. (**j**) IHC of Trop2 (L2 marker) on indicated mice at 3 months age. Scale bars, 1 mm. (**k**) Violin plots comparing proliferative marker expression across all epithelial clusters. The two Tumor-Bas (IM) clusters are numbered according to the UMAP in **G**. (**l**) Gene set enrichment analysis showing EPC-specific pathways (ERG UP) within the Tumor-Bas/IM population.

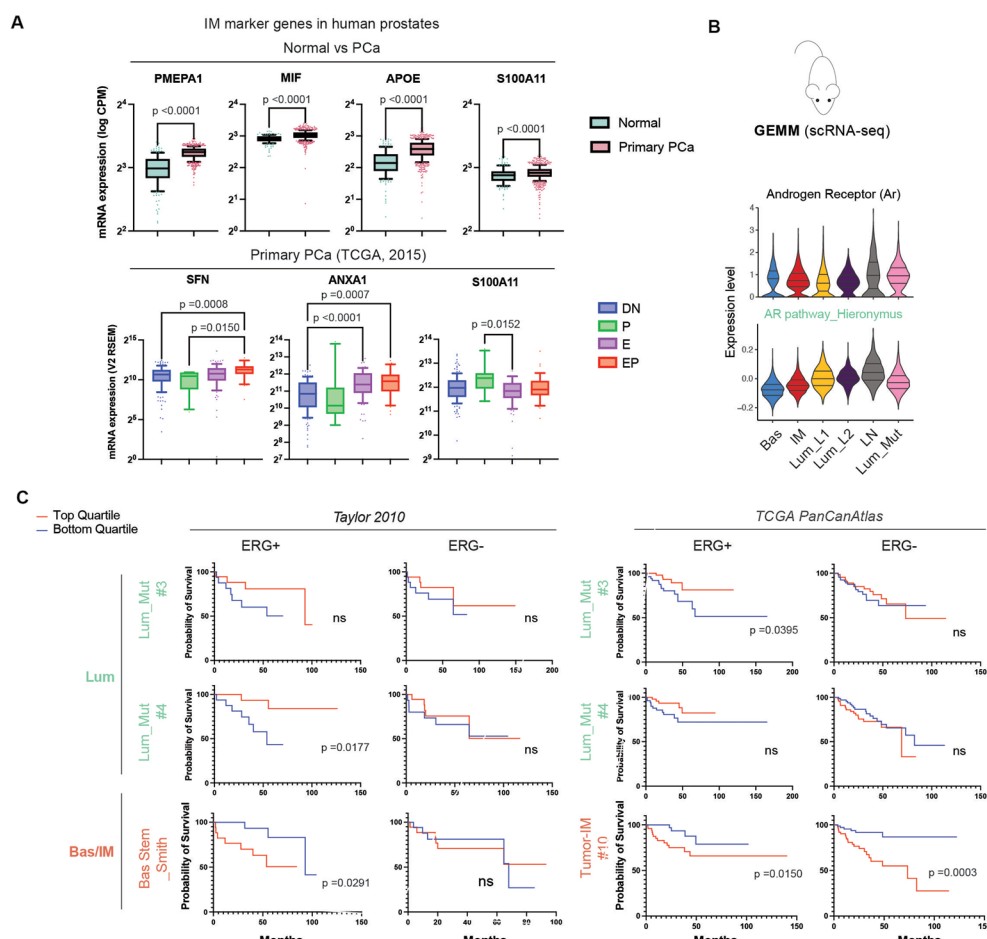

**Extended Data Fig. 8 | Intermediate signatures are associated with a worse clinical outcome in prostate cancer.** (**a**) IM marker gene expression in publicly available human normal and prostate cancer datasets (TCGA 2015; Bolis et al 2021). (Top) n = 173 (normal), 708 (primary PCa). (Bottom) E, ERG-fusion-positive (n = 89); P, PTEN deep deletion (n = 10); EP, ERG-fusion-positive and PTEN deep deletion (n = 40); DN, double negative (ERG fusion-negative, PTEN deep deletion-negative) (n = 151). The center line represents the median, the box limits represent the upper and lower quartiles and the minimum and maximum whiskers represent the 10th and 90th percentiles, respectively. (**b**) Expression of *Ar* and Ar pathway genes across scRNA-seq clusters in mice. (**c**) Progression-free survival outcome using indicated signatures from two independent cohorts of patients with primary PCa, stratified based on ERG status (see also in Fig. 5c). Data represent mean ± s.d.; ns, not significant; unpaired two-tailed t-test (**a**, top); one-way ANOVA with Tukey posttest (**a**, bottom); Log-rank (Mantel-Cox) test (**c**).

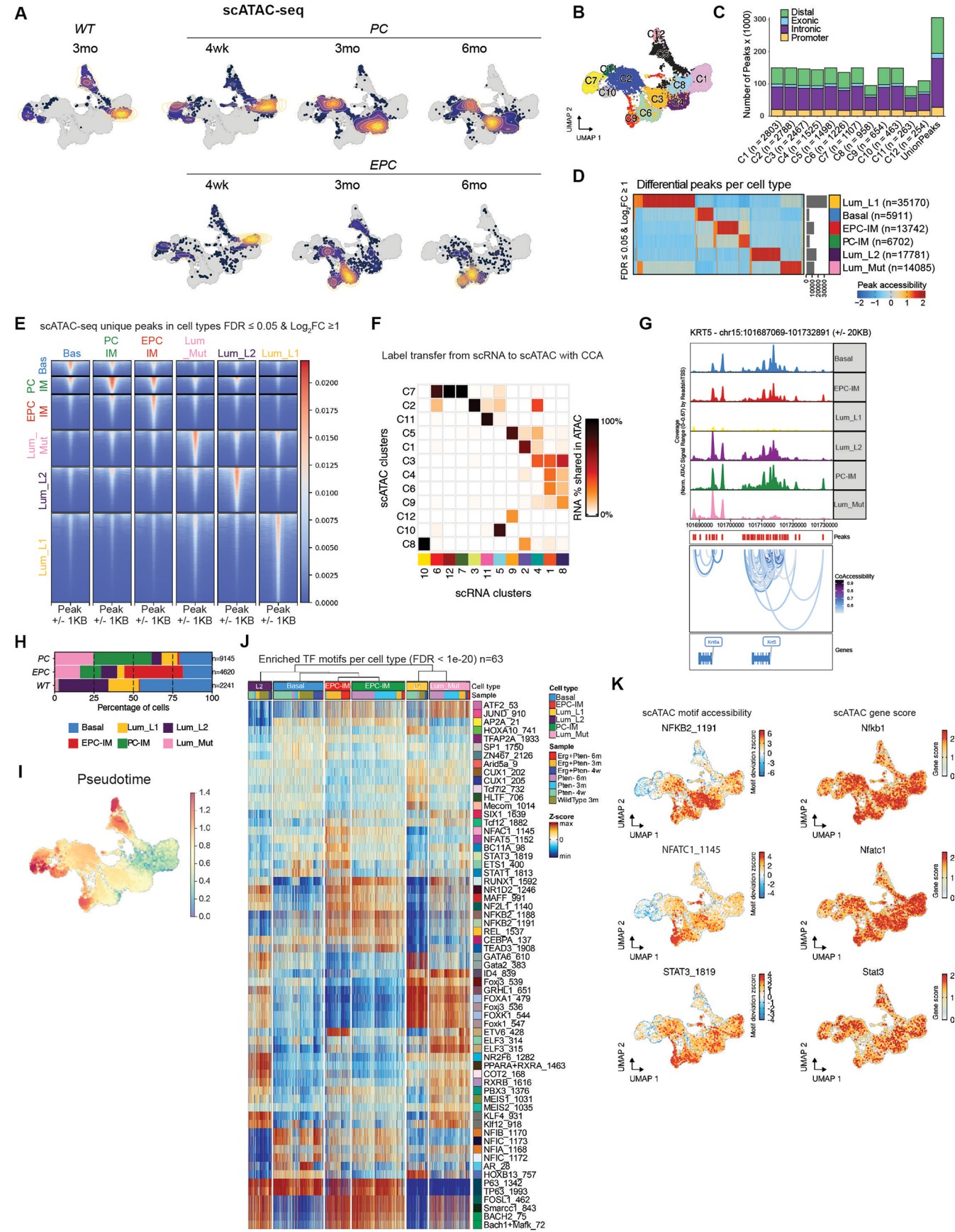

**Extended Data Fig. 9 | See next page for caption.**

**Extended Data Fig. 9 | ERG drives a unique chromatin state in intermediate cells. (a)** scATAC-seq UMAPS by indicated genotypes and time points. Cell density is overlaid and colored by increasing density, from dark to light. **(b)** Clustering generated by leiden algorithm is shown labeled and also colored on scATAC-seq UMAP. **(c)** Barplot showing number of peaks annotated as distal, exonic, intronic, or promoter by cluster. (**d**) Heatmap showing differential peaks between cell types. **(e)** Tornado plot showing significantly enriched peaks (FDR < = 0.05 and Log2FC >= 1) for each annotated cell type (Bas, PC-IM, EPC-IM, Lum_Mut, Lum_L2, Lum_L1) vs all other cell types. Signal plotted is normalized scATAC-seq aggregated by cell type, centered at peak and extended by 1000 nt up and down. **(f)** Grid showing amount of RNA shared with scATAC-seq by cluster. Labels from scRNA were transferred to scATAC using Seurat's CCA. **(g)** scATAC-seq reads piled up per cell type around the mm10 *KRT5* locus, a classical basal cell marker (top panel). As expected, signal in Lum_L1 is low compared to Basal. Second

panel from top shows tick marks to indicate detected peaks. Third panel from top show inferred Co-Accessibility using Cicero, an algorithm used to predict DNA interactions from scATAC-seq data. Fourth panel from top shows *Krt5* gene locus and neighboring *Krt6a* locus in mouse. **(h)** Barplot percentage of cell type per genotype is shown. (**i**) UMAP of cells colored by pseudotime indicating later trajectory branches map to Lum_L1, Lum_L2, and EPC-IM. **(j)** scATAC-seq heatmap of most enriched transcription factor candidates when comparing one cell type to all others. Cells are grouped by cell type as columns and rows indicate motifs from database. The heatmap elements are colored by chromvar inferred z-score. Motif significance was calculated (see Methods) and those having an p-value < 1e-20 after BH-false discovery rate correction are shown. **(k)** Representative UMAPs colored by motif deviation z-score (left) and inferred gene expression (gene score, right) of EPC-IM enriched TFs (NFKB2, NFATC1, STAT3).

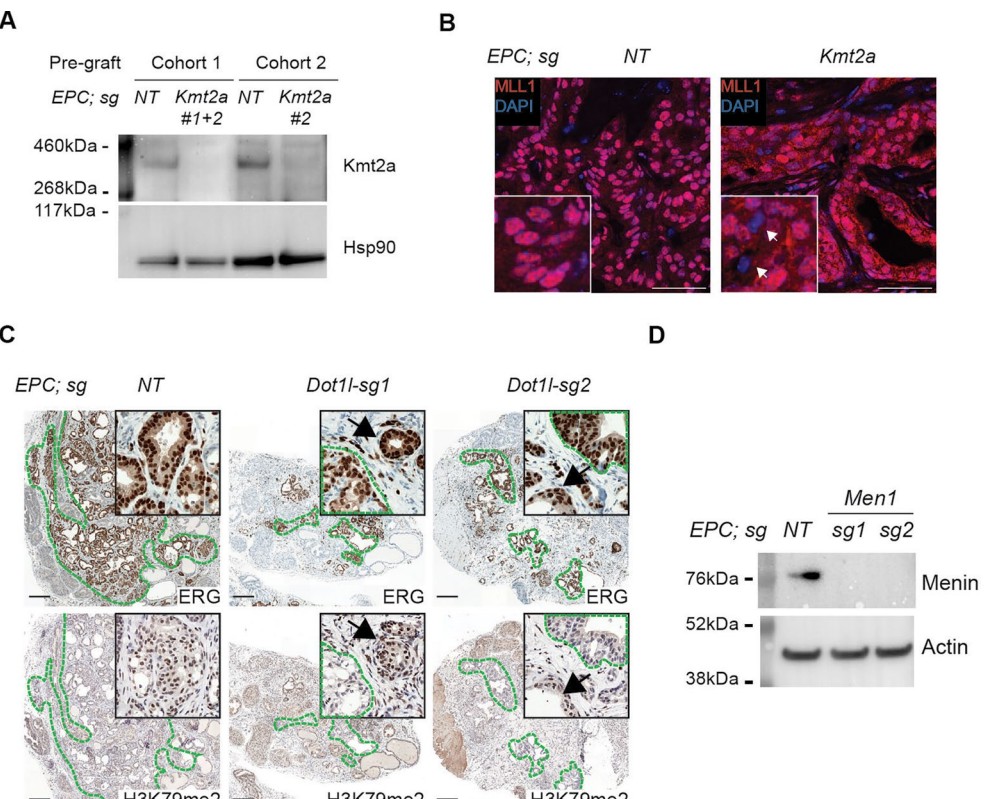

**Extended Data Fig. 10 | Characterization of Kmt2a, Dot1l, Men1 perturbation in pre- and post-grafts. (a)** Western blot confirming the knockout status of Kmt2a/Mll1 in *EPC;sgKmt2a* pre-graft organoids (n = 2 independent cohorts). (**b**) IF showing that the residual cells from *EPC;sgKmt2a* orthografts largely maintained Mll1 expression, suggestive of knockout escapers (n = 6 grafts per group). High-power view from insets highlight the rare existence of Mll1-null cells. Scale bars, 50 μm. (**c**) ERG and H3K79me2 IHC (n = 6 grafts per group). Tumor areas were quantified based on ERG+H3K79me2− signals as highlighted in green dashed lines. Arrows in the sgDot1l groups highlight ERG+ cells that retain H3K79me2 signals. Inset shows high-power view. Scale bars, 250 μm. (**d**) Western blot confirming the knockout status of Menin in *EPC;sgMen1* pre-graft organoids (n = 2 independent sgRNAs).

# Reporting Summary

## Statistics

For all statistical analyses, confirm that the following items are present in the figure legend, table legend, main text, or Methods section.

| n/a | Confirmed | |
|---|---|---|
| ☐ | ☒ | The exact sample size ($n$) for each experimental group/condition, given as a discrete number and unit of measurement |
| ☐ | ☒ | A statement on whether measurements were taken from distinct samples or whether the same sample was measured repeatedly |
| ☐ | ☒ | The statistical test(s) used AND whether they are one- or two-sided *Only common tests should be described solely by name; describe more complex techniques in the Methods section.* |
| ☒ | ☐ | A description of all covariates tested |
| ☐ | ☒ | A description of any assumptions or corrections, such as tests of normality and adjustment for multiple comparisons |
| ☐ | ☒ | A full description of the statistical parameters including central tendency (e.g. means) or other basic estimates (e.g. regression coefficient) AND variation (e.g. standard deviation) or associated estimates of uncertainty (e.g. confidence intervals) |
| ☐ | ☒ | For null hypothesis testing, the test statistic (e.g. $F$, $t$, $r$) with confidence intervals, effect sizes, degrees of freedom and $P$ value noted *Give P values as exact values whenever suitable.* |
| ☒ | ☐ | For Bayesian analysis, information on the choice of priors and Markov chain Monte Carlo settings |
| ☒ | ☐ | For hierarchical and complex designs, identification of the appropriate level for tests and full reporting of outcomes |
| ☒ | ☐ | Estimates of effect sizes (e.g. Cohen's $d$, Pearson's $r$), indicating how they were calculated |

*Our web collection on statistics for biologists contains articles on many of the points above.*

## Software and code

Policy information about availability of computer code

| Data collection | no software was used |
|---|---|
| Data analysis | ArchR : Version 1.0.2<br>ComplexHeatmap: Version 2.20.0<br>Data.table: Version 1.17.0<br>fgsea: Version 1.30.0<br>ggplot2: Version 3.5.1<br>Macs3: Version 3.0.0b1<br>Motifmatchr: Version 1.26.0<br>Presto: Version 1.0.0<br>R: Version 4.4.0<br>Seurat: Version 5.2.1<br>Scanpy: Version 1.10.1<br>FlowJo: 10.9.0 |

For manuscripts utilizing custom algorithms or software that are central to the research but not yet described in published literature, software must be made available to editors and reviewers. We strongly encourage code deposition in a community repository (e.g. GitHub). See the Nature Portfolio guidelines for submitting code & software for further information.

## Data

Policy information about availability of data

All manuscripts must include a data availability statement. This statement should provide the following information, where applicable:
- Accession codes, unique identifiers, or web links for publicly available datasets
- A description of any restrictions on data availability
- For clinical datasets or third party data, please ensure that the statement adheres to our policy

Raw sequencing data are publicly available from the Gene Expression Omnibus: GSE257543 (scRNA-seq), GSE258962 (scATAC-seq), GSE294013 (bulk RNA-seq). Existing scRNA-seq data reanalyzed in our manuscript are available as: GSE176031, GSE181294. All other data are available in the main text or the supplementary materials. Source data are provided with this paper.

## Research involving human participants, their data, or biological material

Policy information about studies with human participants or human data. See also policy information about sex, gender (identity/presentation), and sexual orientation and race, ethnicity and racism.

| | |
|---|---|
| Reporting on sex and gender | n/a |
| Reporting on race, ethnicity, or other socially relevant groupings | n/a |
| Population characteristics | n/a |
| Recruitment | n/a |
| Ethics oversight | n/a |

Note that full information on the approval of the study protocol must also be provided in the manuscript.

# Field-specific reporting

Please select the one below that is the best fit for your research. If you are not sure, read the appropriate sections before making your selection.

☒ Life sciences          ☐ Behavioural & social sciences          ☐ Ecological, evolutionary & environmental sciences

For a reference copy of the document with all sections, see nature.com/documents/nr-reporting-summary-flat.pdf

# Life sciences study design

All studies must disclose on these points even when the disclosure is negative.

| | |
|---|---|
| Sample size | No statistical method was used to predetermine sample size. |
| Data exclusions | No data were excluded |
| Replication | Experiments were repeated with a minimum of two independent experiments and noted in figure legends. |
| Randomization | Age- and litter- matched male mice were either freshly purchased from repository or randomized into different experimental groups used for all animal studies. |
| Blinding | Histological grading for tumor samples were performed by a blinded prostate cancer pathologist. Other analysis (single cell sequencing, flow cytometry, imaging analysis) were automately performed. |

# Reporting for specific materials, systems and methods

We require information from authors about some types of materials, experimental systems and methods used in many studies. Here, indicate whether each material, system or method listed is relevant to your study. If you are not sure if a list item applies to your research, read the appropriate section before selecting a response.

## Materials & experimental systems

| n/a | Involved in the study |
|-----|-----------------------|
| ☐ | ☒ Antibodies |
| ☐ | ☒ Eukaryotic cell lines |
| ☒ | ☐ Palaeontology and archaeology |
| ☐ | ☒ Animals and other organisms |
| ☐ | ☒ Clinical data |
| ☒ | ☐ Dual use research of concern |
| ☒ | ☐ Plants |

## Methods

| n/a | Involved in the study |
|-----|-----------------------|
| ☒ | ☐ ChIP-seq |
| ☐ | ☒ Flow cytometry |
| ☒ | ☐ MRI-based neuroimaging |

# Antibodies

**Antibodies used**

Antibodies for tissue IHC and IF:
GFP Chicken 2ug/ml ER2 Abcam ab13970 IHC
K5 Rabbit 1ug/ml ER2 Abcam ab53121 IHC
K8/18 Rabbit 0.12ug/ml ER2 Abcam ab53280 IHC
P63 mouse prediluted ER2 Ventana 790-4509 IHC
ERG rabbit 1ug/ml ER2 Epitomics 2805-1 IHC
pAkt Rabbit 1ug/ml ER2 Cell signaling technology 4060 IHC
Trop2 Rabbit 0.5ug/ml ER2 Abcam ab214488 IHC
Nkx3-1 Rabbit 0.44ug/ml ER2 Proteintech 13069-1-AP IHC
GFP Chicken 2ug/ml ER2 Abcam ab13970 IF
K5 Rabbit 0.25ug/ml ER2 abcam ab53121 IF
K8/18 Rabbit 0.03ug/ml ER2 Abcam ab53280 IF
Ar Rabbit 0.5ug/ml ER2 Abcam ab108341 IF
ERG Rabbit 0.5ug/ml ER2 Epitomics 2805-1 IF
pAkt Rabbit 0.5ug/ml ER2 Cell signaling technology 4060 IF

Antibodies for flow cytometry:
K8-AF405 Rabbit 1:500 Abcam ab210139 Primary
K8-AF647 Rabbit 1:500 Abcam ab192468 Primary
K18-Biotin Rabbit 1:500 Abcam ab27553 Primary
K5-APC Rabbit 1:500 Abcam ab224984 Primary
K5-PE Rabbit 1:500 Abcam ab224985 Primary
p63-AF647 Rabbit 1:500 Abcam ab246728 Primary
ERG Rabbit 1:1000 Abcam ab92513 Primary
ERG-AF488 Rabbit 1:500 Abcam ab196374 Primary
ERG-AF647 Rabbit 1:500 Abcam ab196149 Primary
Pten Rabbit 1:500 Abcam ab170941 Primary
Ki67-PE Rat 1:50 BioLegend 652404 Primary
CD45-BV605 Rat 1:600 BioLegend 103139 Primary
CD31-BV605 Rat 1:600 BioLegend 102427 Primary
TER-119-BV605 Rat 1:600 BioLegend 116239 Primary
EpCAM-PE/Cy7 Rat 1:1000 BioLegend 118216 Primary
CD49f-PE Rat 1:200 BD 555736 Primary
CD24-AF647 Rat 1:200 BioLegend 101818 Primary
Rabbit IgG Goat 1:1000 Thermo Fisher A-21245 Secondary
Biotin Streptavidin-AF405 1:1000 Thermo Fisher S32351 Secondary

Antibodies for Western Blot:
Kmt2a (1:1,000; Cell Signaling Technology 14689), Menin (1:1,000; Cell Signaling Technology 6891S), Hsp90 (1:1,000; Cell Signaling Technology 4877S), Actin-HRP (horseradish peroxidase) (1:10,000; Abcam ab49900)

**Validation**

Antibodies were all validated according to manufacturer's website:
Abcam (https://www.abcam.com); Cell Signaling Technology (https://www.cellsignal.com); Thermo Fisher Scientific (https://www.thermofisher.com); BioLegend (https://www.biolegend.com/); Ventana (https://diagnostics.roche.com/global/en/products/product-category/product-finder.html?limit=18&tags=%5B%7B%22tags_es%22%3A%22Product%20Families%3AVENTANA%22%7D%5D&facets=%22familyl1%3AVENTANA%22&listing=Products&);
Validation was performed for human (ERG) and/or mouse (other protein targets) species using applications corresponding to those used in this study: Flow cytometry, IHC/IF, Western Blot.

# Eukaryotic cell lines

Policy information about cell lines and Sex and Gender in Research

**Cell line source(s)**

Mouse prostate organoids were derived from mice with the appropriate genotypes.

| Authentication | No authentication was performed since the organoid lines were freshly derived from mice with the appropriate genotypes that have been confirmed. |
| Mycoplasma contamination | All cells were tested negative for mycoplasma contamination. |
| Commonly misidentified lines (See ICLAC register) | N/A |

## Animals and other research organisms

Policy information about studies involving animals; ARRIVE guidelines recommended for reporting animal research, and Sex and Gender in Research

| Laboratory animals | Mice were in NSG, or mixed strain backgrounds. Males and females were used for breeding. Males were used for study since the focus is on prostate. Mice were maintained under 12h light/dark cycle (switching at 6am/pm), with controlled temperature and humidity, and with access to regular chow and sterilized water. |
| Wild animals | The study did not involve wild animals |
| Reporting on sex | The study only applies to males since the focus is on prostate. |
| Field-collected samples | The study did not involve field-collected samples. |
| Ethics oversight | Mouse experiments were conducted under protocol 06-07-012 approved by the Institutional Animal Care and Use Committee of Memorial Sloan Kettering Cancer Center (MSKCC), New York. |

Note that full information on the approval of the study protocol must also be provided in the manuscript.

## Clinical data

Policy information about clinical studies
All manuscripts should comply with the ICMJE guidelines for publication of clinical research and a completed CONSORT checklist must be included with all submissions.

| Clinical trial registration | N/A |
| Study protocol | N/A |
| Data collection | All clinical data from the study were obtained from publicly available dataset (listed in Method section). |
| Outcomes | N/A |

## Plants

| Seed stocks | N/A |
| Novel plant genotypes | N/A |
| Authentication | N/A |

## Flow Cytometry

### Plots

Confirm that:

☒ The axis labels state the marker and fluorochrome used (e.g. CD4-FITC).

☒ The axis scales are clearly visible. Include numbers along axes only for bottom left plot of group (a 'group' is an analysis of identical markers).

☒ All plots are contour plots with outliers or pseudocolor plots.

☒ A numerical value for number of cells or percentage (with statistics) is provided.

## Methodology

Sample preparation
Prostate organoid cells or freshly dissociated prostate tissue cells were stained with surface marker antibodies prior to sorting, for fixed and permeabilized for intracellular staining prior to analysis, as detailed in Methods section.

Instrument
BD Fortessa or MACSQuant 16 for analysis, BD Aria or Symphony for sorting.

Software
FlowJo

Cell population abundance
Sorted population was > 85% pure as determined by post-sort analysis

Gating strategy
Cells were gated on FSC/SSC for debris elimination, singlets (SSC-H vs -W), live cells (Live/Dead NIR exclusion) prior to downstream analysis. Gating were based on either biological (ie non-expressing cells) or technical (ie FMO) negative controls.

☒ Tick this box to confirm that a figure exemplifying the gating strategy is provided in the Supplementary Information.

