## [Peer Review File · Nature Genetics]

ERG-driven prostate cancer initiation is cell context dependent and requires KMT2A and DOT1L

Corresponding Author: Professor Charles Sawyers

Version 0:

Decision Letter:

24th Aug 2024

Dear Professor Sawyers,

sorry for the delay in providing this decision.

Your Article, "ERG-driven prostate cancer initiation is cell context and KMT2A/MLL1-dependent" has now been seen by 2 referees. You will see from their comments copied below that while they find your work of considerable potential interest, they have raised quite substantial concerns that must be addressed. In light of these comments, we cannot accept the manuscript for publication, but would be very interested in considering a revised version that addresses these serious concerns.

We hope you will find the referees' comments useful as you decide how to proceed. If you wish to submit a substantially revised manuscript, please bear in mind that we will be reluctant to approach the referees again in the absence of major revisions.

To guide the scope of the revisions, the editors discuss the referee reports in detail within the team, including with the chief editor, with a view to identifying key priorities that should be addressed in revision and sometimes overruling referee requests that are deemed beyond the scope of the current study. In this case, we ask you to address the Reviewers' comments in full, particularly focusing on the addition of human prostate cancer work, which we agree would substantiate the clinical implications of your findings. We hope that you will find the prioritised set of referee points to be useful when revising your study. Please do not hesitate to get in touch if you would like to discuss these issues further.

If you choose to revise your manuscript taking into account all reviewer and editor comments, please highlight all changes in the manuscript text file. At this stage we will need you to upload a copy of the manuscript in MS Word .docx or similar editable format.

*2) If you have not done so already please begin to revise your manuscript so that it conforms to our Article format instructions, available here. Refer also to any guidelines provided in this letter.

*3) Include a revised version of any required Reporting Summary: <https://www.nature.com/documents/nr-reporting-summary.pdf>

Please be aware of our [guidelines](https://www.nature.com/nature-research/editorial-policies/image-integrity) on digital image standards.

Link Redacted

If you wish to submit a suitably revised manuscript we would hope to receive it within 6 months. If you cannot send it within this time, please let us know. We will be happy to consider your revision so long as nothing similar has been accepted for publication at Nature Genetics or published elsewhere. Should your manuscript be substantially delayed without notifying us in advance and your article is eventually published, the received date would be that of the revised, not the original, version.

Thank you for the opportunity to review your work.

Sincerely,
Chiara

Chiara Anania, PhD
Associate Editor
Nature Genetics
<https://orcid.org/0000-0003-1549-4157>

Referee expertise:

Referee #1: prostate cancer, cancer biology

Referee #2: prostate cancer, cancer epigenetics

Referee #3:

Reviewers' Comments:

Reviewer #1:

Remarks to the Author:

The study by Feng et al investigates the relationship between Erg-driven cancer initiation and cell of origin of prostate cancer. Using a series of complementary approaches based on mouse models, the authors conclude that Erg initiated tumors arise from a specific sub-population of prostatic epithelial cells. They further describe the downstream events associated with the chromatin context, with the intent to identify targetable mechanisms. The study is comprehensive and overall the data are well done and well described. Since Erg is up-regulated in a large number of prostate cancers the current findings have the potential to have a broad impact on disease prognosis and management. Addressing the comments below would further strengthen the study and particularly its potential relevance for human prostate cancer.

1. It is not until discussion that the authors highlight that all aspects of their study are done in mouse models and that their conclusions are derived from these mouse studies - this should be clear in the abstract and introduction as well. For example, sentence 2 of the abstract should clearly state that the analyses of mouse models has... While this oversight is surely not intentional, as currently stated it is misleading nonetheless.

2. Along these lines, the real impact of the study is also not really evident until the discussion, namely that what is really important about the current findings is that they may help to explain why ERG — which is the most prevalent and earliest marker of prostate cancer - is not prognostic for human prostate cancer. In particular, the authors suggest that this depends on the cells in which ERG is expressed — while ERG is up-regulated in many prostate cancer cells, only a small number would be deleterious and most others would not. This is very important because it is well known that most prostate cancers do not progress to more aggressive disease. Therefore, it would be very exciting if ERG expression in a subset of prostate cells helped to identify tumors with the potential to progress.

3. Considering the point above, if the authors could extend this study to show the relevance for a human prostate cancer that would make their work much more impactful. In particular, if they could potentially perform similar analyses with human prostate cells - perhaps using their flow cytometry approach and then implanting in an orthotopic model (?) or some alternative approach to confirm that their findings are relevant to human prostate cancer, the impact of the work would be much more significant. This reviewer thinks that it would be particularly important to convince the naysayers who would simply say that ERG is a luminal marker and that the findings are an anomaly of studying prostate cancer in mice. Besides which if it is indeed the case that knowing which cells ERG is expressed is a (and perhaps even THE) key prognostic factor, this would be very important.

4. Another approach to address the relevance of their findings to human prostate cancer is a careful look at TMAs — surely there are many TMAs of ERG that could be re-probed to show the subset that co-express the relevant markers could help to highlight the specific relationship between the specific cells in which Erg is expressed and the patient outcome.

5. A third way to address this is to do single-cell analyses on ERG positive human tumors to identify whether there is a subset of cells that correspond to the cells they see in the mouse (or similar analyses).

5. At the very least, at the outset they should highlight the limitation of their study in its current form - meaning that currently all of the studies are limited to mouse. This is important because there are many difference in mouse and human prostate cancer - In particular, it is well known that mouse models (particularly those with loss of Pten) have an expansion of the basal cells whereas human prostate cancer has a loss of basal cells. This reviewer does not think that the current results are not reflective of human prostate cancer, but it is really hard to be sure that they are not simply looking at a mouse-specific phenotype.

6. In general the data regarding the chromatin context etc are interesting and informative but (in this Reviewer's opinion) the real importance of the work is helping us to understand the relevance of ERG in human prostate cancer, which the current study does not do. (So they have alot of data but miss the really key data...)

7. This reviewer does not think it is so surprising to have different (multiple) cells of origin for cancer. IF they can show that different cells of origin in human contexts have different outcomes, that would be very important.

Additional points:

1. At several points in the manuscript the authors imply that previous studies have reported NKX3.1 to be only luminal and that is not at all the case. The original papers that the authors cite clearly show some limited basal expression of NKX3.1. It would not detract from their study at all to report this accurately.

2. It is notable that the authors are inducing tumors in the NKX3.1-CreERT2 mice earlier than in published studies; this may impact differences in the relative proportion of basal cells in the current study.

2. Another thing that is not "surprising" at all that systemic deletion of K5 leads to skin tumors - in fact, it is entirely expected... I think the authors should reword this. (It is unfortunate yes but not surprising.) (Page 4 line 30).

3. What is the Ck14-CreERT2 allele they are using? — is this the original line reported (which is NOT a CRE-ET2 line it is a CRE-ERT line) or are they referring to a new Ck14-CreERT2 line? I cannot find a reference or a description of the mice either in the methods or the figure legend.

Minor points:

Page 2 - line 3 - I think this sentence in the Abstract is very unclear. I think this should be re-written (I read it several times and I am still not sure what they are trying to say)

Page 3 - I think it is very confusing that the intro talks about lung cancer - I dont see any relevance to current manuscript at all.

The intro is also missing a thoughtful discussion on cells of origin and their relevance for cancer outcome, which is the important message of the paper.

In general, the manuscript could be better digested to highlight the significant findings. The most important concepts are relegated to the discussion.

Reviewer #2:

Remarks to the Author:

In this study, the authors isolated luminal and basal cells from mouse prostates to determine the cell of origin for ERG-driven prostate cancer. They found that ERG could initiate prostate tumors in a specific subpopulation of basal cells that co-express luminal genes *Tmprss2* and *Nkx3.1*. These Basal-Lum cells, upon ERG activation, exhibit a unique chromatin state and

show elevated expression of MLL1 and DOT1L methyltransferases. Finally, the authors tested the therapeutic potential by genetically perturbing KMT2A and DOT1L, identifying them as compelling targets for existing clinical-grade pharmacologic inhibitors.

Overall, the study provides insights into the role of ERG in prostate cancer initiation and progression. It demonstrates that ERG expression in basal cells leads to a basal-to-luminal transition and expansion of an intermediate cell population, contributing to the development of luminal-type prostate cancers.

While the study provides valuable insights into the role of ERG in prostate cancer initiation, there are several major issues. Further studies addressing these issues, such as increasing more clinical relevance, and providing clear presentation, are necessary to advance our understanding of ERG in prostate cancer.

1. The manuscript includes an overwhelming amount of data, making the story difficult to follow. The authors should provide a more focused and concise presentation of the key findings. A logical description of the main findings and their significance would help readers better understand the key results from the study.
2. The introduction of the STAT3 section is very isolated and detracts from the main focus. Although it may relate to the chromatin state upon ERG expression, as a critical transcription factor ERG overexpression may alter hundreds of chromatin binding sites. This should be moved to the supplementary section.
3. The inclusion of KMT2A/MLL1-dependent findings is very distracting as well. The key experiment of Kmt2a knockdown did not significantly impair EPC-derived tumor growth, making this section appear superfluous. The DepMap score indicates no significant dependency (Figure 6D). Figures 6b-c also do not support its inclusion.
4. This study primarily utilized a mouse model (EPC) with ERG overexpression in a background of Pten loss. Previous studies by the authors (Cell Rep. 2015) showed Pten/Trp53 deletion induced tumor initiation from luminal cells. How do the authors exclude the effects of Pten loss? They also state that ERG alone is insufficient to sustain luminal fate in sorted luminal cells. A more comprehensive analysis between ERG and other co-factors would provide a broader perspective. The authors may consider combining both the STAT3 and MLL1 sections into one concise paragraph.
5. In Figure 1c: In what sites are the invasive adenocarcinomas found?
6. In Figures 1e and 1f: Why would the authors expect a difference in ERG expression between the K8 vs. K5-CRE? Is the extent of Pten loss also different in these two models? The lack of transformation in the K8-CRE model could also be a factor of low oncogene expression, rather than a preference for a cell type.
7. How are Basal-Lum cells different from basal cells? Do Basal-Lum cells express Trp63? In Extended Data fig 3, it seems like even basal cells express AR-regulated genes.
8. Moreover, how different are Basal-Lum cells from the Luminal 2 (Stem-like cells?)
9. Figure 1L: It is still a little unclear why ERG expression is so low in the K8-CRE model. Is this a feature of different strengths of the basal vs luminal promoters?
10. It seems like Fig 2j is incorrectly labeled (should be 2k).
11. Are the findings from the EPC model and the NKX3.1-CRE(ERG+) model similar in terms of disease latency? Could NKX3.1 expression in the club cells also be contributing to this phenotype given the role of club cells in disease initiation?
12. Comment: Figures 3a-c are a beautiful demonstration of this idea.
13. Is the legend on Figure 3f inverted? Shouldn't the ERG knock-out lines (P) have lower SPLum cells after a chase?
14. In Figure 3h: What are the transcriptional differences between ERG expression in a basal vs luminal cell? Moreover, why is there a decline in the % of SP cells in the luminal origin organoids? One would expect no change, but how can this decline be explained?
15. What are the DEGs between the TumLum and TumIM? (Other than lineage basal or luminal genes?)
16. A more overarching question in these models is how do tumors that originate in a basal cell but transition to a luminal cell differ from tumors that originate from a luminal cell? Transcriptionally, which one of these more closely resembles a human tumor?
17. Fig 6f: In these organoids that lack KMT2A or Dot1l, what happens to pSTAT levels (Immunoblot?)
18. Can the authors provide immunoblot evidence for activation of the stat pathway in the EPC vs WT organoids?
19. There is a need for a direct link between the findings in mouse models and their relevance to human prostate cancer to imply clinical implications. The inclusion of ERG-driven prostate cancer data in Figure 6 is unconvincing. The lack of substantial human data weakens the study's impact.

Minor issues:

1. The first paragraph introduces lung cancer, which is unusual for a manuscript focused on prostate cancer. This section should be revised or removed to maintain the manuscript's focus.
2. The Symmetric and Asymmetric IF images in Figure 3B have a high background, making it difficult to distinguish the cells.
3. The flow cytometry data in Figure 3E are very fuzzy. The authors should present clearer and higher-quality flow cytometry plots to ensure the data are interpretable and reliable.

Reviewer #3:

None

Version 1:

Decision Letter:

24th Feb 2025

Dear Professor Sawyers,

Your Article, "ERG-driven prostate cancer initiation is cell context dependent and requires KMT2A and DOT1L" has now been seen by 2 referees. You will see from their comments below that while they find your work improved, Reviewer #1 has a few remaining minor points. We are interested in the possibility of publishing your study in Nature Genetics, but would like you to implement these remaining suggestions before we make a final decision on publication.

To guide the scope of the revisions, the editors discuss the referee reports in detail within the team, including with the chief editor, with a view to identifying key priorities that should be addressed in revision and sometimes overruling referee requests that are deemed beyond the scope of the current study. In this case, please address Reviewer #1's remaining suggestions in full. We hope that you will find the prioritized set of referee points to be useful when revising your study. Please do not hesitate to get in touch if you would like to discuss these issues further.

We therefore invite you to revise your manuscript taking into account all reviewer and editor comments. Please highlight all changes in the manuscript text file. At this stage we will need you to upload a copy of the manuscript in MS Word .docx or similar editable format.

*2) If you have not done so already please begin to revise your manuscript so that it conforms to our Article format instructions, available [here](http://www.nature.com/ng/authors/article_types/index.html). Refer also to any guidelines provided in this letter.

EXTENDED DATA FIGURES

Link Redacted

We hope to receive your revised manuscript within four to eight weeks. If you cannot send it within this time, please let us know.

Sincerely,
Chiara

Chiara Anania, PhD
Associate Editor
Nature Genetics
<https://orcid.org/0000-0003-1549-4157>

Referee expertise:

Referee #1:

Referee #2:

Reviewers' Comments:

Reviewer #1 (Remarks to the Author):

This reviewer appreciates the extensive revision of the study and the addition of much new data that expands the scope of the study significantly. The authors were very responsive to the previous comments with new data and text revisions. I particularly appreciate that the K5 Cre mice that develop skin lesions, but not previously with Erg; that is an important point. I also appreciate the connection of their findings to human prostate cancer (in the new results and the text regarding previous studies). The new data regarding chromatin studies are much deeper and add a lot to the paper.

My comments below suggest further text edits (no additional experiments) that will improve the clarity and precision of the study and its endurance over time. In particular, my concerns relate to how precise the authors are in describing their data and referring to the conclusions of their study — In some (maybe even many) points the authors are unnecessarily making very definitive statements that may not be fully supported by their data or other studies in the literature. I think this detracts from this excellent study.

Specific suggestions:

One question is how their cells compare with other published studies; this might be helpful to directly address for the reader. I am thinking in particular of a study by Dong Gao's laboratory, and one on bioarchive from Flaminia Talos' laboratory. (I believe there are others). Here I think the authors could add a sentence or two that very clearly says that the cells they are talking about are similar to X or Y cells in these other studies.

Although I agree that their data strongly imply that the basal cells are the preferred cell of origin (and now nicely in humans and mice) and based on multiple types models, I think the authors are too rigid in their conclusions regarding the basal cells. Also they refer to the intermediate cells as basal cells with luminal characteristics but can they rule out that they are luminal cells with basal like features? I like the diagram in figure 1 (panel j), which implies some fluidity between these cells; I would like to see that image highlighted a bit more rather than buried in the figure. One of the reasons I think the study is too rigid in its conclusions is that they only minimally consider prostatic lobes (Extended Figure 5) - actually I was very surprised that they did not discuss lobe specificity (not in the results, or figure legend or methods). Extended Figure 1 seems to imply that they are dissecting regions of the prostate but it is not clear which ones, etc.

Another confounding factor is age of the mice at the time of analyses, which I don't think is emphasized in the results, legends or methods. (For example, what is the age of the mice that they used for the orthotopic grafting studies and what lobe(s) did they analyze? In summary, I think the prostatic lobe and age of the mice could be a factor in the preferred cell of origin.

Another concern that I have is about the various Cre drivers - I am not sure if it possible to directly compare across these models especially since they will have very different efficiencies of gene recombination following tamoxifen induction (I note below that the conditions for tumor induction are not provided in the methods (!!)). I am particularly referring to the comparison of the K8 and Nkx3.1 drivers — I suspect that they have very different recombination frequencies. I understand that the K5 mice are not viable (skin lesions) however it seems a missed opportunity to not have recovered the prostates before the mice succumbed to the skin tumors (~8 weeks or so) and done the orthotopic implantation of the sorted cells (I am not asking the authors to do this experiment, I am just saying that these data might have added a lot.) Overall, although this reviewer appreciates the use of multiple different types of models each of which is informative in its own right, the findings are still not completely definitive and therefore the conclusions should be toned down or qualified. Or perhaps add a paragraph to the discussion regarding the limitations of the data and/or the various models used..

I would also like to suggest that the authors do a very careful edit to improve the clarity and the precision of their writing throughout (especially the methods). Some examples:

Page 4, line 12 - From the references, I think they are referring to Erg models of initiation not all models? That should be clarified.

Page 6, line 21 - I believe that one of these refs is to lung; K5 has also been reported in bladder in two paper (each of these bladder papers use a different K5-Cre mouse)

Page 19, line 6 — this seems like way too strong a statement (I am referring to the choice of the word “must” in this sentence.

Methods:

The methods really need a lot of revision (probably should be re-written). Key details are missing - for example, for a study on lineage tracing they have very few details on the tumor induction. In addition, they do not specify the lobe that they are using and whether the studies were lobe specific. In most places, they do not state the age of the mice they are using, such as in the organoids culture. Age and lobe could be very important for the results that they are observing. Also there some mistakes (see p38 line 9) and various inconsistencies; eg., they refer in the text to K14 CreERT2 mice but the methods refer to the published K14CreER mice. [Page 6, line 30 — is the reference to K14 Cre mice or a typo?]

Comments on the figures:

Figure 1 — I find the labeling above the panels more confusing than helpful. I understand why the authors are labeling the sub-panels but maybe there is a better way to do this, particularly since the two sides are not parallel. Panel G is the outlier here - since it is human single cell data. I understand that the authors were addressing the comments of the reviewers, but maybe find a better place for these data. (In subsequent figures that show a human next to the human data which is very helpful - maybe add a human to panel G)

Figure 2 — similar to the point above. Also - how do the authors account for differences in cell viability among the sorted cells. (This should be addressed somewhere)

Figure 7 - the histology - IHC - is not very compelling in Panel H

Extended Figure 1 seems to imply that they are dissecting regions of the prostate but it is not clear which ones, etc. (mentioned above)

Extended Figure 2 really unfortunate that they did not sort and implant the cells from panel E,F

Extended Figure 3 — this would be a good place to relate their cells to other publications.

Extended Figure 4 — it would be nice to have some details about these patients (if it is elsewhere in the paper, include here as well).

Extended Figure 5 — this figure could highlight lobe differences better as well as efficiency of gene recombination following tamoxifen (quantification) -

Extended Figure 11 — I dont quite understand the conclusion of panel C. Are the data going in different directions?

Reviewer #2 (Remarks to the Author):

The revised manuscript on ERG-driven prostate cancer initiation presents a compelling and comprehensive study that significantly advances our understanding of the molecular mechanisms underlying prostate cancer development. The authors have diligently addressed all previous concerns and provided substantial additional evidence to support their findings, making a strong case for publication in Nature Genetics. The enhanced clinical relevance, expanded essentiality data for STAT3, KMT2A/MLL1, and DOT1L, and the intriguing mechanistic insights regarding the potential independence of STAT3 and KMT2A/DOT1L pathways significantly strengthen the overall impact of the study.

Given the comprehensive nature of the study, the robust experimental evidence, and the clinical relevance of the findings, I strongly support the publication of this manuscript in Nature Genetics. While the narrative connecting KMT2A/MLL1 and DOT1L to the overall study could be further emphasized, the work as a whole represents a substantial contribution to the field of prostate cancer research. The study's findings have important implications for both basic cancer biology and potential therapeutic strategies, and are likely to stimulate further investigations into the complex mechanisms of cancer initiation and progression.

Version 2:

Decision Letter:

Our ref: NG-A65812R1

13th Mar 2025

Dear Dr. Sawyers,

Thank you for submitting your revised manuscript "ERG-driven prostate cancer initiation is cell context dependent and requires KMT2A and DOT1L" (NG-A65812R1). We found that the paper has improved in revision, and therefore we'll be happy in principle to publish it in Nature Genetics, pending minor revisions to comply with our editorial and formatting guidelines.

We are now performing detailed checks on your paper and will send you a checklist detailing our editorial and formatting requirements soon. Please do not upload the final materials and make any revisions until you receive this additional

information from us.

Thank you again for your interest in Nature Genetics. Please do not hesitate to contact me if you have any questions.

Congratulations!

Sincerely,
Chiara

Chiara Anania, PhD
Associate Editor
Nature Genetics
<https://orcid.org/0000-0003-1549-4157>

Response to Reviewers' Comments

We thank both reviewers for their inciteful comments. A point-by-point response follows, but we first want to highlight two major changes in the revision.

To address the comments about human relevance, we have now performed extensive analyses that clearly document the presence of the intermediate (IM) basal-luminal populations, initially identified in the mouse, in ERG+ human prostate cancers. The results are in new **Fig 4 G-H** and new **Fig 5A-C** and discussed in detail below. In addition, we highlight human single cell data (**Fig 1C, 1K**) and human cell line data (new **Fig 7C**) that were mentioned in the original manuscript but not sufficiently emphasized.

We have also made substantial progress in characterizing the functional consequences of the ERG-induced chromatin landscape changes that we mentioned in the original submission. Specifically, we now have definitive evidence that STAT3, KMT2/MLL1 and DOT1L are all required to ERG-induced tumorigenicity *in vivo*. We appreciate the reviewers may have found the earlier version of the chromatin data less compelling than the cell of origin experiments, but we feel this new data is quite exciting and impactful (new **Fig 6G, 7F-H**).

With this in mind, the abstract, introduction and discussion have been extensively rewritten to reflect this narrative, beginning with the new cell of origin insights, then using those insights to identify ERG-induced chromatin changes specifically in those subpopulations, and concluding with the evidence that STAT3, MLL1 and DOT1L are critical for transformation.

Reviewer #1:

Remarks to the Author:

The study by Feng et al investigates the relationship between Erg-driven cancer initiation and cell of origin of prostate cancer. Using a series of complementary approaches based on mouse models, the authors conclude that Erg initiated tumors arise from a specific sub-population of prostatic epithelial cells. They further describe the downstream events associated with the chromatin context, with the intent to identify targetable mechanisms. The study is comprehensive and overall the data are well done and well described. Since Erg is up-regulated in a large number of prostate cancers the current findings have the potential to have a broad impact on disease prognosis and management. Addressing the comments below would further strengthen the study and particularly its potential relevance for human prostate cancer.

1. It is not until discussion that the authors highlight that all aspects of their study are done in mouse models and that their conclusions are derived from these mouse studies - this should be clear in the abstract and introduction as well. For example, sentence 2 of the abstract should clearly state that the that analyses of mouse models has... While this oversight is surely not intentional, as currently stated it is misleading nonetheless.

Thank you for calling this to our attention. We have revised the abstract and introduction to clarify the use of mouse models and to highlight the inclusion of new human data, particularly in new **Fig 5**.

2. Along these lines, the real impact of the study is also not really evident until the discussion, namely that what is really important about the current findings is that they may help to explain why ERG — which is the most prevalent and earliest marker of prostate cancer - is not prognostic for human prostate cancer. In particular, the authors suggest that this depends on the cells in which ERG is expressed — while ERG is up-regulated in many prostate cancer cells, only a small number would be deleterious and most others would not. This is very important because it is well known that most prostate cancers do not progress to more aggressive disease. Therefore, it would be very exciting if ERG expression in a subset of prostate cells helped to identify tumors with the potential to progress.

We are glad to hear the reviewer's excitement about the concept that prognosis may be linked to the cell type in which the ERG translocation originates and have emphasized this much earlier in the text. This is a challenging question to address definitively in the absence of single cell data from a large cohort of ERG+ human prostate cancers with sufficient follow-up to assess outcome (time to progression, survival).

As a proxy (i.e., inferring cell of origin from RNA signatures), we used mouse-derived signatures (i.e., a low Lum_Mut and high basal/IM, representing Basal^{Lum} origin), to interrogate RNA-seq data from two publicly available primary PCa cohorts with mature time to progression data (TCGA, Taylor 2010). This analysis shows that the ERG+ human tumors with a high Lum_Mut signature have a better outcome than those with a high IM signature (new **Fig 5C**). While this finding is consistent with the hypothesis that the cell of origin for ERG expression impacts prognosis, more definitive evidence will likely require tracing of evolutionary trajectories at a single cell level, as has been done in breast and ovarian cancer (PMID: 36289342).

3. Considering the point above, if the authors could extend this study to show the relevance for a human prostate cancer that would make their work much more impactful. In particular, if they could potentially perform similar analyses with human prostate cells - perhaps using their flow cytometry approach and then implanting in an orthotopic model (?) or some alternative approach to confirm that their findings are relevant to human prostate cancer, the impact of the work would be much more significant. This reviewer thinks that it would be particularly important to convince the naysayers who would simply say that ERG is a luminal marker and that the findings are an anomaly of studying prostate cancer in mice. Besides which if it is indeed the case that knowing which cells ERG is expressed is a (and perhaps even THE) key prognostic factor, this would be very important.

The suggestion to use flow cytometry to perform the human version of the mouse experiment (**Fig 1A-C**) is interesting. A very similar experiment was published in 2010 by Owen Witte's group, in which the combination of ERG + AR + myristoylated AKT

initiates prostate cancer from human prostate basal cells but not luminal cells following subcutaneous injection with murine urogenital sinus mesenchyme (UGSM) cells (PMID: 20671189). We have highlighted this work in the revision, while also pointing out the additional insights gained through the lineage tracing reported here.

4. Another approach to address the relevance of their findings to human prostate cancer is a careful look at TMAs — surely there are many TMAs of ERG that could be re-probed to show the subset that co-express the relevant markers could help to highlight the specific relationship between the specific cells in which Erg is expressed and the patient outcome.

We appreciate the suggestion. Angelo De Marzo's group at Hopkins has previously documented the presence of intermediate (CK5+/CK18+) cells in primary and advanced prostate cancers as well as the precursor lesion proliferative inflammatory atrophy (PIA). ERG-positive cells have also been demonstrated in PIA (PMID: 12707036; PMID: 37550827; PMID: 34341114). In addition, single cell RNA-seq data from ERG-positive human prostate cancer documents co-expression of ERG in a subset of basal cells, as well as in nearly all luminal cells (**Fig 1K**).

5. A third way to address this is to do single-cell analyses on ERG positive human tumors to identify whether there is a subset of cells that correspond to the cells they see in the mouse (or similar analyses).

Single cell analysis of ERG-positive human tumors is now highlighted in **Fig 1K**, demonstrating co-expression of ERG in a subset of basal cells, as well as in nearly all luminal cells.

5. At the very least, at the outset they should highlight the limitation of their study in its current form - meaning that currently all of the studies are limited to mouse. This is important because there are many difference in mouse and human prostate cancer - In particular, it is well known that mouse models (particularly those with loss of Pten) have an expansion of the basal cells whereas human prostate cancer has a loss of basal cells. This reviewer does not think that the current results are not reflective of human prostate cancer, but it is really hard to be sure that they are not simply looking at a mouse-specific phenotype.

This is a fair point. In the revision we have acknowledged the limitations of our study but now include new human data that supports the mouse findings.

6. In general the data regarding the chromatin context etc are interesting and informative but (in this Reviewer's opinion) the real importance of the work is helping us to understand the relevance of ERG in human prostate cancer, which the current study does not do. (So they have a lot of data but miss the really key data...)

In the revision we now provide a more fully developed series of functional studies on chromatin context. Specifically, new **Fig 6G-H** and new **Fig 7E-H** shows that STAT3,

KMT2A/MLL1 and DOT1L are all required for ERG-driven tumorigenicity. In all three cases, the effects on histology and tumor volume are profound - mirroring those seen with ERG deletion. The fact that all three proteins are “druggable” raises the prospect of near-term translational impact, which we are actively pursuing in ongoing work.

7. This reviewer does not think it is so surprising to have different (multiple) cells of origin for cancer. IF they can show that different cells of origin in human contexts have different outcomes, that would be very important.

We agree. We hope the new data in **Fig 5C** is sufficient to persuade the reviewer that this may be the case, pending confirmation by single cell analysis of larger human prostate cancer cohorts.

Additional points:

1. At several points in the manuscript the authors imply that previous studies have reported NKX3.1 to be only luminal and that is not at all the case. The original papers that the authors cite clearly show some limited basal expression of NKX3.1. It would not detract from their study at all to report this accurately.

Thank you for pointing this out. We have revised the manuscript accordingly.

2. It is notable that the authors are inducing tumors in the NKX3.1-CreERT2 mice earlier than in published studies; this may impact differences in the relative proportion of basal cells in the current study.

We induced tumors in NKX3.1-CreERT2 mice at age 7~9 weeks, which matches the age reported by others (8 weeks) (PMID: 22340597; PMID: 23434823; PMID: 23313138; PMID: 25176651; PMID: 27703144).

2. Another thing that is not “surprising” at all that systemic deletion of K5 leads to skin tumors - in fact, it is entirely expected... I think the authors should reword this. (It is unfortunate yes but not surprising.) (Page 4 line 30).

We have rewritten this section to remove the word “surprising.” That said, it is worth noting that the skin phenotype is not seen in the setting of Pten KO only. It is clearly ERG dependent. We are not aware of any prior reports linking ERG to skin phenotypes.

3. What is the Ck14-CreERT2 allele they are using? — is this the original line reported (which is NOT a CRE-ET2 line it is a CRE-ERT line) or are they referring to a new Ck14-CreERT2 line? I cannot find a reference or a description of the mice either in the methods or the figure legend.

Thank you for catching this. Indeed we used the original K14-CreERT line (PMID: 10411913; obtained from JAX, strain# 005107). We have corrected the text to clarify it is CreERT and not CreERT2.

Minor points:

Page 2 - line 3 - I think this sentence in the Abstract is very unclear. I think this should be re-written (I read it several times and I am still not sure what they are trying to say)

We have rewritten the sentence. Hopefully it is now clear.

Page 3 - I think it is very confusing that the intro talks about lung cancer - I dont see any relevance to current manuscript at all.

We understand the reviewer's point and have removed this section.

The intro is also missing a thoughtful discussion on cells of origin and their relevance for cancer outcome, which is the important message of the paper.

Thank you for pointing out this omission. We have revised the introduction to include a broader discussion of cell of origin and relevance to cancer outcome.

In general, the manuscript could be better digested to highlight the significant findings. The most important concepts are relegated to the discussion.

We have taken this comment to heart and rewritten the abstract and introduction to better digest the significant findings that are subsequently reported in the results.

Reviewer #2:

Remarks to the Author:

In this study, the authors isolated luminal and basal cells from mouse prostates to determine the cell of origin for ERG-driven prostate cancer. They found that ERG could initiate prostate tumors in a specific subpopulation of basal cells that co-express luminal genes *Tmprss2* and *Nkx3.1*. These Basal-Lum cells, upon ERG activation, exhibit a unique chromatin state and show elevated expression of *MLL1* and *DOT1L* methyltransferases. Finally, the authors tested the therapeutic potential by genetically perturbing *KMT2A* and *DOT1L*, identifying them as compelling targets for existing clinical-grade pharmacologic inhibitors.

Overall, the study provides insights into the role of ERG in prostate cancer initiation and progression. It demonstrates that ERG expression in basal cells leads to a basal-to-luminal transition and expansion of an intermediate cell population, contributing to the development of luminal-type prostate cancers.

While the study provides valuable insights into the role of ERG in prostate cancer initiation, there are several major issues. Further studies addressing these issues, such as increasing more clinical relevance, and providing clear presentation, are necessary to advance our understanding of ERG in prostate cancer.

1. The manuscript includes an overwhelming amount of data, making the story difficult to follow. The authors should provide a more focused and concise presentation of the key findings. A logical description of the main findings and their significance would help readers better understand the key results from the study.

Reviewer 1 raised a similar point. We have rewritten the abstract and introduction to provide a more focused and concise presentation of the key findings that are subsequently reported in the results.

2. The introduction of the STAT3 section is very isolated and detracts from the main focus. Although it may relate to the chromatin state upon ERG expression, as a critical transcription factor ERG overexpression may alter hundreds of chromatin binding sites. This should be moved to the supplementary section.

We acknowledge that the STAT3 section in the original submission may have detracted from the main focus, but we now have additional *in vivo* functional data providing compelling evidence that STAT3 is required for ERG-driven tumorigenicity (new **Fig 6G-H**). The effect is striking – comparable to the phenotype of ERG deletion. With this new data, coupled with the fact that STAT3 inhibitors are in clinical development for other indications, we prefer to keep the STAT3 results in the main figure.

3. The inclusion of KMT2A/MLL1-dependent findings is very distracting as well. The key experiment of Kmt2a knockdown did not significantly impair EPC-derived tumor growth, making this section appear superfluous. The DepMap score indicates no significant dependency (Figure 6D). Figures 6b-c also do not support its inclusion.

We acknowledge that the KMT2A/MLL1 data in the first submission may not have been fully developed, but we now have *in vivo* data implicating both KMT2A/MLL1 and DOT1L as dependencies in ERG-driven tumorigenicity (new **Fig 7E-H**). As with STAT3, the effects are comparable to that seen with ERG deletion. These new results confirm and extend the KMT2A knockdown findings in the original submission and, in our view, greatly enhance the impact of the work.

We are puzzled by the comment that “Kmt2a knockdown did not significantly impair EPC-derived tumor growth.” The data in the original submission and in the revision show a very clear anti-tumor phenotype (new **Fig 7E-H**).

We also do not understand the comment that “the DepMap scores indicate no significant dependency.” The analysis shows selective dependency of the two ETS-positive prostate cancer cell lines (LNCaP, VCaP) KMT2A and DOT1L relative to the seven ETS-negative prostate lines that were examined in DepMap (new **Fig 7C**). Perhaps we did a poor job of displaying this data in the original submission. We hope the revised presentation of this data makes the point more clearly.

4. This study primarily utilized a mouse model (EPC) with ERG overexpression in a background of Pten loss. Previous studies by the authors (Cell Rep. 2015) showed Pten/Trp53 deletion induced tumor initiation from luminal cells. How do the authors exclude the effects of Pten loss? They also state that ERG alone is insufficient to sustain luminal fate in sorted luminal cells. A more comprehensive analysis between ERG and other co-factors would provide a broader perspective. The authors may consider combining both the STAT3 and MLL1 sections into one concise paragraph.

The reviewer raises several points in this comment.

Regarding “excluding the effects of PTEN loss”: We have addressed this point through a side-by-side single cell analysis of ERG/Pten vs Pten alone. (ERG alone is not a useful comparison because it fails to confer a phenotype in a wild-type Pten background, PMID: 23817021). Intermediate (IM) cells are generated in both ERG/Pten and Pten contexts (**Fig 4C**). New lineage tracing data shows that IM cells in Pten alone mice arise from basal cells (new **Extended Data Fig. 8**), consistent data communicated to us by Cedric Blanpain’s group. However, ERG is essential for invasive adenocarcinoma (Pten alone mice develop intraductal hyperplasia and PIN - not invasive cancer).

Regarding the Pten/Trp53 interaction: The reviewer is correct that we previously showed Pten/Trp53 co-deletion preferentially induces tumors from luminal cells versus basal cells (PMID: 34353917). As with ERG, Trp53 deletion alone is capable of initiating cancer – only the combination can score in readouts of invasion. To account for the difference in preferred cell of origin, we hypothesize that ERG and Trp53 specify a basal versus luminal origin, respectively, whereas Pten initiates expansion of an IM population in both.

Regarding the comment that “ERG alone is insufficient to sustain luminal fate in sorted luminal cells”: We assume the reviewer is referring to **Fig 3H** (right panel), where sorted luminal cells (regardless of ERG status) are not sustained at a high percentage during passage in organoid culture. We believe this is a consequence of organoid culture growth conditions, which favor proliferation of basal cells and stem-like luminal cells (L2) over secretory luminal cells (L1) that account for ~90% of the luminal cells in the mouse prostate gland. The other panels in **Fig 3** (pulse chase and lineage tracing data) definitively show that ERG drives luminal cell fate from basal and IM cells. The text has been revised to clarify this.

Regarding “combining both the STAT3 and MLL1 sections”: As discussed earlier, we feel the new data showing that ERG-driven *in vivo* tumorigenicity is dependent on STAT3, KMT2A/MLL1 and DOT1L is an important new section, which has been rewritten accordingly.

5. In Figure 1c: In what sites are the invasive adenocarcinomas found?

This panel shows invasive adenocarcinoma in the anterior prostate lobe, the site of orthotopic transplantation. This detail is now included in the method section and figure legend.

6. In Figures 1e and 1f: Why would the authors expect a difference in ERG expression between the K8 vs. K5-CRE? Is the extent of Pten loss also different in these two models? The lack of transformation in the K8-CRE model could also be a factor of low oncogene expression, rather than a preference for a cell type.

The ERG expression data in **Fig 1E-F** is at the endpoint of experiment (40 wks) and therefore reflects the fact that K5+ basal cells expand to form invasive K8+ tumors whereas K8+ luminal cells do not. Prior to performing this experiment, we compared the efficiency of Cre-mediated excision by the K5- and K8-Cre drivers and found no difference (in fact, K8-Cre shows slightly higher activity) (**Extended Data Fig. 2H**). In all cases, ERG expression is driven by the R26 locus, so there is no difference in expression on a per cell basis.

7. How are Basal-Lum cells different from basal cells? Do Basal-Lum cells express Trp63? In Extended Data fig 3, it seems like even basal cells express AR-regulated genes.

Basal^{Lum} cells are a subset of basal cells (defined by many criteria, including UMAP localization to a basal cluster) that also express several genes more generally associated with luminal identity, such as *Tmprss2* and *Nkx3.1*. We label these cells basal because: (i) their overall gene expression profiles overlaps with canonical basal cells in UMAP space and (ii) their morphology and tissue localization matches that of canonical basal cells (flat cells, adjacent to basement membrane). Basal^{Lum} cells do express Trp63 and a subset of AR regulated genes (**Extended Data Fig 3**). Of note, these cells appear similar to a subset of basal cells published last month by Dong Gao's lab (PMID: 39537874).

8. Moreover, how different are Basal-Lum cells from the Luminal 2 (Stem-like cells?)

Basal^{Lum} cells are clearly distinct from luminal 2 cells for three reasons: (i) different spatial localization, (ii) different morphology, and (iii) distinct transcriptomes (separate UMAP locations).

9. Figure 1L: It is still a little unclear why ERG expression is so low in the K8-CRE model. Is this a feature of different strengths of the basal vs luminal promoters?

See answer to comment #6. ERG expression is driven by the R26 locus and therefore identical across Cre drivers.

10. It seems like Fig 2j is incorrectly labeled (should be 2k).

Thank you for pointing this out. We have fixed the labeling.

11. Are the findings from the EPC model and the NKX3.1-CRE(ERG+) model similar in terms of disease latency? Could NKX3.1 expression in the club cells also be contributing to this phenotype given the role of club cells in disease initiation?

The disease latency from the *Nkx3-1-CreER^{T2}* driver is longer than from the *Pb-Cre* driver (*EPC*) (~27wk of median PCa-free survival by the *Nkx3-1* driver vs ~8wk by *Pb-Cre*). This is expected because *ERG* is induced transiently in *Nkx3-1-CreER^{T2}* in adulthood but constitutively and earlier (starting postnatally) in *EPC* mice (*Pb-Cre*).

Several lines of evidence suggest that club cells are unlikely to contribute to the phenotype seen in the *Nkx3.1-Cre ERG+* model. First, single cell RNA data fail to show expression *Nkx3.1* in club cells (this study and PMID 32355025). Second, *Nkx3.1* protein is not detectable by IHC in proximal *Trop2+* luminal 2 cells (which share club cell features) (**Extended Data Fig. 5A** from this study and Crowley 2020, PMID: 32915138). Finally, club cells are efficiently labeled using the *K8-CreER^{T2}* driver (**Extended Data Fig. 5A**) but these mice do not develop *ERG*-driven prostate cancer.

12. Comment: Figures 3a-c are a beautiful demonstration of this idea.

Thank you for the complement.

13. Is the legend on Figure 3f inverted? Shouldn't the *ERG* knock-out lines (P) have lower SPLum cells after a chase?

Correct. Thanks for catching this oversight. We have now fixed it.

14. In Figure 3h: What are the transcriptional differences between *ERG* expression in a basal vs luminal cell? Moreover, why is there a decline in the % of SP cells in the luminal origin organoids? One would expect no change, but how can this decline be explained?

To address the first question, we performed transcriptomic profiling of sorted basal vs luminal cells from EP organoids +/- *ERG* sgRNA knockdown, then defined *ERG*-dependent differentially expressed genes (DEGs). This analysis revealed that many *ERG* DEGs are specific to the cell lineage in which *ERG* is expressed (basal or luminal) (see reviewer **Fig A**). Thus, although *ERG* clearly drives luminal fate (**Fig 3**), the "global" gene programs induced by *ERG* vary depending on cell lineage.

Fig A. *ERG* DEG in basal vs luminal organoid cells.

As for the second question (“why is there a decline in the % of SP cells in the luminal origin organoids?”), we addressed this in a response to comment #4. Briefly, current organoid culture conditions do not support expansion of pure luminal cell populations (PMID: 25201529). The data in the right panel of **Fig 3H** shows that ERG does not “rescue” luminal cell growth in this assay; however, ERG does drive luminal differentiation when this experiment is done using isolated basal cells (**Fig 3H**, left panel).

15. What are the DEGs between the TumLum and TumIM? (Other than lineage basal or luminal genes?)

Through single cell analysis, we identified MYC, NF- κ B, Stemness, and RNA processing/translation upregulated in Tumor-IM cells. Interferon and immunity pathways were enriched in Tumor-Lum cells (now named Lum_Mut), and pathways with stemness features were downregulated (new **Fig 4E, G**).

16. A more overarching question in these models is how do tumors that originate in a basal cell but transition to a luminal cell differ from tumors that originate from a luminal cell? Transcriptionally, which one of these more closely resembles a human tumor?

This is an intriguing question but requires data from a well-documented luminal cell of origin mouse model to make a direct comparison. Even if such a dataset exists, another challenge would be to adjust for potentially different oncogenic drivers.

17. Fig 6f: In these organoids that lack KMT2A or Dot1l, what happens to pSTAT levels (Immunoblot?)

We measured pStat3 levels in prostate organoids lacking *Kmt2a* or *Dot1l* and did not observe substantial differences between knockout and control (**Fig B**).

Fig B. pStat3 immunoblot in indicated organoids.

18. Can the authors provide immunoblot evidence for activation of the stat pathway in the EPC vs WT organoids?

We measured pStat3 levels in prostate organoids derived from *WT*, *PYC* (*Pten* only) and *EPC* mice and did not observe substantial differences (**Fig C**), suggesting that the requirement for STAT3 for ERG-driven tumorigenicity is likely a consequence of TF cooperativity at the chromatin level, rather than an effect of ERG on STAT activation.

Fig C. pStat3 immunoblot in indicated organoids.

19. There is a need for a direct link between the findings in mouse models and their relevance to human prostate cancer to imply clinical implications. The inclusion of ERG-driven prostate cancer data in Figure 6 is unconvincing. The lack of substantial human data weakens the study's impact.

The point was also raised by reviewer 1. See response to Rev 1, comments 2 thru 6.

Minor issues:

1. The first paragraph introduces lung cancer, which is unusual for a manuscript focused on prostate cancer. This section should be revised or removed to maintain the manuscript's focus.

Reviewer 1 made the same comment. We have removed this section.

2. The Symmetric and Asymmetric IF images in Figure 3B have a high background, making it difficult to distinguish the cells.

We have replaced the original images with higher quality confocal images.

3. The flow cytometry data in Figure 3E are very fuzzy. The authors should present clearer and higher-quality flow cytometry plots to ensure the data are interpretable and reliable.

We have replaced the original panels with higher resolution plots.

Response to Reviewers' Comments

Reviewer #1 (Remarks to the Author):

This reviewer appreciates the extensive revision of the study and the addition of much new data that expands the scope of the study significantly. The authors were very responsive to the previous comments with new data and text revisions. I particularly appreciate that the K5 Cre mice that develop skin lesions, but not previously with Erg; that is an important point. I also appreciate the connection of their findings to human prostate cancer (in the new results and the text regarding previous studies). The new data regarding chromatin studies are much deeper and add a lot to the paper.

My comments below suggest further text edits (no additional experiments) that will improve the clarity and precision of the study and its endurance over time. In particular, my concerns relate to how precise the authors are in describing their data and referring to the conclusions of their study — In some (maybe even many) points the authors are unnecessarily making very definitive statements that may not be fully supported by their data or other studies in the literature. I think this detracts from this excellent study.

Thank you for the thorough review of our revised manuscript. We are pleased by the positive feedback but also appreciate the need to qualify several of the conclusions. We have revised the manuscript accordingly as described below.

Specific suggestions:

One question is how their cells compare with other published studies; this might be helpful to directly address for the reader. I am thinking in particular of a study by Dong Gao's laboratory, and one on bioarchive from Flaminia Talos' laboratory. (I believe there are others). Here I think the authors could add a sentence or two that very clearly says that the cells they are talking about are similar to X or Y cells in these other studies.

We are aware of the Gao and Talos manuscripts and agree that all of us may be describing the same subset of basal cells. We have added a sentence to this effect on page 8, line 19-21 and cited the manuscripts.

Although I agree that their data strongly imply that the basal cells are the preferred cell of origin (and now nicely in humans and mice) and based on multiple types models, I think the authors are too rigid in their conclusions regarding the basal cells. Also they refer to the intermediate cells as basal cells with luminal characteristics but can they rule out that they are luminal cells with basal like features? I like the diagram in figure 1 (panel j), which implies some fluidity between these cells; I would like to see that image highlighted a bit more rather than buried in the figure. One of the reasons I think the study is too rigid in its conclusions is that they only minimally consider prostatic lobes (Extended Figure 5) - actually I was very surprised that they did not discuss lobe specificity (not in the results, or figure legend or methods). Extended Figure 1 seems to imply that they are dissecting regions of the prostate but it is not clear which ones, etc.

We appreciate the importance of not being too rigid and have made edits throughout the manuscript to qualify our conclusions.

Regarding lobe specificity: we apologize for this oversight and have clarified these details at relevant places in the text, methods and figure legends. We have also included this topic in the new “limitations” paragraph in the discussion (**page 21, lines 27-29**), acknowledging that our study was not designed to address the possibility of lobe-specific differences.

Regarding “basal with luminal features” vs “luminal cells with basal features”, we call Basal^{Lum} cells basal with luminal features because we feel the morphologic features (flat, adjacent to basement membrane), cytokeratin profile (single positive for basal cytokeratin K5, **Fig. 1J**), and transcriptome (colocalize with other basal cells in the UMAP space, **Fig. 1G**) support the former. We avoid calling IM cells one state vs another because they represent a mixture of multiple lineage states (basal, luminal, hillock and club) and no longer simply a basal or luminal state.

We have revised Fig 1 to highlight panel J.

Another confounding factor is age of the mice at the time of analyses, which I don't think is emphasized in the results, legends or methods. (For example, what is the age of the mice that they used for the orthotopic grafting studies and what lobe(s) did they analyze? In summary, I think the prostatic lobe and age of the mice could be a factor in the preferred cell of origin.

Thank you for pointing this out. We have now specified the age of mice in all experiments (8-10 weeks) as well as the lobes that were included in the analysis. For the orthograft experiments, luminal or basal cells were isolated by flow cytometry from whole prostates of donor mice (all lobes), then implanted into the anterior lobe of recipient mice. The new limitations paragraph in the discussion acknowledges prostate lobe and age of mice as variables that may impact cell of origin, and the fact that these are not addressed by our study.

Another concern that I have is about the various Cre drivers - I am not sure if it possible to directly compare across these models especially since they will have very different efficiencies of gene recombination following tamoxifen induction (I note below that the conditions for tumor induction are not provided in the methods (!!)). I am particularly referring to the comparison of the K8 and Nkx3.1 drivers — I suspect that they have very different recombination frequencies. I understand that the K5 mice are not viable (skin lesions) however it seems a missed opportunity to not have recovered the prostates before the mice succumbed to the skin tumors (~8 weeks or so) and done the orthotopic implantation of the sorted cells (I am not asking the authors to do this experiment, I am just saying that these data might have added a lot.) Overall, although this reviewer appreciates the use of multiple different types of models each of which is informative in its own right, the findings are still not completely definitive and therefore

the conclusions should be toned down or qualified. Or perhaps add a paragraph to the discussion regarding the limitations of the data and/or the various models used..

These are important points which we now acknowledge in the new “limitations” paragraph, as suggested. We have also added a new paragraph in results (pages 9-10) that directly addresses this point by comparing YFP and ERG expression across the K8 and Nkx3.1 drivers (new **Extended Data Fig 4**). While Nkx3.1 is a more efficient driver, the results show clear evidence of K8-driven recombination, including acquisition of luminal cell state with low/absent Nkx3.1 expression, likely a consequence of Pten loss and subsequent shift to a more stem-like (L2) state.

I would also like to suggest that the authors do a very careful edit to improve the clarity and the precision of their writing throughout (especially the methods). Some examples: Page 4, line 12 - From the references, I think they are referring to Erg models of initiation not all models? That should be clarified.

Thanks for catching this. We have made the appropriate edit.

Page 6, line 21 - I believe that one of these refs is to lung; K5 has also been reported in bladder in two paper (each of these bladder papers use a different K5-Cre mouse)

Correct. We have rephrased this sentence accordingly.

Page 19, line 6 — this seems like way too strong a statement (I am referring to the choice of the word “must” in this sentence.

We agree. We changed “must” to “may.”

Methods:

The methods really need a lot of revision (probably should be re-written). Key details are missing - for example, for a study on lineage tracing they have very few details on the tumor induction. In addition, they do not specify the lobe that they are using and whether the studies were lobe specific. In most places, they do not state the age of the mice they are using, such as in the organoids culture. Age and lobe could be very important for the results that they are observing. Also there some mistakes (see p38 line 9) and various inconsistencies; eg., they refer in the text to K14 CreERT2 mice but the methods refer to the published K14CreER mice. [Page 6, line 30 — is the reference to K14 Cre mice or a typo?]

Thanks for the careful reading of methods. We have rewritten several sections to provide all these necessary details. We have also fixed the text to clarify that we used K14CreER mice (not K14CreERT2).

Comments on the figures:

Figure 1 — I find the labeling above the panels more confusing than helpful. I understand why the authors are labeling the sub-panels but maybe there is a better way

to do this, particularly since the two sides are not parallel. Panel G is the outlier here - since it is human single cell data. I understand that the authors were addressing the comments of the reviewers, but maybe find a better place for these data. (In subsequent figures that show a human next to the human data which is very helpful - maybe add a human to panel G)

Thanks for this feedback. We removed the labels above the figures and added human or mouse cartoons next to relevant panels to make it easier for readers to follow the data flow.

Figure 2 — similar to the point above. Also - how do the authors account for differences in cell viability among the sorted cells. (This should be addressed somewhere)

We added a sentence in the results (**page 6, line 7-9**) that acknowledges the fragility of luminal cells in flow cytometry-based isolation protocols.

Figure 7 - the histology - IHC - is not very compelling in Panel H

Panel H shows the effect of Menin deletion on the ERG phenotype in the orthotopic allograft assay. The conclusion is that ERG-driven tumorigenicity remains intact (tumor area, by H&E) despite strong evidence (by IHC) that Menin expression is absent in the ERG+ orthografts that were targeted with Men1 sgRNAs. We have shared the data with multiple colleagues and feel the result is clear with the images shown. If this explanation does not suffice, perhaps the reviewer can be more specific about this point.

Extended Figure 1 seems to imply that they are dissecting regions of the prostate but it is not clear which ones, etc. (mentioned above)

As mentioned above, these issues are now addresses through edits in results, methods and figure legends throughout.

Extended Figure 2 really unfortunate that they did not sort and implant the cells from panel E,F

We agree this was a missed opportunity. Samples were not harvested in a way that could allow us to address this point now.

Extended Figure 3 — this would be a good place to relate their cells to other publications.

As mentioned above, we addressed this in the main text (**page 8, line 19-21**).

Extended Figure 4 — it would be nice to have some details about these patients (if it is elsewhere in the paper, include here as well).

We added details in methods that all human transcriptomic analyses are from patients with localized primary prostate cancer, and included this in the relevant figure legends.

Extended Figure 5 — this figure could highlight lobe differences better as well as efficiency of gene recombination following tamoxifen (quantification) –

This point is discussed above and addressed in the new Extended Data Figure Fig 4.

Extended Figure 11 — I don't quite understand the conclusion of panel C. Are the data going in different directions?

The “different directions” reflects the fact that luminal signatures in ERG+ patients are associated with favorable outcome (consistent with the data in Main Fig 5C, using different luminal signatures), whereas basal/IM signatures are associated with poor outcome. It is worth noting that the negative prognostic impact of the basal/IM signatures is not ERG-specific, likely reflecting the fact that this cell state can emerge in other genotypes.

Reviewer #2 (Remarks to the Author):

The revised manuscript on ERG-driven prostate cancer initiation presents a compelling and comprehensive study that significantly advances our understanding of the molecular mechanisms underlying prostate cancer development. The authors have diligently addressed all previous concerns and provided substantial additional evidence to support their findings, making a strong case for publication in Nature Genetics. The enhanced clinical relevance, expanded essentiality data for STAT3, KMT2A/MLL1, and DOT1L, and the intriguing mechanistic insights regarding the potential independence of STAT3 and KMT2A/DOT1L pathways significantly strengthen the overall impact of the study.

Given the comprehensive nature of the study, the robust experimental evidence, and the clinical relevance of the findings, I strongly support the publication of this manuscript in Nature Genetics. While the narrative connecting KMT2A/MLL1 and DOT1L to the overall study could be further emphasized, the work as a whole represents a substantial contribution to the field of prostate cancer research. The study's findings have important implications for both basic cancer biology and potential therapeutic strategies, and are likely to stimulate further investigations into the complex mechanisms of cancer initiation and progression.

We thank the reviewer for the favorable comments.